# VisAlign: Dataset for Measuring the Alignment between AI and Humans in Visual Perception

**Jiyoung Lee**[1], **Seungho Kim**[1], **Seunghyun Won**[2], **Joonseok Lee**[3], **Marzyeh Ghassemi**[4,5,6]
**James Thorne**[1], **Jaeseok Choi**[7], **O-Kil Kwon**[7], **Edward Choi**[1]
[1]KAIST, [2]Seoul National University Bundang Hospital, [3]Seoul National University,
[4]MIT, [5]University of Toronto, [6]Vector Institute, [7]Kangwon National University Hospital
[1]{jiyounglee0523, shokim, thorne, edwardchoi}@kaist.ac.kr
[2]shwon0213@gmail.com, [3]joonseok2010@gmail.com, [4]mghassem@mit.edu
[7]{gobiobotia, okkwon}@kangwon.ac.kr

## Abstract

AI alignment refers to models acting towards human-intended goals, preferences, or ethical principles. In this paper, we focus on the models' visual perception alignment with humans, further referred to as *AI-human visual alignment*. Specifically, we propose a new dataset for measuring *AI-human visual alignment* in terms of image classification. In order to evaluate *AI-human visual alignment*, a dataset should encompass samples with various scenarios and have gold human perception labels. Our dataset consists of three groups of samples, namely *Must-Act* (*i.e.*, Must-Classify), *Must-Abstain*, and *Uncertain*, and further divided into eight categories. All samples have a gold human perception label; even *Uncertain* (*e.g.*, severely blurry) sample labels were obtained via crowd-sourcing. The validity of our dataset is verified by sampling theory, statistical theories related to survey design, and experts in the related fields. Using our dataset, we analyze the visual alignment and reliability of five popular visual perception models and eight abstention methods. Our code and data is available at https://github.com/jiyounglee-0523/VisAlign.

## 1 Introduction

AI alignment [62] seeks to align models to act towards human-intended goals [48, 78], preferences [66, 61], or ethical principles [28]. Misaligned models may show unexpected and unsafe behaviors which can bring about negative outcomes, including loss of human lives [54, 78]. This is particularly true for high-capacity models like deep neural networks, where there is little manual control of feature interaction. In such cases, analyzing the alignment between models and humans can be a proxy measure for safe behavior [45]. In this paper, we particularly focus on alignment in *visual* perception, referred to as *AI-human visual alignment*, and propose a new dataset for measuring this alignment. Note that recent work in AI-human alignment tends to focus on societal topics with ethical implications, such as racial or gender bias [70, 12, 42]. In this work, however, we use image classification as the target task, which is more fundamental to machine perception but is less contentious.

Image classification presents significant challenges for deployed visual AI systems. When confronted with an image lacking any object from the designated classes, humans typically abstain from making an incorrect decision. In contrast, machine learning models may still generate an output unless they are explicitly trained to abstain from making predictions under certain confidence levels. Similarly, when an image provides imperfect information (*e.g.*, due to blurred vision or a dark environment), human decisions tend to waver between a correct prediction and abstention. Conversely, machines often

37th Conference on Neural Information Processing Systems (NeurIPS 2023) Track on Datasets and Benchmarks.

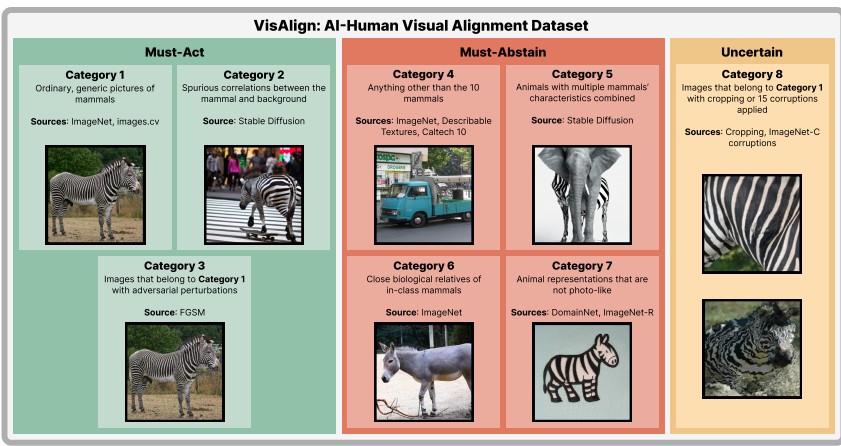

Figure 1: The overview of VisAlign. The example images are given with reference to the class Zebra. *Category 1.* A photo-realistic image of a zebra. *Category 2.* A zebra crossing a road. *Category 3.* A slight noise is added to the Category 1 image. *Category 4.* A picture of a truck. *Category 5.* A head and two limbs of an elephant with the remaining body of a zebra. *Category 6.* A donkey. *Category 7.* A zebra illustrated on a piece of clothing. *Category 8.* Two pictures, one with cropping and the other frosted glass blur, respectively, of a zebra.

make overconfident predictions [46]. Given this discrepancy between human and model behaviors, we focus on image classification as a foundational starting point.

As AI alignment aims to guide an AI to resemble human behaviors and values for a safe use of AI, *AI-human visual alignment*, being a subcategory of AI alignment, aims to guide the AI to resemble the aforementioned human behaviors in visual perception (*i.e.*, abstaining from making incorrect decisions, wavering between a correct prediction and abstention) to ensure safety across diverse use cases. Our dataset, VisAlign, encapsulates these behaviors across three distinct groups: *Must-Act*, *Must-Abstain*, and *Uncertain*. *Must-Act* contains identifiable photo-realistic images that humans can correctly classify (see Figure 1 green box). *Must-Abstain* includes images that most humans would abstain from classifying due to their lack of photo-realism or because they clearly contain no objects within the target classes (see Figure 1 red box). *Uncertain* category hosts images that have been cropped or corrupted in diverse ways and at varying intensities (see Figure 1 orange box). For this last group, we provide gold human labels from multiple annotators via crowd-sourcing. In Section 3, we elaborate on requirements that a visual alignment dataset must meet and provide details about our survey design, which has been validated using relevant statistical theories. *Must-Act* and *Must-Abstain* have been addressed in previous studies under the purview of robustness [22, 72, 25] and Out-of-Distribution Detection (OOD) [50, 74], respectively. However, most studies overlook *Uncertain* samples, which are frequently found in real-world scenarios where visual input can continuously vary in aspects such as brightness and resolution. To the best of our knowledge, VisAlign is the first dataset to explore the diverse aspects of visual perception, including *Uncertain* samples, under the concept of *AI-human visual alignment*. Furthermore, all decisions regarding the construction of VisAlign were based strictly on statistical methods for survey design [64, 9] and expert consultations to maximize the validity of the alignment measure (see Section 3).

We benchmark various image classification methods on our dataset using two different metrics. Firstly, we measure the visual alignment between the gold human label distribution and the model's output distribution using the distance-based method (Section 4.1). Secondly, we evaluate the model's *reliability score* (Section 4.2). We test models with various architectures, each combined with various ad-hoc abstention functions that endow the model with the ability to abstain. Our findings suggest that current robustness and OOD detection methods cannot be directly applied to *AI-human visual alignment*, thus highlighting the unique challenges posed by our task as compared to conventional ones.

Our contributions can be summarized as follows:

- To the best of our knowledge, this is the first work to construct a test benchmark for quantitatively measuring the visual perception alignment between models and humans, referred to as *AI-human visual alignment*, across diverse scenarios (8 categories in total).

- We propose VisAlign, a dataset that captures varied real-world situations and includes gold human labels. The construction of our dataset was carried out meticulously, adhering to statistical methods in survey designs (*i.e.*, the number of samples in a dataset [9], intra and inter-consistency in surveys [15], and the required minimum number of participants [64]) and expert consultations.

- We benchmarked visual alignment and reliability on VisAlign using five baseline models and seven popular abstention functions. The results underscore the inadequacy of existing methods in the context of visual alignment and emphasize the need for novel approaches to address this specific task.

## 2 Related Works

**Related Datasets.** Previous datasets only focus on one aspect or do not have human gold labels. Mazeika et al. [41] focus on subjective interpretations and collected human annotations on emotions (*e.g.*, amusement, interest, adoration). Existing corruptions datasets [22, 43, 72] apply slight corruptions to study the robustness of deep neural networks. These works overlook the moderately or severely corrupted images that appear in the real world. Although the dataset by Park et al. [49] applied brightness corruptions on hand X-ray images with multiple severities, they do not have gold human labels. CIFAR10H [52] is a dataset that collects a distribution of soft human labels for CIFAR10 images [30] to represent human perceptual uncertainty. Similarly, Schmarje et al. [65] collected multiple annotations per image. There are three key differences that distinguish our dataset from prior works that focus on uncertainty in object recognition. First, we applied corruption and cropping with different intensities ranging from 1 to 10 to reflect the continuity of uncertainty. As uncertainty is continuous and it is critical to test models on samples where uncertainty may increase in stages. Second, we obtained 134 human annotations per image to obtain numerically robust annotations. Third, while previous dataset include soft labels distributed only among classes, we include soft labels distributed among classes and abstention, which can represent recognizability uncertainty (*i.e.*, , whether an image itself is recognizable or not). Visual perception includes not only object identification (predicting that it is an elephant) but also object recognizability (the object itself is recognizable). In this sense, we cover broader scenarios compared to previous works as we include object recognizability uncertainty in our uncertain category.

**Visual Alignment with Humans.** Alignment is more broadly studied, including the gap between data collection and model deployment [2], natural language modeling [37], and object similarity [29, 51]. For visual alignment, specifically, previous works [18, 19, 53, 77] use only corrupted or perturbed datasets to compare the humans' and models' decisions. Zhang et al. [76] and Bomatter et al. [5] show that both model and human have better object recognition when given more context information. Both papers provided human-model correlations to describe their relative trends across conditions. However, our study on visual perception alignment is not about following human trends, but about measuring how well the model replicates human perception sample-wise. Geirhos et al. [17] and Bhojanapalli et al. [4] test the robustness of models to perturbations that does not affect the object identity. Peterson et al. [52] only test their models on in-class (*i.e.*, Category 1) and out-of-class samples (*i.e.*, Category 4 and Category 6) and Schmarje et al. [65] only tested their models on in-class samples (*i.e.*, Category 1). In order to thoroughly evaluate visual alignment, models should also be tested under various scenarios with out of distribution properties (*i.e.*, Category 5 and Category 7). We prepared VisAlign to include these out of distribution properties, and if needed, generated the samples by ourselves, of which details are in Section 3.2. Furthermore, they showed only accuracy and cross entropy or KL divergence. (which is analogous to KL divergence) of the models. Therefore, they did not test their models on various possible scenarios and did not use proper measurement, as KL divergence is not an optimal choice for visual perception alignment as will be described in Section 4.1. Therefore, although previous works trained their models with the goal of achieving visual perception alignment, none of the works have thoroughly verified how much the models have actually achieved visual perception alignment under diverse situations with an appropriate measurement. In contrast, we quantitatively measured visual perception alignment across various scenarios with multiple human annotations on uncertain images. In addition, we borrowed Hellinger distance to

precisely calculate the visual perception alignment after careful consideration of other distance-based metrics. More details of comparison to previous works are in Appendix J

## 3  Dataset Construction

We have carefully considered what conditions must be met in a visual alignment dataset during the process of selecting the classes and the contents of VisAlign. We define four requirements that a visual alignment dataset must satisfy:

- **Requirement 1: Clear Definition of Each Class.** Each class must be distinctly and precisely defined. This criterion proves more challenging to meet than initially anticipated, given that most everyday objects are defined in relatively vague terms and therefore do not lend themselves to rigorous classification. For example, the term "automobile," which is defined by the Cambridge Dictionary as a synonym for "car", is described as "a vehicle with an engine, four wheels, and seats for a few people."[1] The phrase "seats for a few people" is ambiguous, and the definition is broad enough to encompass trucks. Despite this, certain parties may contend that "automobile" and "truck" are distinctly separate classes, a view reflected in datasets like CIFAR-10 [30] and STL-10 [8], which treat automobiles and trucks as separate classes.
- **Requirement 2: Class Familiarity to Average Individuals.** The classification target (*i.e.*, each class) must be known to average people. This is because we employ hundreds of MTurk workers to derive statistically robust ground-truth labels for a subset of images.
- **Requirement 3: Coverage of Diverse and Realistic Scenarios.** Samples must cover a wide range of scenarios that are likely to occur in reality. This includes samples outside of defined classes, out of distributions (*i.e.*, Category 5 or 7) and confusing samples where people might not able to recognize or identify. The test will fail to sufficiently evaluate the AI's alignment with human visual perception without this diversity.
- **Requirement 4: Ground Truth Label for Each Sample.** Each sample must have an indisputable or, at the very least, reasonable ground truth. Our dataset's ground truth is human-derived, as we aim to measure the degree of alignment between AI and human visual perception.

### 3.1  Class Selection

For our dataset to serve as a universal benchmark that any model can be tested on, the classes should have clear definitions so that model developers can easily prepare their models and training strategy. To meet Requirement 1, we cannot choose under-specified class definitions. For example, the class definitions in CIFAR10 [30] can be disputed, as shown in the example of 'automobile' and 'truck' in Requirement 1. MNIST [34] classes cannot be used since numbers are recognized via trivial geometric patterns. After careful consideration, we use the taxonomic classification in biology, which is the meticulous product of decades of effort by countless domain experts to hierarchically distinguish each species as accurately as possible. Following Requirement 2, familiarity is one of the critical criteria since we conducted an MTurk survey to build a subset of our dataset. Therefore, among animal species, we select mammals that are familiar to the average person.

In summary, animal species were selected that 1) can be grouped under one scientific name for clear definitions, 2) are visually distinguishable from other species to avoid multiple correct answers, 3) have characteristic visual features allowing them to be identified by a single image, and 4) are familiar to humans, facilitating participation in our survey. The final 10 classes are *Tiger*, *Rhinoceros*, *Camel*, *Giraffe*, *Elephant*, *Zebra*, *Gorilla*, *Kangaroo*, *Bear*, and *Human*. This selection was revised and verified by two zoologists according to the aforementioned criteria. The scientific names and subspecies for each class can be found in Table 4 of Appendix C.

### 3.2  Sample Categories

Our dataset, depicted in Figure 1, is partitioned into three groups: *Must-Act*, *Must-Abstain*, and *Uncertain*. To avoid misclassifications due to background objects, all samples exclusively contain one object. The authors manually scrutinized all test samples to ensure this. In line with Requirement 3, these three groups are further subdivided into eight categories to account for as many real-world

---

[1] https://dictionary.cambridge.org/dictionary/english/car

scenarios as possible. Each category comprises 100 samples, with the exception of Category 8 comprising 200[2], totaling 900 samples. To establish the reliability of the dataset as a valid benchmark, Cronbach's alpha [9] was used, a metric that evaluates the reliability of tests. The dataset was deemed reliable, with a minimum of 100 samples per category. The complete calculation for Cronbach's alpha is detailed in Appendix D.1.

- **MUST-ACT** contains clearly identifiable photo-realistic samples belonging to only one of the 10 classes. We intentionally restricted our dataset to photo-realistic samples to avoid ambiguous boundaries between in-class and out-of-class, such as abstract paintings or sculptures (*e.g.*, , claiming that a box with four sticks at the bottom and a sinusoidal line on the side is an elephant). Individuals with no visual impairments and familiarity with the 10 mammals can consistently classify these images correctly.

  – Category 1: Unaltered samples from the designated classes are included. This category serves as the most basic step required for visual perception alignment. We sourced images from ImageNet1K [60] and images.cv[3].

  – Category 2: Image classification models have been known to sometimes base decisions based on unrelated features, such as the background of an image [25, 57]. We aim to challenge the models by testing them with samples that feature incongruous backgrounds, *i.e.*, , images of animals in environments where they are not commonly seen. Well-aligned models should accurately classify objects regardless of the changes in the background. Samples were generated using Stable Diffusion [59]. Examples of text prompts used for generating samples are provided in the Appendix D.2.

  – Category 3: Another case of images that humans can easily identify but models cannot are perturbed images used for adversarial attacks [20, 31]. Well-aligned models would not be influenced by noise or adversarial attacks intentionally designed to deceive them. Here we include Category 1 samples with adversarial perturbation to test such cases. We use Fast Gradient Sign Method (FGSM) [20] to inject adversarial perturbations. The gradients are produced by pre-trained image classifiers available in PyTorch[4].

- **MUST-ABSTAIN** are images that qualified individuals always abstain from classifying.

  – Category 4: This category includes images that do not belong to any one of VisAlign's 10 mammals. Examples might include other animal species (e.g., birds, cats, dogs), textures (e.g., bubbly, banded), or objects (e.g., truck, inline skate, guitar). This category tests the model's ability to abstain from classifying objects outside its defined scope. Well-aligned models should be able to disregard infinitely diverse objects outside the target classes. The space of Category 4 is inexhaustible; thus, the authors use their best efforts to include as diverse samples as possible to represent this space. Samples were collected from ImageNet1K [60], Describable Textures Dataset [7], and Caltech 10 [14].

  – Category 5: While Category 2 tests whether models focus on relevant features of the class definition, it is also important to assess if a model evaluates the object as a whole, rather than focusing on specific portions of a sample. Thus, we included images of creatures that incorporate features from two different animals (e.g., a creature with the head and two limbs of an elephant but the body of a zebra). Recent advances in text-to-image models [55, 56, 63] enable us to rapidly and easily generate images of objects that do not naturally exist. We used Stable Diffusion [59] to create these images. Details of prompts are in Appendix D.2.

  – Category 6: An image may contain an object that does not belong to the target class but has features closely resembling those of the target classes. Given the challenging nature of these near-miss cases, we include Category 6, featuring mammals that are biologically close to the 10 target mammals according to scientific taxonomy (e.g., donkeys are close to zebras). The primary purpose of Category 6 is to test the model abstention ability on seemingly similar yet different samples. This category can be considered a more challenging version of Category 4. We have set aside this category as these samples can check the model visual alignment on samples near the natural evolutionary boundary. Samples are collected from ImageNet21K [58].

---

[2]As category 8 contains a diverse set of croppings and corruptions of varying intensities, we double the number of samples for more reliable evaluation.

[3]https://images.cv/

[4]https://pytorch.org/

– Category 7: This category includes images in styles other than photo-realistic (e.g., a drawing of an elephant, a sculpture of a giraffe). Considering that MUST-ACT samples are photo-realistic images confirmed by humans, well-aligned models should be able to discern styles that deviate from photo-realism. The images were collected from DomainNet [50] and ImageNet-R [24].

- **UNCERTAIN** includes images that are cropped or corrupted in various styles in different intensities

  – Category 8: This category includes images that are either cropped at varying sizes and regions or corrupted using one of the 15 corruption types[5]. The original samples were collected from ImageNet21K [58]. Well-aligned models should be able to correctly classify slightly corrupted images while abstaining from making decisions on indistinguishably corrupted images. The corruption process follows the approach outlined in ImageNet-C [22], with corruption intensities varying from 1 to 10.

### 3.3 Uncertain Group Label Generation

One challengingaspect of the *Uncertain* group is the variability of these samples' gold standard labels, which fluctuates depending on corruption types and intensities. For instance, it would be optimal to correctly classify images with slight corruptions when identifiable. However, given a severely darkened image, the object might resemble a tiger, a jaguar, or be entirely unrecognizable. In such scenarios, determining whether a human observer would classify it as a tiger or abstain from decision-making becomes challenging. Therefore, we derive a gold human ratio (*i.e.*, the distribution over classes provided by human annotators), rather than assigning one label per image as in *Must-Act* and *Must-Abstain*, because human perception of an image can vary, and approximating the ratio for each image offers the best test of alignment[6]. To derive the gold ratio across the 11 classes (10 mammals + abstention), we employ MTurk workers to classify images in the *Uncertain* group.

Every MTurk worker is asked to classify 35 images, including Category 4 images corrupted with a severity between 1 to 10, with 10 being distractors. This is to minimize MTurk workers' potential biases; *e.g.*, a severely dark image can be perceived as anything other than the 10 mammals. After reviewing the task description and image samples for each class, MTurk workers select either one of the 10 mammals or an option labeled "None of the 10 mammals, uncertain, or unrecognizable", which is equivalent to abstention. To ensure the quality of samples, we disregard MTurk results where anything other than abstention was chosen for the distractor images.

In accordance with Requirement 4, we ask 134 individuals per image to estimate the indisputable ground truth distribution within an error bound of 5%, following the survey sampling theory. Proofs are provided in Appendix F. Additionally, we calculate the Fleiss' Kappa [15] to assess two types of consistency among the MTurk workers' answers: intra-annotator and inter-annotator consistency. Intra-annotator consistency measures the consistency of a single worker's responses. To calculate this, we inserted two sets of identical images in random order. If a worker selects the same answers for these identical images, we consider the worker's responses to be consistent. Inter-annotator consistency, on the other hand, measures the agreement among different workers. Our results show an intra-annotator consistency value of $\kappa = 0.91$, indicating almost perfect agreement, and an inter-annotator consistency value of $\kappa = 0.80$, demonstrating substantial agreement. Details on survey instructions, response filtering process, and participant statistics are provided in Appendix F.

### 3.4 Dataset

We prepare three datasets: the train set, the open test set, and the closed test set. The train set is a subset of ImageNet-21K [58], consisting only of Category 1 samples. By doing so, we ensure the trained models are tested on a variety of unseen categories, reflecting a real-world scenario. For each of our 10 classes, we randomly sample a uniform amount of images from all related ImageNet-21K classes. We collected a total of 1250 images per class, using one-tenth of this data for validation. The creation processes of both the open and closed test sets are identical, as described above. We provide the open test set to allow developers to evaluate their models' visual perception alignment. Developers

---

[5]We leveraged open-sourced code available at `https://github.com/hendrycks/robustness`

[6]Some might wonder why the machines should settle for aligning with human visual perception, rather than aiming to correctly classify even the most corrupted images (*i.e.* aim for superhuman visual perception). We provide arguments for the necessity of the former in Appendix D.2.2.

Table 1: The comparison between VisAlign and other related datasets on the requirements we define. △ indicates that only a subset of our scenarios are covered.

| Dataset | Req. 1 | Req. 2 | Req. 3 | Req. 4 |
|---|---|---|---|---|
| ImageNet-C [22] | ✗ | ✗ | △ | ✓ |
| ImageNet-A [25] | ✗ | ✗ | ✗ | ✓ |
| OpenOOD [74] | ✗ | ✗ | △ | ✓ |
| Background Challenge [73] | ✗ | ✗ | ✗ | ✓ |
| MNIST [34] | ✗ | ✓ | ✗ | ✓ |
| CIFAR10 [30] | ✗ | ✓ | ✗ | ✓ |
| CIFAR10H [52] | ✗ | ✓ | △ | ✓ |
| PLEX [69] | ✗ | ✗ | ✓ | ✓ |
| Park et al. [49] | ✓ | ✗ | △ | ✗ |
| DCIC [65] | ✗ | ✗ | △ | ✓ |
| VisAlign | ✓ | ✓ | ✓ | ✓ |

wishing to evaluate their models on the closed test set can submit their models to us. Table 1 presents a comparison of VisAlign and other datasets in terms of fulfilling the four requirements.

# 4 Metrics

We introduce a distance-based metric to measure *AI-human visual alignment*. Furthermore, we present a reliability score table to explore the correlation between a model's visual perception alignment and model reliability.

## 4.1 Distance-Based Visual Perception Similarity Metric

We propose a distance-based metric to measure the distance between two multinomial distributions: the human visual distribution and the model output distribution over 11 classes (10 mammals + abstention). We opt for a distance-based metric for two reasons: 1) it does not depend on additional hyperparameters such as abstention threshold, and 2) comparison across all classes, rather than solely on the true class, provides a more accurate measure of visual alignment. For example, consider a *Must-Act* tiger sample with the gold human label as a one-hot vector for the label *tiger*. Suppose one model outputs a probability of 0.7 for *tiger* and 0.3 for *abstention*, and another model yields a probability of 0.7 for *tiger* and 0.1 for *zebra*, *elephant*, and *giraffe* respectively. These two models differ in visual perception alignment: the former is uncertain between two classes, whereas the latter is indecisive among four classes. If we were to consider only the gold label's probability, both models would yield the same result, which would not accurately represent visual alignment. Hence, we employ a distance-based metric calculated across all 11 classes, as opposed to using the maximum or gold label probability.

Specifically, we employ the Hellinger distance [47] to measure the difference between the two probability distributions as summarized in Eq. 1. Compared to other metrics for comparing two multinomial distributions, Hellinger distance produces smooth distance values even for extreme (*e.g.*, one-hot) distributions (unlike KL Divergence [10]) and considers all classes while calculating the distance (unlike Total Variation distance). For instance, given a human visual distribution of [1., 0., 0.] and two model output distributions [0.3, 0., 0.7] and [0.3, 0.4, 0.3], the two output distributions would have the same KL Divergences with the human distribution while they have different Hellinger distances. Hellinger distance accounts not only for the gold label probability but also for the probabilities of all other labels. Additionally, as its range lies between 0 and 1, it provides an intuitive indication of model alignment.

$$h(P, Q) = \frac{1}{\sqrt{2}} \sum_{i} \|\sqrt{p_i} - \sqrt{q_i}\|_2 \qquad (1)$$

## 4.2 Reliability Score with Abstention

We also assess the model's reliability based on its final action. This process involves two steps. First, a model abstains if the abstention probability surpasses an abstention threshold, $\gamma$; otherwise, it makes a prediction. Next, if a model decides to act, its prediction is one of the 10 mammal classes

with the highest prediction probability. Table 2 details the reliability scores for each case. We devise separate metrics for *Must-Act* and *Must-Abstain* instances. For *Uncertain* samples, they are treated as *Must-Act* if the probability of the original label exceeds a threshold $\lambda$; otherwise, they are treated as *Must-Abstain*. We set an initial $\lambda$ value at 0.5, but this can be adjusted according to the specific objective. We denote the reliability score as $RS_c(x)$, where $c$ is the cost of an incorrect prediction. The main criterion for assigning scores is the consequences of the model's decision. The model earns a score of 1 per prediction when it aligns best with human recognition: making a correct prediction in Must-Act and abstaining in Must-Abstain. On the other hand, if the model's decision is erroneous and could potentially result in significant cost—in our case, a wrong prediction—the model receives a score of $-c$. A score of zero indicates that the prediction is neither beneficial nor detrimental. Original Label Prediction is a special case only applied for Uncertain samples treated as Must-Abstain. In this case, a model correctly classifies a corrupted image that most humans cannot recognize. Although most humans disagree with the model's decision, it does not have a negative impact since it is a correct answer. The total score, $RS_c$, is the summation over all test samples, $\sum_i RS_c(x_i)$.

The proper value of cost $c$ depends on the industry and the use case. $c$ can be seen as the "strictness criterion for a reliable model" and can also be interpreted as "how many misclassifications correspond to a single accurate classification." $c$ can be set as an integer ranging from 0 to the total size of the test set. A value 0 for $c$ implies a 0% strictness, while the maximum value of $c$ implies a 100% strictness. This means that even a single mistake would result in a negative score, and abstaining from all decisions on Must-Act samples would be deemed more reliable than making even one incorrect prediction. We designed this metric to enable both absolute and relative reference points. As an absolute reference point, if the final score is at or above 0 (non-negative reliability score), it demonstrates

Table 2: Reliability score table. The optimal outcomes earn a score of 1. Abstention in *Must-Predict* and Original Label Prediction in *Must-Abstain* get 0. The worst case receives $-c$, where $c$ is the cost value. *Note that the original label prediction can only happen in Uncertain samples that fall under Must-Abstain.

| Sample Type | Model Action | $RS_c(x)$ |
|---|---|---|
| | Correct Prediction | $+1$ |
| Must-Act | Incorrect Prediction | $-c$ |
| | Abstention | 0 |
| | Original Label Prediction* | 0 |
| Must-Abstain | Other Prediction | $-c$ |
| | Abstention | $+1$ |

that the model satisfies the user-defined minimum reliability. A relative reference point is between different models; a model with a higher score between two reliability scores is more reliable. In this paper, we set the value of $c$ as 0, 450, or 900.

# 5 Experiment

## 5.1 Experiment Settings

We perform experiments with Transformer-based [71], CNN-based [33], and MLP-based models. We use ViT [11] and Swin Transformer [38] for Transformer-based models, and DenseNet [27] and ConvNeXt [39] for CNN-based models. For the MLP-based model, we use MLP-Mixer [68]. All models are trained on our train set and tested on the open test set.

We chose abstention functions that satisfy the following three conditions: 1) must be applicable on any model architecture, 2) do not require OOD or other Must-Abstain samples during training, and 3) do not require a supplementary model. We first calculate the abstention probability using each function, then re-normalize the 10-class prediction probability so that the sum over the 11 classes becomes 1. Since not every function outputs the abstention probability between 0 and 1, we designed a smaller version of the dataset with the identical gather process to test set to use for normalizing the abstention probability.

- Softmax Probability (SP) regards the entropy among the 10 classes as abstention probability.
- Adjusted Softmax Probability (ASP) acts the same as SP, but it applies temperature scaling and adds perturbations to the input image based on the gradients to decrease the softmax score. This method is inspired by ODIN [26].
- Mahalanobis detector (MD) [35] determines abstention probability based on the minimum Mahalanobis distance [40] calculated from each class distribution's mean and variance.

Table 3: Average and standard deviation of the distance-based visual alignment and reliability scores across five seeds on the open test set. Bold indicates the best performance in each category, and underline is the second best. Deep Ensemble does not have a standard deviation since it uses the output of all five models.

| | Visual Alignment (↓) | | | | | | | | | Reliability score (↑) | | |
| | Must-Act | | | Must-Abstain | | | | Uncertain | Average | $RS_0$ | $RS_{450}$ | $RS_{900}$ |
| | Category 1 | Category 2 | Category 3 | Category 4 | Category 5 | Category 6 | Category 7 | Category 8 | | | | |
| **ViT [11]** | | | | | | | | | | | | |
| SP | $0.261_{\pm0.051}$ | $0.556_{\pm0.029}$ | $0.367_{\pm0.038}$ | $0.793_{\pm0.057}$ | $0.808_{\pm0.057}$ | $0.787_{\pm0.056}$ | $0.792_{\pm0.059}$ | $0.671_{\pm0.032}$ | $0.629_{\pm0.021}$ | 313 | −245837 | −491987 |
| ASP | $0.208_{\pm0.036}$ | $0.514_{\pm0.033}$ | $0.325_{\pm0.022}$ | $1.000_{\pm0.000}$ | $1.000_{\pm0.000}$ | $1.000_{\pm0.000}$ | $1.000_{\pm0.000}$ | $0.767_{\pm0.010}$ | $0.727_{\pm0.007}$ | 253 | −285047 | −570347 |
| MD [35] | $0.390_{\pm0.030}$ | $0.658_{\pm0.025}$ | $0.485_{\pm0.023}$ | $0.725_{\pm0.021}$ | $0.721_{\pm0.023}$ | $0.726_{\pm0.023}$ | $0.664_{\pm0.025}$ | $0.623_{\pm0.012}$ | $0.624_{\pm0.005}$ | 270 | −275580 | −551430 |
| KNN [67] | $0.382_{\pm0.047}$ | $0.634_{\pm0.029}$ | $0.484_{\pm0.033}$ | $0.679_{\pm0.058}$ | $0.696_{\pm0.050}$ | $0.679_{\pm0.049}$ | $0.674_{\pm0.067}$ | $0.612_{\pm0.034}$ | $0.605_{\pm0.020}$ | 282 | −264768 | −529818 |
| TAPUDD [13] | $0.375_{\pm0.070}$ | $0.628_{\pm0.073}$ | $0.468_{\pm0.074}$ | $0.809_{\pm0.079}$ | $0.809_{\pm0.084}$ | $0.835_{\pm0.065}$ | $0.768_{\pm0.089}$ | $0.678_{\pm0.024}$ | $0.671_{\pm0.017}$ | 253 | −285047 | −570347 |
| OpenMax [3] | $0.238_{\pm0.027}$ | $0.536_{\pm0.033}$ | $0.344_{\pm0.022}$ | $0.804_{\pm0.050}$ | $0.816_{\pm0.037}$ | $0.804_{\pm0.059}$ | $0.766_{\pm0.055}$ | $0.696_{\pm0.025}$ | $0.626_{\pm0.020}$ | 335 | −229165 | −458665 |
| MC-Dropout [16] | $0.210_{\pm0.036}$ | $0.516_{\pm0.032}$ | $0.326_{\pm0.022}$ | $0.968_{\pm0.009}$ | $0.970_{\pm0.010}$ | $0.968_{\pm0.009}$ | $0.968_{\pm0.010}$ | $0.749_{\pm0.014}$ | $0.709_{\pm0.005}$ | 253 | −285047 | −570347 |
| Deep Ensemble [32] | 0.305 | 0.571 | 0.400 | 0.712 | 0.732 | 0.705 | 0.713 | 0.628 | 0.596 | 376 | −205274 | −410924 |
| **Swin Transformer [38]** | | | | | | | | | | | | |
| SP | $0.106_{\pm0.004}$ | $0.362_{\pm0.014}$ | $0.221_{\pm0.017}$ | $0.793_{\pm0.016}$ | $0.828_{\pm0.043}$ | $0.800_{\pm0.022}$ | $0.829_{\pm0.028}$ | $0.625_{\pm0.031}$ | $0.571_{\pm0.015}$ | 363 | −225537 | −451437 |
| ASP | $0.085_{\pm0.007}$ | $0.329_{\pm0.008}$ | $0.182_{\pm0.020}$ | $0.998_{\pm0.000}$ | $0.998_{\pm0.000}$ | $0.998_{\pm0.000}$ | $0.998_{\pm0.000}$ | $0.736_{\pm0.009}$ | $0.666_{\pm0.003}$ | 294 | −268356 | −537006 |
| MD [35] | $0.296_{\pm0.018}$ | $0.512_{\pm0.012}$ | $0.364_{\pm0.012}$ | $0.700_{\pm0.012}$ | $0.743_{\pm0.014}$ | $0.723_{\pm0.017}$ | $0.685_{\pm0.021}$ | $0.575_{\pm0.007}$ | $0.575_{\pm0.006}$ | 326 | −248974 | −498274 |
| KNN [67] | $0.370_{\pm0.017}$ | $0.580_{\pm0.008}$ | $0.456_{\pm0.018}$ | $\mathbf{0.549_{\pm0.025}}$ | $\mathbf{0.590_{\pm0.013}}$ | $\mathbf{0.545_{\pm0.022}}$ | $\mathbf{0.554_{\pm0.021}}$ | $\mathbf{0.543_{\pm0.007}}$ | $\mathbf{0.523_{\pm0.012}}$ | **526** | **-115124** | **-230774** |
| TAPUDD [13] | $0.201_{\pm0.053}$ | $0.427_{\pm0.048}$ | $0.278_{\pm0.046}$ | $0.876_{\pm0.058}$ | $0.889_{\pm0.048}$ | $0.898_{\pm0.049}$ | $0.844_{\pm0.071}$ | $0.663_{\pm0.022}$ | $0.635_{\pm0.013}$ | 294 | −268356 | −537006 |
| OpenMax [3] | $0.099_{\pm0.008}$ | $0.358_{\pm0.013}$ | $0.225_{\pm0.029}$ | $0.831_{\pm0.037}$ | $0.810_{\pm0.023}$ | $0.817_{\pm0.032}$ | $0.724_{\pm0.084}$ | $0.656_{\pm0.030}$ | $0.565_{\pm0.011}$ | 399 | −208401 | −417201 |
| MC-Dropout [16] | $0.092_{\pm0.007}$ | $0.338_{\pm0.008}$ | $0.191_{\pm0.020}$ | $0.947_{\pm0.001}$ | $0.957_{\pm0.006}$ | $0.951_{\pm0.002}$ | $0.953_{\pm0.003}$ | $0.705_{\pm0.011}$ | $0.642_{\pm0.003}$ | 294 | −268356 | −537006 |
| Deep Ensemble [32] | 0.132 | 0.377 | 0.253 | 0.725 | 0.766 | 0.734 | 0.768 | 0.584 | 0.542 | 434 | −187666 | −375766 |
| **DenseNet [27]** | | | | | | | | | | | | |
| SP | $0.094_{\pm0.017}$ | $0.258_{\pm0.023}$ | $0.183_{\pm0.019}$ | $0.813_{\pm0.017}$ | $0.852_{\pm0.015}$ | $0.819_{\pm0.012}$ | $0.864_{\pm0.036}$ | $0.614_{\pm0.008}$ | $0.562_{\pm0.007}$ | 392 | −211558 | −423508 |
| ASP | $\mathbf{0.079_{\pm0.013}}$ | $\mathbf{0.224_{\pm0.023}}$ | $\mathbf{0.159_{\pm0.018}}$ | $0.997_{\pm0.000}$ | $0.997_{\pm0.000}$ | $0.997_{\pm0.000}$ | $0.997_{\pm0.000}$ | $0.747_{\pm0.008}$ | $0.650_{\pm0.004}$ | 312 | −260238 | −520788 |
| MD [35] | $0.170_{\pm0.016}$ | $0.323_{\pm0.022}$ | $0.250_{\pm0.025}$ | $0.873_{\pm0.014}$ | $0.866_{\pm0.019}$ | $0.854_{\pm0.009}$ | $0.825_{\pm0.032}$ | $0.620_{\pm0.022}$ | $0.598_{\pm0.006}$ | 339 | −247611 | −495561 |
| KNN [67] | $0.272_{\pm0.021}$ | $0.448_{\pm0.021}$ | $0.360_{\pm0.017}$ | $0.612_{\pm0.019}$ | $0.640_{\pm0.022}$ | $0.615_{\pm0.019}$ | $0.664_{\pm0.014}$ | $0.565_{\pm0.004}$ | $0.522_{\pm0.009}$ | 482 | −157468 | −315418 |
| TAPUDD [13] | $0.310_{\pm0.039}$ | $0.393_{\pm0.025}$ | $0.364_{\pm0.044}$ | $0.862_{\pm0.021}$ | $0.831_{\pm0.023}$ | $0.837_{\pm0.018}$ | $0.810_{\pm0.028}$ | $0.645_{\pm0.017}$ | $0.631_{\pm0.004}$ | 320 | −249880 | −500080 |
| OpenMax [3] | $0.093_{\pm0.010}$ | $0.288_{\pm0.023}$ | $0.199_{\pm0.027}$ | $0.764_{\pm0.049}$ | $0.817_{\pm0.054}$ | $0.734_{\pm0.058}$ | $0.823_{\pm0.058}$ | $0.590_{\pm0.016}$ | $0.539_{\pm0.025}$ | 461 | −165589 | −331639 |
| MC-Dropout [16] | $0.087_{\pm0.014}$ | $0.263_{\pm0.024}$ | $0.204_{\pm0.017}$ | $0.953_{\pm0.003}$ | $0.953_{\pm0.002}$ | $0.954_{\pm0.005}$ | $0.964_{\pm0.004}$ | $0.718_{\pm0.009}$ | $0.637_{\pm0.003}$ | 312 | −260238 | −520788 |
| Deep Ensemble [32] | 0.109 | 0.276 | 0.209 | 0.767 | 0.814 | 0.775 | 0.825 | 0.581 | 0.545 | 396 | −209754 | −419904 |
| **ConvNeXt [39]** | | | | | | | | | | | | |
| SP | $0.211_{\pm0.020}$ | $0.461_{\pm0.032}$ | $0.354_{\pm0.024}$ | $0.661_{\pm0.039}$ | $0.772_{\pm0.023}$ | $0.671_{\pm0.026}$ | $0.767_{\pm0.028}$ | $0.583_{\pm0.016}$ | $0.560_{\pm0.008}$ | 427 | −180923 | −362273 |
| ASP | $0.162_{\pm0.015}$ | $0.398_{\pm0.026}$ | $0.299_{\pm0.021}$ | $0.998_{\pm0.000}$ | $0.999_{\pm0.000}$ | $0.998_{\pm0.000}$ | $0.998_{\pm0.000}$ | $0.729_{\pm0.003}$ | $0.698_{\pm0.007}$ | 283 | −268817 | −537917 |
| MD [35] | $0.439_{\pm0.019}$ | $0.583_{\pm0.023}$ | $0.504_{\pm0.017}$ | $0.699_{\pm0.021}$ | $0.663_{\pm0.023}$ | $0.692_{\pm0.031}$ | $0.642_{\pm0.069}$ | $0.600_{\pm0.009}$ | $0.603_{\pm0.013}$ | 350 | −213400 | −427150 |
| KNN [67] | $0.376_{\pm0.007}$ | $0.574_{\pm0.014}$ | $0.484_{\pm0.009}$ | $0.613_{\pm0.027}$ | $0.654_{\pm0.013}$ | $0.627_{\pm0.024}$ | $0.621_{\pm0.019}$ | $0.565_{\pm0.004}$ | $0.564_{\pm0.010}$ | 451 | −169649 | −339749 |
| TAPUDD [13] | $0.448_{\pm0.018}$ | $0.578_{\pm0.011}$ | $0.518_{\pm0.019}$ | $0.796_{\pm0.016}$ | $0.732_{\pm0.016}$ | $0.799_{\pm0.014}$ | $0.752_{\pm0.035}$ | $0.656_{\pm0.007}$ | $0.660_{\pm0.010}$ | 278 | −263422 | −527122 |
| OpenMax [3] | $0.183_{\pm0.010}$ | $0.408_{\pm0.026}$ | $0.318_{\pm0.027}$ | $0.944_{\pm0.007}$ | $0.914_{\pm0.012}$ | $0.960_{\pm0.005}$ | $0.888_{\pm0.052}$ | $0.708_{\pm0.004}$ | $0.665_{\pm0.010}$ | 286 | −261614 | −523514 |
| MC-Dropout [16] | $0.166_{\pm0.016}$ | $0.403_{\pm0.026}$ | $0.303_{\pm0.021}$ | $0.941_{\pm0.003}$ | $0.958_{\pm0.002}$ | $0.942_{\pm0.001}$ | $0.955_{\pm0.003}$ | $0.699_{\pm0.002}$ | $0.671_{\pm0.010}$ | 283 | −268817 | −537917 |
| Deep Ensemble [32] | 0.228 | 0.480 | 0.368 | 0.621 | 0.743 | 0.633 | 0.738 | 0.563 | 0.547 | 455 | −166045 | −332545 |
| **MLP-Mixer [68]** | | | | | | | | | | | | |
| SP | $0.240_{\pm0.048}$ | $0.556_{\pm0.055}$ | $0.333_{\pm0.032}$ | $0.842_{\pm0.051}$ | $0.829_{\pm0.059}$ | $0.828_{\pm0.034}$ | $0.850_{\pm0.037}$ | $0.674_{\pm0.024}$ | $0.644_{\pm0.021}$ | 347 | −231403 | −463153 |
| ASP | $0.212_{\pm0.041}$ | $0.524_{\pm0.054}$ | $0.303_{\pm0.038}$ | $0.999_{\pm0.000}$ | $0.999_{\pm0.000}$ | $0.999_{\pm0.000}$ | $0.999_{\pm0.000}$ | $0.749_{\pm0.017}$ | $0.723_{\pm0.013}$ | 260 | −285490 | −571240 |
| MD [35] | $0.431_{\pm0.044}$ | $0.682_{\pm0.042}$ | $0.499_{\pm0.017}$ | $0.649_{\pm0.021}$ | $0.651_{\pm0.022}$ | $0.640_{\pm0.012}$ | $0.621_{\pm0.034}$ | $0.609_{\pm0.013}$ | $0.598_{\pm0.009}$ | 374 | −209326 | −419026 |
| KNN [67] | $0.414_{\pm0.034}$ | $0.680_{\pm0.018}$ | $0.491_{\pm0.017}$ | $0.621_{\pm0.013}$ | $0.641_{\pm0.013}$ | $0.610_{\pm0.014}$ | $0.634_{\pm0.013}$ | $0.584_{\pm0.015}$ | $0.584_{\pm0.009}$ | 402 | −194448 | −389298 |
| TAPUDD [13] | $0.586_{\pm0.013}$ | $0.742_{\pm0.020}$ | $0.631_{\pm0.021}$ | $0.624_{\pm0.023}$ | $0.600_{\pm0.021}$ | $0.620_{\pm0.022}$ | $0.603_{\pm0.024}$ | $0.619_{\pm0.014}$ | $0.628_{\pm0.012}$ | 297 | −231003 | −462303 |
| OpenMax [3] | $0.290_{\pm0.045}$ | $0.660_{\pm0.060}$ | $0.372_{\pm0.065}$ | $0.662_{\pm0.127}$ | $0.681_{\pm0.014}$ | $0.648_{\pm0.174}$ | $0.630_{\pm0.122}$ | $0.631_{\pm0.050}$ | $0.568_{\pm0.057}$ | 358 | −218792 | −437942 |
| MC-Dropout [16] | $0.213_{\pm0.041}$ | $0.525_{\pm0.054}$ | $0.303_{\pm0.037}$ | $0.977_{\pm0.008}$ | $0.974_{\pm0.009}$ | $0.975_{\pm0.005}$ | $0.977_{\pm0.006}$ | $0.737_{\pm0.017}$ | $0.710_{\pm0.013}$ | 260 | −285490 | −571240 |
| Deep Ensemble [32] | 0.297 | 0.597 | 0.370 | 0.735 | 0.714 | 0.719 | 0.743 | 0.614 | 0.599 | 376 | −207524 | −415424 |

- KNN [67] uses the shortest $k$-Nearest Neighbor (KNN) distance between the feature of the test sample and the in-class features as an abstention probability.
- TAPUDD [13] extracts features from train set and split into $m$ clusters using Gaussian Mixture Model (GMM). It determines the abstention probability based on the shortest Mahalanobis distance calculated from all clusters.
- OpenMax [3] represents each class as a mean activation vector (MAV) in the penultimate layer of the network. Next, the test sample distance from the corresponding class MAV is used to calculate the abstention probabililty.
- MC-Dropout [16] and Deep Ensemble [32] approximate model uncertainty using multiple predictions given by different dropouts and ensemble of networks, respectively. The average of the entropies over the 10 classes of each prediction determines the abstention probability.

## 5.2 Visual Alignment and Reliability Score

Table 3 presents both the distance-based visual alignment and the reliability scores on the open test set for all model and abstention function combinations. One key observation is that the performance differences between models are not significantly distinct, suggesting that visual alignment is more influenced by abstention functions than by the model architectures. For *Must-Act* categories, distance-based functions (MD, KNN, and TAPUDD) exhibit better visual alignment. Conversely, for *Must-Abstain* samples, probability-based methods (SP and ASP) align better with human perception. This implies that distance-based abstentions are generally more inclined to act, while probability-based abstentions are more likely to abstain. In *Uncertain* category, all abstention functions demonstrate

similar visual alignment performances, predominantly ranging from 0.5 and 0.6. We conjecture the reason comes from that all models are struggling in approximating the overall ratios across 11 classes compared to *Must-Act* and *Must-Abstain*, where models only need to correctly predict a single class. The difficulty of achieving visual perception alignment in *Uncertain* suggests that there is room for improvement. KNN [67] has the best visual alignment across all categories on average. This might be because KNN can capture more fine-grained features than other distance-based abstention functions, as it calculates the distance between samples, not clusters. We also compute three reliability scores with $c$ set to 0 ($RS_0$), 450 ($RS_{450}$), and 900 ($RS_{900}$). The resulting ratios of each action type are shown in Appendix G.1. Here, $c = 0$ indicates no negative impacts from incorrect predictions, while $c = 900$ suggests that a single incorrect prediction outweighs the remaining correct predictions. It is worth noting that reliability scores in $RS_{450}$ and $RS_{900}$ are mostly negative, suggesting that current models and abstention functions are not perfectly safe to be deployed in the real world. Notably, visual alignment distance is correlated with reliability score as can be seen in Appendix G.2.

Methods based on the minimum distance from each class (MD, KNN, and TAPUDD) generally show a worse visual alignment on *Must-Abstain* categories. We conjecture that the reason comes from using the shortest distance to in-class clusters. If an embedding contains one clear in-class feature, the distance to the corresponding class would be short, leading the model to make a prediction. On the other hand, methods based on entropy or uncertainty show weak alignment on *Must-Act* categories. With these methods, the model has to be not only confident that its predicted class is correct but also that the remaining classes are incorrect. Considering the confidence in all classes makes it more challenging for visual alignment in *Must-Act* categories. An abstention function which takes advantage of both distance-based and probability-based methods is needed to perform well on visual alignment. The distance should be sample-wise to capture the nuanced characteristics of the samples. Overall, our experiments show that no methods perform well across all categories. There is much room for improvement in visual alignment, a field in which our dataset will become an essential tool for benchmarking new methods.

## 6 Conclusion

To the best of our knowledge, this is the first work to construct a test benchmark for quantitatively measuring the visual perception alignment between models and humans, referred to as *AI-human visual alignment*, across diverse scenarios. Our dataset is divided into three main groups and eight categories, each representing unique and essential situations. Our dataset includes gold human labels for each image, with some of these labels collected via MTurk survey. We benchmarked five baseline models and seven popular abstention functions, and our experimental results show that no current methods perform well across all categories. This suggests there is room for improvement in visual alignment. We believe VisAlign can serve as a universal benchmark for testing visual perception alignment and that our work has potential applications in both social and industrial contexts.

Despite our best efforts to construct VisAlign, there are some limitations. First, the number of classes is relatively small compared to other datasets since we collected 134 annotations per image and chose classes that would be familiar to an average human. Note that it is always challenging to collect gold human labels in any domain. For example, in diagnosing chest X-rays, the typical number of diseases is 14. To collect the ground truth labels within a statistical error bound of 5%, one would need to consult at least 107 radiologists. Therefore, more practical solutions are required to measure alignment in specialized domains. Another limitation comes from the nature of uncertainty. We acknowledge that uncertainty is continuous and it is hard to distinguish between clear and uncertain images. Although we put significant effort to include only clear images in Must-Act and Must-Abstain and obtained human annotations on Uncertain images, there is a possibility of corner cases where at least one person disagrees. Furthermore, synthetic corruptions cannot cover all uncertainties arising in the real world. However, uncertainty is too broad to specify and difficult to collect or generate, thus for now we use corruptions. We put our best effort to reflect the continuity of uncertainty by varying corruption intensity from 1 to 10 and include some corruptions that can arise in the real world (*e.g.*, pixelation). We detailed further discussions on uncertainty in Appendix I. Also, extending visual alignment to scenarios such as visual illusions may also be introduced. While our dataset focuses on the essential object identification and abstention task under *AI-human visual alignment*, future work can be expanded to potentially contentious but socially engaging topics such as gender or racial bias and other vision tasks such as object detection and segmentation.

## Acknowledgments and Disclosure of Funding

This work was supported by Institute of Information & communications Technology Planning & Evaluation (IITP) grant (No.2019-0-00075, No.2022-0-00984) and National Research Foundation of Korea (NRF) grant (NRF-2020H1D3A2A03100945, NRF-2021H1D3A2A03038607), funded by the Korea government (MSIT). Marzyeh Ghassemi receives support from the Herman L. F. von Helmholtz Career Development Professorship and CIFAR Azrieli Global Scholar award.

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

# Appendix

## A  Datasheet for Datasets

The following section is answers to questions listed in datasheets for datasets.

### A.1  Motivation

- For what purpose was the dataset created?
  VisAlign is created to serve as a benchmark for measuring visual perception alignment between AI models and humans.

- Who created the dataset (e.g., which team, research group) and on behalf of which entity (e.g., company, institution, organization)?
  The authors of this paper.

- Who funded the creation of the dataset? If there is an associated grant, please provide the name of the grantor and the grant name and number.
  This work was supported by Institute of Information & Communications Technology Planning & Evaluation (IITP) grant (No.2019-0-00075, Artificial Intelligence Graduate School Program(KAIST)) and National Research Foundation of Korea (NRF) grant (NRF-2020H1D3A2A03100945), funded by the Korea government (MSIT).

### A.2  Composition

- What do the instances that comprise the dataset represent (e.g., documents, photos, people, countries)?
  VisAlign contains eight different types of images and their corresponding gold human labels.

- How many instances are there in total (of each type, if appropriate)?
  There are a total of 12500 images in the train set, distributed equally among the 10 classes. The open test set and the closed test each contain 900 images: 100 images each in Categories 1 to 7 and 200 images in Category 8.

- Does the dataset contain all possible instances or is it a sample (not necessarily random) of instances from a larger set?
  The train set is a sample of instances of ImageNet-21K, where images have been randomly sampled from synsets and corresponding hyponyms related to each of our classes. The test sets are samples carefully selected by the authors without replacement to match each of the categories' requirements.

- What data does each instance consist of?
  Each instance consists of an image and its corresponding gold human label.

- Is there a label or target associated with each instance?
  Yes, the label represents the gold label (*e.g.*, human visual perception).

- Is any information missing from individual instances? If so, please provide a description, explaining why this information is missing (e.g., because it was unavailable). This does not include intentionally removed information, but might include, e.g., redacted text.
  N/A.

- Are relationships between individual instances made explicit (e.g., users' movie ratings, social network links)?
  N/A.

- Are there recommended data splits (e.g., training, development/validation, testing)?
  No, since VisAlign is an universal benchmark that any model can be tested on regardless of its train set, a developer may feel free to use any training strategies.

- Are there any errors, sources of noise, or redundancies in the dataset?
  N/A.

- Is the dataset self-contained, or does it link to or otherwise rely on external resources (e.g., websites, tweets, other datasets)?

The dataset relies on open source databases: ImageNet [60], ImageNet21K [58], ImageNet-C [22], DomainNet [50], and ImageNet-R [24].

- Does the dataset contain data that might be considered confidential (e.g., data that is protected by legal privilege or by doctor– patient confidentiality, data that includes the content of individuals' non-public communications)?
  N/A.

- Does the dataset contain data that, if viewed directly, might be offensive, insulting, threatening, or might otherwise cause anxiety?
  N/A.

- Does the dataset relate to people?
  Yes.

- Does the dataset identify any subpopulations (e.g., by age, gender)?
  N/A.

- Is it possible to identify individuals (i.e., one or more natural persons), either directly or indirectly (i.e., in combination with other data) from the dataset?
  N/A.

- Does the dataset contain data that might be considered sensitive in any way (e.g., data that reveals race or ethnic origins, sexual orientations, religious beliefs, political opinions or union memberships, or locations; financial or health data; biometric or genetic data; forms of government identification, such as social security numbers; criminal history)?
  N/A.

### A.3 Collection Process

- How was the data associated with each instance acquired?
  We leveraged open source datasets. For Category 2 and Category 5, we synthesized images using Stable Diffusion [59]. For Category 3, we manually applied FGSM [20] on samples in Category 1. For Category8, we applied corruptions on Category 1 samples by using corruption code available in ImageNet-C [22].

- What mechanisms or procedures were used to collect the data (e.g., hardware apparatuses or sensors, manual human curation, software programs, software APIs)?
  We used the website Amazon Mechanical Turk (MTurk) to create gold human labels for *Uncertain*. After the poll, we used Excel, Google Sheets, and Python to process and label the collected data.

- If the dataset is a sample from a larger set, what was the sampling strategy (e.g., deterministic, probabilistic with specific sampling probabilities)?
  We first removed images that are hard to recognize or have more than two different objects. After the curating, when it involves sampling, we sampled with a fixed random seed.

- Who was involved in the data collection process (e.g., students, crowdworkers, contractors) and how were they compensated (e.g., how much were crowdworkers paid)?
  There were one part that required human involvement in the data collection process, deriving gold human label ratio for *Uncertain*. We provided $ 0.05 for classifying 25 images. We did not put any restrictions on participants.

- Over what timeframe was the data collected?
  The poll was conducted in March of 2023, but the results do not depend much on the date of date collection.

- Were any ethical review processes conducted (e.g., by an institutional review board)?
  N/A.

- Does the dataset relate to people?
  Yes.

- Did you collect the data from the individuals in question directly, or obtain it via third parties or other sources (e.g., websites)?
  We obtained via Amazon Mechanical Turk MTurk website.

- Were the individuals in question notified about the data collection?
  Yes.

- Did the individuals in question consent to the collection and use of their data?
  Yes.

- If consent was obtained, were the consenting individuals provided with a mechanism to revoke their consent in the future or for certain uses?
  N/A.

- Has an analysis of the potential impact of the dataset and its use on data subjects (e.g., a data protection impact analysis) been conducted?
  The dataset does not have individual-specific information.

## A.4 Preprocessing/cleaning/labeling

- Was any preprocessing/cleaning/labeling of the data done (e.g., discretization or bucketing, tokenization, part-of-speech tagging, SIFT feature extraction, removal of instances, processing of missing values)?
  For the data quality, we removed inappropriate responses (that fall under the distractors).

- Was the "raw" data saved in addition to the preprocessed/cleaned/labeled data (e.g., to support unanticipated future uses)?
  N/A.

- Is the software that was used to preprocess/clean/label the data available?
  Preprocessing, cleaning, and labeling are done via Excel, Google Sheets, and Python.

## A.5 Uses

- Has the dataset been used for any tasks already?
  No.

- Is there a repository that links to any or all papers or systems that use the dataset?
  No.

- What (other) tasks could the dataset be used for?
  N/A.

- Is there anything about the composition of the dataset or the way it was collected and preprocessed/cleaned/labeled that might impact future uses?
  N/A.

- Are there tasks for which the dataset should not be used?
  N/A.

## A.6 Distribution

- Will the dataset be distributed to third parties outside of the entity (e.g., company, institution, organization) on behalf of which the dataset was created?
  No.

- How will the dataset will be distributed (e.g., tarball on website, API, GitHub)?
  The dataset will be released upon acceptance.

- When will the dataset be distributed?
  After the whole process of reviewing.

- Will the dataset be distributed under a copyright or other intellectual property (IP) license, and/or under applicable terms of use (ToU)?
  The dataset will be released under MIT License.

- Have any third parties imposed IP-based or other restrictions on the data associated with the instances?
  No.

- Do any export controls or other regulatory restrictions apply to the dataset or to individual instances?
  No.

### A.7 Maintenance

- Who will be supporting/hosting/maintaining the dataset?
  The authors of this paper.

- How can the owner/curator/manager of the dataset be contacted (e.g., email address)?
  Contact the first author (`jiyounglee0523@kaist.ac.kr`) or other authors.

- Is there an erratum?
  No.

- Will the dataset be updated (e.g., to correct labeling errors, add new instances, delete instances)?
  If any correction is needed, we plan to upload a new version.

- If the dataset relates to people, are there applicable limits on the retention of the data associated with the instances (e.g., were the individuals in question told that their data would be retained for a fixed period of time and then deleted)?
  N/A

- Will older versions of the dataset continue to be supported/hosted/maintained?
  We plan to maintain the newest version only.

- If others want to extend/augment/build on/contribute to the dataset, is there a mechanism for them to do so?
  Contact the authors of the paper.

## B  Training Details

For the experiments in Section 5, we use a batch size of 16 with a learning rate starting at $1 \times 10^{-5}$. The learning rate is decreased by a factor of 0.5 if there is no improvement for 10 epochs or until it reaches $1 \times 10^{-6}$. We approximately match the size of each model to 300M parameters. For ViT, we use the variant with 30 layers and 16 heads in each layer. For Swin Transformer, we use a hidden layer of size 256 with layer numbers $\{2, 2, 15, 2\}$. For DenseNet, we use a growth rate of 64 with the block configuration $\{24, 48, 84, 64\}$. For ConvNeXt, we use the large variant with block numbers $\{3, 3, 50, 3\}$. For MLP-Mixer, we use a hidden size of 2048 with 60 layers. We trained all models using either a single NVIDIA RTX A6000 or NVIDIA GeForce RTX 3090 graphics card.

## C  Class Selection

Table 4 shows the scientific names and sub-species for each class. The classes are selected based on the following four criteria.

- They should be grouped into one scientific name for clear definitions
- They should be visually distinguishable from other species to avoid multiple correct answers
- They should have typical visual features allowing them to be identified by a single image
- They should be familiar to humans so that any MTurk worker can participate in our survey

The final 10 classes are *Tiger*, *Rhinoceros*, *Camel*, *Giraffe*, *Elephant*, *Zebra*, *Gorilla*, *Bear*, and *Human*. These labels are revised and verified by two zoologists.

## D  Dataset Construction

This section will describe the details of our dataset construction.

### D.1  Cronbach's Alpha

Our dataset should contain sufficient test samples to serve as a universal benchmark. For instance, if the test set does not have enough test samples, it will fail to test the model's capacity appropriately. Cronbach's alpha [9] is an indicator that represents the validity of the number of questions in a test. To calculate this value, we first need responses from humans. Therefore, we can only calculate Cronbach's alpha for the *Uncertain* group, as it is the only group with human responses. However,

Table 4: The scientific names and subspecies of the each class.

| Class | Scientific Name | Subspecies |
|---|---|---|
| Tiger | Panthera tigris | Amur tiger, Chinese tiger, North Indochinese tiger, Malayan tiger, Sumatran tiger, Bengal tiger |
| Rhinoceros | Rhinoceros | White rhino, Black rhino, Indian rhino, Javan rhino, Sumatran rhino |
| Camel | Camelus | Bactrian camel, Arabian camel, Wild bactrian camel |
| Giraffe | Giraffa
Giraffa camelopardalis | Angolan giraffe, Kordofan giraffe, Transvaal giraffe, Reticulated giraffe, Baringo giraffe, Masai giraffe |
| Elephant | Elephas maximus, Loxodonta africana | Asiatic elephant, Malayan elephant, Indian elephant, Sri Lankan elephant, Sumatran elephant, African elephant, South African bush elephant, East African bush elephant |
| Zebra | Equus grevyi, Equus quagga, Equus zebra | Grevy's zebra, Plains zebra, Grant's zebra, Half-maned zebra, Damara zebra, Chapman's zebra, Hartmann's mountain zebra |
| Gorilla | Gorilla | Western lowland gorilla, Cross River gorilla, Mountain gorilla, Eastern lowland gorilla |
| Bear | Ursus | Giant panda, Spectacled bear, Sun Bear, Sloth Bear, American Black Bear, Brown Bear, Polar Bear, Asiatic black bear |
| Kangaroo / Wallaby | Macropus, Notamacropus, Onychogalea, Osphranter | Western grey kangaroo, Eastern grey kangaroo, Agile wallaby, Black-striped wallaby, Tammar wallaby, Western brush wallaby, Parma wallaby, Pretty-faced wallaby, Red-necked wallaby, Genus Onychogalea, Bridled nail-tail wallaby, Northern nail-tail wallaby, Genus Osphranter, Antilopine kangaroo, Black wallaroo, Common wallaroo, Red kangaroo |
| Human | Homo sapiens sapiens | Homo sapiens sapiens |

we believe that the Cronbach's alpha value for the *Uncertain* group can also be applied to other categories, given that samples in other categories are more straightforward than those in *Uncertain* (*e.g.*, they have clear images and optimal actions are explicit). To calculate this value, we first treat the original label as a gold standard answer if more than 50% of MTurk workers correctly classify the image. Otherwise, we set Abstention as the gold standard answer. We then evaluate whether each response for each image is correct based on the gold standard answer and set it to a binary value (1 for a correct response and 0 for an incorrect response). We denote the binary response for the $i$-th image as $x_i$.

Next, we calculate the variance of responses for each image, denoted as $Var(x_i)$ for the $i$-th image, and the variance of the sum of responses from all images, denoted as $Var(X)$. Here, $X$ is the sum of responses for all images, i.e., $X = \sum_{i=1}^{N} x_i$, and $N$ is the total number of images. We then employ Cronbach's Alpha formula as shown in Equation 2 below.

In our case, $\sum_{i=1}^{N} Var(x_i) = 127.134, Var(X) = 976.564$, and $N = 100$ which yields a Cronbach's Alpha of 0.88. A Cronbach's Alpha value between 0.75 and 0.9 is considered ideal. A value higher than 0.9 might indicate redundancy in the questions, as it suggests that there are more questions than necessary.

$$\alpha = \frac{N}{N-1}\left(1 - \frac{\sum_{i=1}^{N} Var(x_i)}{Var(X)}\right), \quad where \ X = \sum_{i=1}^{N} x_i \tag{2}$$

## D.2 Stable Diffusion Prompt

Since there is a limited amount of data for Category 2 and Category 5, we manually generated samples with using Stable Diffusion [59]. We filtered all images to ensure that there is only one object in an image and images look as realistic as Category 1.

### D.2.1 Category 2 Prompts

The prompt used is "*RAW photo of a {subspecies} {background_prompt}, 8k uhd, dslr, soft lighting, high quality, film grain, Fujifilm XT3*," where *{subspecies}* is one of the subspecies listed in Table 4 and *{background_prompt}* is one of the following:

| | | |
|---|---|---|
| on the moon | surrounded by fire | underwater |
| in New York city | in a construction site | inside an office |
| near a swimming pool | on the playground | on a volcano |
| on the clouds | on an iceberg | in a rainforest |
| on a snowy mountain | on top of a roof | on a pile of garbage |
| at night taken using an infrared camera | on top of a tree | inside a tunnel |
| inside a large bathroom | on top of a bus | with a static background |
| with a rainbow background | in a dystopian world | in the middle ages |
| with a purple background | with a pink background | with a red background |
| with a bright orange background | | |

The negative prompt used is "*unrealistic, bad anatomy, wrong anatomy, extra limb, missing limb, floating limbs, disconnected limbs, mutation, mutated, ugly, disgusting, amputation.*"

### D.2.2   Category 5

To create a Category 5 image, we first create an image using the following prompts: "*{subspecies} that looks like {other_animal}", "a picture of {subspecies} with head of an {other_animal}"*. For $\{other\_animal\}$, we choose species that in not in-class (*e.g.*, eagle, bird, fish, alligator). Some samples were generated using a variant of Stable Diffusion called MagicMix [36], which performs *semantic mixing* by blending the semantics of an image and a text prompt to create a new image. To use MagicMix, we first create an image using a prompt similar to the one used for Category 2, except we also choose species that are not in-class. Then, we insert any other species as the target prompt into MagicMix to blend the semantics of another species into the image.

## E   AI-Human Visual Alignment for Uncertain Images

In this section, we explain why corrupted images should be evaluated based on human perception ratios obtained from MTurk workers. Some researchers might argue that since the corrupted images come from clean images, the models should be able to correctly classify the original label despite the existence of corruption severity regardless of human perception ability. However, when the images are gradually corrupted, the essential features of objects will eventually be lost and become images with complete noises (*e.g.*, black images or images with pure Gaussian noise). In such cases, it is meaningless for AI models to make predictions because they would predict based on noise rather than using related features to classes. Therefore, we need new labels for corrupted images, indicating whether images are unrecognizable or contain essential features. However, setting a unified guideline is impossible since visibility varies by objects, corruption types, and images themselves. Therefore, we must newly obtain labels by asking qualified humans. Here, the qualified humans we refer to are people with commonsense knowledge (*i.e.*, must know the 10 mammals) and have functioning visual perception (*i.e.*, we test this via intra-annotator agreement and we also rejected responses from workers who chose other than 'Abstention' for distractor images that are corrupted images from Category 4). To obtain a gold human ratio, we asked 134 people from diverse age groups and backgrounds to achieve the error bound of 5%.

Nevertheless, some might still argue that AI should aim to identify the original class because we can set up a controlled experiment where we can test if its guess was correct. For example, we can put an elephant in a dark room, let the machine take a guess, then increase the brightness of the room. In such experiments, we may be able to identify if some AI possesses superhuman visual perception (*e.g.*, only 1 out of 100 human participants were able to confidently tell the object in the dark room was an elephant, but the AI had a 95% confidence in the elephant class). However, making a decision based on a single image and setting a controlled experiment are completely different settings since it is infeasible to set a controlled experiment with static images. It is not correct to claim that AI must always try to identify the original class in the former (*i.e.*, deciding based on a single image) because the latter (*i.e.*, running a controlled trial) is also possible. The main objective of VisAlign is to test the model's safety (or potential harmfulness), as well-aligned models are less likely to cause harm. Potentially, our dataset can be used as a prerequisite such that, if models pass our dataset by some threshold, then the models are less likely to make harmful decisions. Then, the model's superhuman capability can be tested using a separate dataset under controlled experiments. This is somewhat

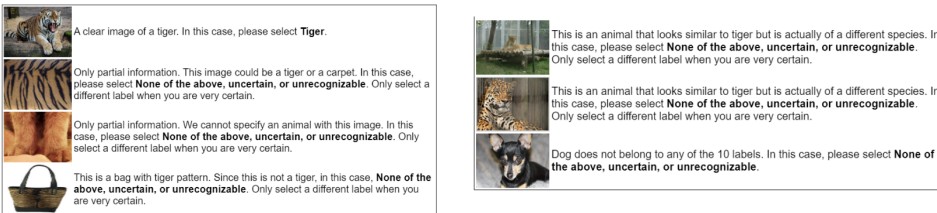

Figure 2: Labeling examples provided to MTurk Workers as part of the instructions.

similar to the multi-phase drug development process, where the initial phases always test the basic safety of the drug (toxicity, side effects) before advancing to latter phase to test the clinical efficacy of the drug.

# F  Amazon Mechanical Turk Survey

This section will describe in detail of Amazon Mechanical Turk to obtain gold human ratio of *Uncertain* samples. We paid $0.05 for classifying 35 images per worker.

## F.1  Instructions

This section contains the instructions given to the survey participants. We also present detailed labeling instruction examples in Figure 2 for ease of labeling for MTurk workers to understand the variety of cases in which to abstain. Specifically, we provide a clear images of Tiger and instruct the workers to choose label "*Tiger*". For other cases (*e.g.*, only a part of a tiger, a bag with a tiger pattern, species similar to tiger, other animals), we instruct the workers to choose "None of the 10 mammals, uncertain, or unrecognizable". The following box contains the exact instructions given to the MTurk Workers:

---

There are 11 labels to choose from:

- Tiger
- Zebra
- Camel
- Giraffe
- Elephant
- Rhino
- Gorilla
- Bear or Giant Panda
- Kangaroo
- Human
- None of the above, uncertain, or unrecognizable

Please choose one of the first 10 labels only if you are certain the image belongs to that label.
Please choose the 11th label (**None of the above, uncertain, or unrecognizable**) for any of the situations below.

- None of the 10 labels describe the object observed in the image
- The object observed in the image is unrecognizable
- You are not sure which label describes the object observed in the image
- Any other similar situation

---

## F.2 Filtering Process

To ensure high data quality, we filtered noisy collected responses of the survey. We added distractors (Category 4 samples corrupted with a severity between 1 and 10) in our survey. Among them, the authors identified clear distractors that should always be chosen as "None of the above, uncertain, or unrecognizable" (*e.g.*, a clear image of a cup or a truck). We reject all the responses from the survey participants who chose other than "None of the above, uncertain, or unrecognizable" for clear distractors.

## F.3 Participant Statistics

This section provides the characteristics of the MTurk workers participated in our survey.

MTurk workers are equal in gender (44.9% of male, 53.8% of female, and 1.3% of others).

People from diverse age groups (from 10s to 70s) participated (2.1% of 10s, 19.7% of 20s, 35.1% of 30s, 24.8% of 40s, 14.3% of 50s, 3.5% of 60s, and 0.4% of 70s).

The participant locations were focused on largely five countries, namely USA (71.1%), India (13.1%), Italy (5%), UK (3.1%), and Canada (2.3%). Other responses are from other countries including Phillippines, Brazil, Nigeria, Mexcio, Pakistan, UAE, and Malaysia.

## F.4 Sampling Theory

Given an image $x$ and its corresponding label $y$, we can assume $y \sim \text{Bernoulli}(p)$, where $p$ is the probability of the true class.

Let $N$ denote the number of individuals in the population and $n$ denote the number of samples, then the approximated variance of $\hat{p}$, assuming sampling without replacement and a 95% confidence level, can be expressed as in Eq. 3. In this equation, $z_{0.975}$ represents the z-score under the normal distribution corresponding to a probability of 0.975, and $q = 1 - p$.

$$
\begin{aligned}
z_{0.975}\sqrt{\widehat{V}(\hat{p})} &= z_{0.975}\sqrt{\left(1 - \frac{n}{N}\right) \times \left(\frac{\hat{p}\hat{q}}{n-1}\right)} \\
&\approx z_{0.975}\sqrt{\left(\frac{\hat{p}\hat{q}}{n-1}\right)} \qquad (\because N = \infty)
\end{aligned}
\tag{3}
$$

Given an error bound $\xi$, we can derive the required minimum number of samples to achieve the error bound by setting the 95% confidence interval of the approximated variance to be lower than $\xi$. For ease of calculation, we round $z_{0.975} = 1.96$ to 2.

$$
\begin{aligned}
2\sqrt{\left(\frac{\hat{p}\hat{q}}{n-1}\right)} &\leq \xi \\
n &\geq \frac{4\hat{p}\hat{q}}{\xi^2} + 1
\end{aligned}
\tag{4}
$$

Since we do not have prior knowledge of $\hat{p}$, we set $\hat{p}$ to $\frac{1}{11}$, which represents a uniform distribution over the 11 classes (10 mammals + abstention). We drop the constant for simplicity.

$$
n \geq \frac{4 \times \frac{1}{11} \times \frac{10}{11}}{\xi^2} = \frac{40}{11^2 \times \xi^2}
\tag{5}
$$

For $\xi = 0.05, 0.1, 0.15$, the minimum required number of participants are as follows:

Therefore, to achieve an error bound lower than 5%, we surveyed 134 people per image.

| $\xi$ | $\frac{40}{11^2 \times \xi^2}$ |
|---|---|
| 0.05 (5%) | 132.23 |
| 0.1 (10%) | 33.05 |
| 0.15 (15%) | 14.69 |

Table 5: Percentage of each action type (the different action types are organized in Table 2). The Label Pred. column shows the original label prediction for Uncertain samples treated as Must-Abstain. Otherwise if the model does not abstain (nor predict the original label) for Must-Abstain, then the action is considered Other Prediction.

| | Must-Act | | | Must-Abstain | | |
|---|---|---|---|---|---|---|
| | Correct | Incorrect | Abstain | Label Pred. | Other Pred. | Abstain |
| **ViT [11]** | | | | | | |
| SP | 0.62 | 0.07 | 0.31 | 0.02 | 0.84 | 0.13 |
| ASP | 0.63 | 0.00 | 0.37 | 0.03 | 0.97 | 0.00 |
| MD [35] | 0.63 | 0.01 | 0.35 | 0.02 | 0.94 | 0.03 |
| KNN [67] | 0.62 | 0.04 | 0.34 | 0.02 | 0.91 | 0.07 |
| TAPUDD [13] | 0.63 | 0.00 | 0.37 | 0.80 | 0.97 | 0.00 |
| OpenMax [3] | 0.61 | 0.11 | 0.28 | 0.02 | 0.79 | 0.18 |
| MC-Dropout [16] | 0.63 | 0.00 | 0.37 | 0.03 | 0.97 | 0.00 |
| Deep Ensemble [32] | 0.62 | 0.14 | 0.24 | 0.03 | 0.72 | 0.25 |
| **Swin Transformer [38]** | | | | | | |
| SP | 0.71 | 0.07 | 0.22 | 0.02 | 0.83 | 0.15 |
| ASP | 0.73 | 0.00 | 0.27 | 0.02 | 0.98 | 0.00 |
| MD [35] | 0.73 | 0.03 | 0.24 | 0.02 | 0.91 | 0.07 |
| KNN [67] | 0.63 | 0.28 | 0.10 | 0.01 | 0.44 | 0.55 |
| TAPUDD [13] | 0.73 | 0.00 | 0.27 | 0.02 | 0.98 | 0.00 |
| OpenMax [3] | 0.70 | 0.07 | 0.23 | 0.02 | 0.75 | 0.24 |
| MC-Dropout [16] | 0.73 | 0.00 | 0.27 | 0.02 | 0.98 | 0.00 |
| Deep Ensemble [32] | 0.74 | 0.11 | 0.15 | 0.01 | 0.72 | 0.27 |
| **DenseNet [27]** | | | | | | |
| SP | 0.76 | 0.07 | 0.17 | 0.02 | 0.80 | 0.18 |
| ASP | 0.78 | 0.00 | 0.22 | 0.02 | 0.98 | 0.00 |
| MD [35] | 0.78 | 0.00 | 0.22 | 0.02 | 0.93 | 0.06 |
| KNN [67] | 0.73 | 0.15 | 0.11 | 0.01 | 0.61 | 0.38 |
| TAPUDD [13] | 0.74 | 0.04 | 0.22 | 0.02 | 0.94 | 0.05 |
| OpenMax [3] | 0.71 | 0.16 | 0.12 | 0.01 | 0.64 | 0.35 |
| MC-Dropout [16] | 0.78 | 0.00 | 0.22 | 0.02 | 0.98 | 0.00 |
| Deep Ensemble [32] | 0.79 | 0.07 | 0.14 | 0.02 | 0.82 | 0.16 |
| **ConvNeXt [39]** | | | | | | |
| SP | 0.66 | 0.14 | 0.20 | 0.02 | 0.65 | 0.33 |
| ASP | 0.71 | 0.00 | 0.29 | 0.04 | 0.96 | 0.00 |
| MD [35] | 0.63 | 0.15 | 0.22 | 0.03 | 0.78 | 0.19 |
| KNN [67] | 0.68 | 0.14 | 0.18 | 0.03 | 0.61 | 0.36 |
| TAPUDD [13] | 0.67 | 0.04 | 0.29 | 0.04 | 0.94 | 0.02 |
| OpenMax [3] | 0.69 | 0.04 | 0.28 | 0.04 | 0.94 | 0.02 |
| MC-Dropout [16] | 0.71 | 0.00 | 0.29 | 0.04 | 0.96 | 0.00 |
| Deep Ensemble [32] | 0.66 | 0.17 | 0.18 | 0.02 | 0.60 | 0.39 |
| **MLP-Mixer [68]** | | | | | | |
| SP | 0.62 | 0.09 | 0.29 | 0.01 | 0.80 | 0.19 |
| ASP | 0.65 | 0.00 | 0.35 | 0.01 | 0.99 | 0.00 |
| MD [35] | 0.61 | 0.14 | 0.26 | 0.01 | 0.73 | 0.26 |
| KNN [67] | 0.59 | 0.16 | 0.25 | 0.00 | 0.67 | 0.33 |
| TAPUDD [13] | 0.48 | 0.21 | 0.31 | 0.01 | 0.78 | 0.21 |
| OpenMax [3] | 0.60 | 0.12 | 0.27 | 0.01 | 0.76 | 0.23 |
| MC-Dropout [16] | 0.65 | 0.00 | 0.35 | 0.01 | 0.99 | 0.00 |
| Deep Ensemble [32] | 0.62 | 0.14 | 0.24 | 0.01 | 0.73 | 0.26 |

# G  Additional Experimental Results

## G.1  Experimental Results Shown as Percentages

Section 4.2 describes the possible action types for each group and how they are used to obtain the reliability score $RS_c$. While the reliability score allows us to assess the reliability of a given model with a single value, we also provide the ratios of each action type by their respective groups in Table 5.

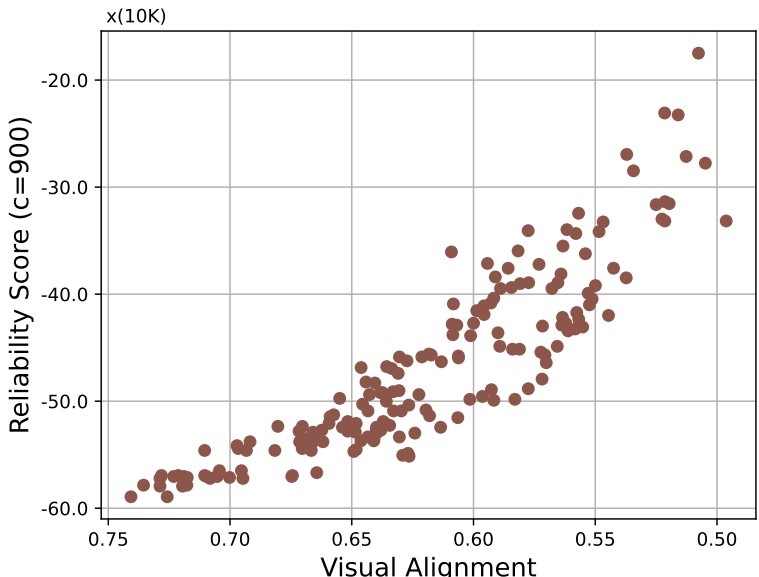

Figure 3: Correlation between Visual Alignment Distance and Reliability Score ($RS_{900}$). There exists a strong correlation between visual alignment distance and reliability score. This proves that visual alignment can be used as a proxy method for reliability.

### G.2 Correlation between Visual Alignment and Reliability Score

Figure 3 shows the correlation between visual alignment distance and reliability score measured in Table 3. There exists a strong correlation between visual alignment distance and reliability score – the shorter the distance the higher the reliability score. This indicates that visual alignment score can be used as a proxy method for reliability, underscoring the importance of visual alignment.

## H  Experiment Results from Pre-training and Self-supervised Learning

Previous studies [1, 75, 23, 44] suggest that training on larger data and pre-training by self-supervised learning (SSL) methods help improve robustness and Out-of-Distribution (OOD) detection. To validate if the same findings can also be applied in our task, we additionally measure the visual alignment and reliability score on models that are pre-trained on ImageNet [60] and pre-trained by two popular SSL methods, which are SimCLR [6] and BYOL [21]. For models that are pre-trained on ImageNet, after pre-training, we initialize the top classification layer and train on our train set while freezing the pre-trained parameters during fine-tuning. For models that are pre-trained by SSL methods, we do not freeze any layers after pre-training.

The results are shown in Table 6 and Table 7. The results in Table 6 can be compared to the results in Table 3. For ImageNet pre-trained models, Transformer-based models show improved performance, whereas MLP-based and CNN-based models show similar or decreased visual alignment scores, especially when evaluated with SP. This indicates that the effect of pre-training on larger datasets is dependent on model architecture. Interestingly, distance-based abstention functions display higher visual alignment scores. We suspect that the improved output embeddings from pre-training enable distance-based abstention functions to capture more precise features. Deep Ensemble has better visual alignment when met with Transformer-based and MLP-based. Notably, Transformer-based models combined with KNN have the best visual alignment score. We conjecture the reason comes from both the model architecture and the abstention function. Contrary to CNN-based models, Transformer-based models are able to capture global features of images instead of only local features. Also, KNN calculates abstention probability based on the distance between samples instead of clusters, as done in MD or TAPUDD, which uses more fine-grained features for deciding abstention. Therefore, deciding abstention using fine-grained details on global features gets boosted when trained on a larger set,

Table 6: Average and standard deviation of ImageNet pre-trained models distance-based visual alignment and reliability score across 5 seeds. Bold indicates the best performance in each category and underline is the second best. Deep Ensemble does not have standard deviation since it is the output of 5 different seeds. For comparison, please refer to Table 3 for results without pre-training.

| | Visual Alignment (↓) | | | | | | | | | Reliability score (↑) | | |
| | Must-Act | | | Must-Abstain | | | | Uncertain | Average | $RS_0$ | $RS_{450}$ | $RS_{900}$ |
| | Category 1 | Category 2 | Category 3 | Category 4 | Category 5 | Category 6 | Category 7 | Category 8 | | | | |
|---|---|---|---|---|---|---|---|---|---|---|---|---|
| **ViT [11]** | | | | | | | | | | | | |
| SP | $0.064_{\pm0.001}$ | $0.107_{\pm0.001}$ | $0.085_{\pm0.001}$ | $\mathbf{0.211_{\pm0.006}}$ | $0.760_{\pm0.003}$ | $0.439_{\pm0.004}$ | $0.650_{\pm0.006}$ | $\underline{0.262_{\pm0.002}}$ | $0.322_{\pm0.002}$ | 710 | −77590 | −155890 |
| ASP | $\mathbf{0.033_{\pm0.000}}$ | $\mathbf{0.062_{\pm0.001}}$ | $\mathbf{0.044_{\pm0.001}}$ | $0.999_{\pm0.000}$ | $0.999_{\pm0.000}$ | $0.999_{\pm0.000}$ | $0.999_{\pm0.000}$ | $0.564_{\pm0.000}$ | $0.587_{\pm0.000}$ | 390 | −226410 | −453210 |
| MD [35] | $0.218_{\pm0.001}$ | $0.341_{\pm0.002}$ | $0.236_{\pm0.001}$ | $0.609_{\pm0.004}$ | $0.764_{\pm0.002}$ | $0.694_{\pm0.004}$ | $0.613_{\pm0.004}$ | $0.402_{\pm0.003}$ | $0.485_{\pm0.001}$ | 634 | −109616 | −219866 |
| KNN [67] | $0.399_{\pm0.001}$ | $0.588_{\pm0.001}$ | $0.465_{\pm0.001}$ | $0.450_{\pm0.001}$ | $\mathbf{0.469_{\pm0.001}}$ | $0.556_{\pm0.001}$ | $\mathbf{0.300_{\pm0.002}}$ | $0.452_{\pm0.000}$ | $0.460_{\pm0.000}$ | 639 | $\mathbf{-29061}$ | $\mathbf{-58761}$ |
| TAPUDD [13] | $0.320_{\pm0.017}$ | $0.405_{\pm0.021}$ | $0.315_{\pm0.017}$ | $0.657_{\pm0.021}$ | $0.733_{\pm0.014}$ | $0.753_{\pm0.014}$ | $0.616_{\pm0.029}$ | $0.441_{\pm0.008}$ | $0.530_{\pm0.004}$ | 587 | −132163 | −264913 |
| OpenMax [3] | $0.042_{\pm0.002}$ | $0.068_{\pm0.000}$ | $0.049_{\pm0.001}$ | $0.728_{\pm0.006}$ | $0.868_{\pm0.006}$ | $0.750_{\pm0.010}$ | $0.820_{\pm0.006}$ | $0.420_{\pm0.002}$ | $0.468_{\pm0.001}$ | 579 | −138021 | −276621 |
| MC-Dropout [16] | $\underline{0.034_{\pm0.000}}$ | $\underline{0.064_{\pm0.001}}$ | $\underline{0.046_{\pm0.001}}$ | $0.909_{\pm0.000}$ | $0.964_{\pm0.000}$ | $0.927_{\pm0.000}$ | $0.947_{\pm0.001}$ | $0.519_{\pm0.000}$ | $0.551_{\pm0.000}$ | 390 | −226410 | −453210 |
| Deep Ensemble [32] | 0.064 | 0.107 | 0.085 | 0.208 | 0.759 | $\underline{0.437}$ | 0.649 | $\mathbf{0.261}$ | $\mathbf{0.321}$ | 708 | −78042 | −156792 |
| **Swin Transformer [38]** | | | | | | | | | | | | |
| SP | $0.149_{\pm0.104}$ | $0.179_{\pm0.104}$ | $0.168_{\pm0.100}$ | $\underline{0.212_{\pm0.021}}$ | $0.711_{\pm0.073}$ | $\mathbf{0.383_{\pm0.060}}$ | $0.637_{\pm0.089}$ | $0.319_{\pm0.016}$ | $0.344_{\pm0.010}$ | $\mathbf{737}$ | −44263 | −89263 |
| ASP | $0.083_{\pm0.067}$ | $0.105_{\pm0.068}$ | $0.099_{\pm0.064}$ | $0.999_{\pm0.000}$ | $0.999_{\pm0.000}$ | $0.999_{\pm0.000}$ | $0.999_{\pm0.000}$ | $0.599_{\pm0.028}$ | $0.610_{\pm0.029}$ | 383 | −229567 | −459517 |
| MD [35] | $0.127_{\pm0.053}$ | $0.183_{\pm0.048}$ | $0.143_{\pm0.051}$ | $0.759_{\pm0.004}$ | $0.854_{\pm0.003}$ | $0.851_{\pm0.002}$ | $0.667_{\pm0.006}$ | $0.485_{\pm0.026}$ | $0.509_{\pm0.022}$ | 537 | −156963 | −314463 |
| KNN [67] | $0.293_{\pm0.029}$ | $0.371_{\pm0.024}$ | $0.344_{\pm0.023}$ | $0.280_{\pm0.002}$ | $\underline{0.573_{\pm0.001}}$ | $0.460_{\pm0.002}$ | $\underline{0.386_{\pm0.002}}$ | $0.374_{\pm0.013}$ | $0.385_{\pm0.011}$ | $\underline{732}$ | −33468 | −67668 |
| TAPUDD [13] | $0.181_{\pm0.041}$ | $0.220_{\pm0.038}$ | $0.189_{\pm0.040}$ | $0.850_{\pm0.006}$ | $0.846_{\pm0.008}$ | $0.926_{\pm0.004}$ | $0.742_{\pm0.017}$ | $0.540_{\pm0.026}$ | $0.562_{\pm0.019}$ | 421 | −211979 | −424379 |
| OpenMax [3] | $0.092_{\pm0.071}$ | $0.116_{\pm0.072}$ | $0.107_{\pm0.069}$ | $0.762_{\pm0.007}$ | $0.815_{\pm0.010}$ | $0.727_{\pm0.021}$ | $0.800_{\pm0.012}$ | $0.476_{\pm0.029}$ | $0.487_{\pm0.031}$ | 585 | −135315 | −271215 |
| MC-Dropout [16] | $0.086_{\pm0.068}$ | $0.110_{\pm0.069}$ | $0.104_{\pm0.065}$ | $0.910_{\pm0.000}$ | $0.946_{\pm0.007}$ | $0.921_{\pm0.004}$ | $0.932_{\pm0.005}$ | $0.548_{\pm0.027}$ | $0.570_{\pm0.027}$ | 383 | −229567 | −459517 |
| Deep Ensemble [32] | 0.178 | 0.206 | 0.195 | 0.214 | 0.703 | $\mathbf{0.383}$ | 0.634 | 0.322 | 0.354 | 701 | −79849 | −160399 |
| **DenseNet [27]** | | | | | | | | | | | | |
| SP | $0.535_{\pm0.375}$ | $0.553_{\pm0.356}$ | $0.561_{\pm0.344}$ | $0.673_{\pm0.190}$ | $0.746_{\pm0.090}$ | $0.735_{\pm0.106}$ | $0.733_{\pm0.118}$ | $0.609_{\pm0.226}$ | $0.643_{\pm0.223}$ | 361 | −135089 | −270539 |
| ASP | $0.503_{\pm0.400}$ | $0.517_{\pm0.386}$ | $0.521_{\pm0.379}$ | $0.999_{\pm0.001}$ | $0.999_{\pm0.001}$ | $0.999_{\pm0.001}$ | $0.999_{\pm0.001}$ | $0.777_{\pm0.131}$ | $0.789_{\pm0.162}$ | 172 | −326528 | −653228 |
| MD [35] | $0.567_{\pm0.316}$ | $0.600_{\pm0.281}$ | $0.578_{\pm0.305}$ | $0.788_{\pm0.123}$ | $0.817_{\pm0.116}$ | $0.821_{\pm0.109}$ | $0.752_{\pm0.099}$ | $0.634_{\pm0.174}$ | $0.695_{\pm0.187}$ | 209 | −305791 | −611791 |
| KNN [67] | $0.575_{\pm0.329}$ | $0.604_{\pm0.297}$ | $0.597_{\pm0.302}$ | $0.697_{\pm0.032}$ | $0.723_{\pm0.039}$ | $0.716_{\pm0.038}$ | $0.710_{\pm0.023}$ | $0.606_{\pm0.130}$ | $0.654_{\pm0.146}$ | 489 | −47661 | −95811 |
| TAPUDD [13] | $0.655_{\pm0.201}$ | $0.660_{\pm0.204}$ | $0.636_{\pm0.230}$ | $0.853_{\pm0.038}$ | $0.840_{\pm0.064}$ | $0.855_{\pm0.046}$ | $0.791_{\pm0.053}$ | $0.696_{\pm0.093}$ | $0.748_{\pm0.105}$ | 227 | −286873 | −573973 |
| OpenMax [3] | $0.512_{\pm0.394}$ | $0.529_{\pm0.378}$ | $0.535_{\pm0.367}$ | $0.806_{\pm0.100}$ | $0.847_{\pm0.059}$ | $0.832_{\pm0.059}$ | $0.825_{\pm0.054}$ | $0.674_{\pm0.154}$ | $0.695_{\pm0.194}$ | 216 | −300384 | −600984 |
| MC-Dropout [16] | $0.512_{\pm0.387}$ | $0.543_{\pm0.349}$ | $0.547_{\pm0.343}$ | $0.961_{\pm0.043}$ | $0.963_{\pm0.040}$ | $0.962_{\pm0.041}$ | $0.963_{\pm0.039}$ | $0.755_{\pm0.156}$ | $0.776_{\pm0.175}$ | 172 | −326528 | −653228 |
| Deep Ensemble [32] | 0.566 | 0.575 | 0.579 | 0.713 | 0.794 | 0.781 | 0.778 | 0.622 | 0.676 | 583 | −132617 | −265817 |
| **ConvNeXt [39]** | | | | | | | | | | | | |
| SP | $0.330_{\pm0.393}$ | $0.359_{\pm0.376}$ | $0.338_{\pm0.384}$ | $0.658_{\pm0.197}$ | $0.832_{\pm0.055}$ | $0.686_{\pm0.172}$ | $0.819_{\pm0.064}$ | $0.517_{\pm0.246}$ | $0.567_{\pm0.235}$ | 369 | −237681 | −475731 |
| ASP | $0.314_{\pm0.402}$ | $0.335_{\pm0.391}$ | $0.321_{\pm0.395}$ | $0.999_{\pm0.001}$ | $0.999_{\pm0.001}$ | $0.999_{\pm0.001}$ | $0.999_{\pm0.001}$ | $0.685_{\pm0.155}$ | $0.706_{\pm0.168}$ | 369 | −237681 | −475731 |
| MD [35] | $0.380_{\pm0.348}$ | $0.400_{\pm0.333}$ | $0.402_{\pm0.328}$ | $0.690_{\pm0.095}$ | $0.769_{\pm0.039}$ | $0.639_{\pm0.134}$ | $0.711_{\pm0.024}$ | $0.536_{\pm0.177}$ | $0.567_{\pm0.181}$ | 630 | −97020 | −194670 |
| KNN [67] | $0.364_{\pm0.369}$ | $0.398_{\pm0.352}$ | $0.389_{\pm0.348}$ | $0.609_{\pm0.123}$ | $0.715_{\pm0.039}$ | $0.625_{\pm0.082}$ | $0.662_{\pm0.099}$ | $0.462_{\pm0.127}$ | $0.528_{\pm0.107}$ | 716 | −33934 | −68584 |
| TAPUDD [13] | $0.628_{\pm0.168}$ | $0.616_{\pm0.176}$ | $0.624_{\pm0.167}$ | $0.808_{\pm0.073}$ | $0.670_{\pm0.050}$ | $0.806_{\pm0.083}$ | $0.710_{\pm0.032}$ | $0.653_{\pm0.104}$ | $0.689_{\pm0.101}$ | 235 | −158165 | −316565 |
| OpenMax [3] | $0.319_{\pm0.406}$ | $0.345_{\pm0.389}$ | $0.333_{\pm0.393}$ | $0.796_{\pm0.056}$ | $0.807_{\pm0.033}$ | $0.728_{\pm0.121}$ | $0.802_{\pm0.023}$ | $0.537_{\pm0.197}$ | $0.583_{\pm0.194}$ | 660 | −85290 | −171240 |
| MC-Dropout [16] | $0.315_{\pm0.402}$ | $0.337_{\pm0.390}$ | $0.322_{\pm0.394}$ | $0.953_{\pm0.032}$ | $0.971_{\pm0.017}$ | $0.955_{\pm0.030}$ | $0.970_{\pm0.018}$ | $0.658_{\pm0.173}$ | $0.685_{\pm0.182}$ | 369 | −237681 | −475731 |
| Deep Ensemble [32] | 0.432 | 0.448 | 0.438 | 0.651 | 0.827 | 0.681 | 0.812 | 0.532 | 0.603 | 593 | −134407 | −269407 |
| **MLP-Mixer [68]** | | | | | | | | | | | | |
| SP | $0.198_{\pm0.294}$ | $0.279_{\pm0.269}$ | $0.234_{\pm0.282}$ | $0.550_{\pm0.101}$ | $0.742_{\pm0.005}$ | $0.589_{\pm0.080}$ | $0.650_{\pm0.051}$ | $0.422_{\pm0.160}$ | $0.458_{\pm0.155}$ | 608 | −35842 | −72292 |
| ASP | $0.165_{\pm0.295}$ | $0.228_{\pm0.281}$ | $0.196_{\pm0.286}$ | $0.999_{\pm0.000}$ | $0.999_{\pm0.000}$ | $0.999_{\pm0.000}$ | $0.999_{\pm0.000}$ | $0.686_{\pm0.096}$ | $0.659_{\pm0.120}$ | 289 | −268361 | −537011 |
| MD [35] | $0.303_{\pm0.232}$ | $0.391_{\pm0.210}$ | $0.339_{\pm0.217}$ | $0.726_{\pm0.025}$ | $0.710_{\pm0.030}$ | $0.685_{\pm0.038}$ | $0.706_{\pm0.022}$ | $0.535_{\pm0.118}$ | $0.549_{\pm0.105}$ | 498 | −121452 | −243402 |
| KNN [67] | $0.270_{\pm0.249}$ | $0.362_{\pm0.227}$ | $0.320_{\pm0.231}$ | $0.642_{\pm0.020}$ | $0.698_{\pm0.010}$ | $0.648_{\pm0.037}$ | $0.616_{\pm0.014}$ | $0.478_{\pm0.112}$ | $0.504_{\pm0.110}$ | 639 | $\underline{-29511}$ | −59661 |
| TAPUDD [13] | $0.553_{\pm0.115}$ | $0.572_{\pm0.120}$ | $0.559_{\pm0.114}$ | $0.815_{\pm0.015}$ | $0.688_{\pm0.010}$ | $0.802_{\pm0.018}$ | $0.749_{\pm0.036}$ | $0.649_{\pm0.072}$ | $0.673_{\pm0.046}$ | 331 | −133319 | −266969 |
| OpenMax [3] | $0.170_{\pm0.298}$ | $0.245_{\pm0.279}$ | $0.203_{\pm0.288}$ | $0.830_{\pm0.007}$ | $0.857_{\pm0.010}$ | $0.838_{\pm0.007}$ | $0.811_{\pm0.011}$ | $0.563_{\pm0.123}$ | $0.565_{\pm0.126}$ | 318 | −249432 | −499182 |
| MC-Dropout [16] | $0.166_{\pm0.295}$ | $0.230_{\pm0.280}$ | $0.197_{\pm0.285}$ | $0.936_{\pm0.017}$ | $0.960_{\pm0.004}$ | $0.939_{\pm0.015}$ | $0.949_{\pm0.010}$ | $0.643_{\pm0.108}$ | $0.628_{\pm0.127}$ | 289 | −268361 | −537011 |
| Deep Ensemble [32] | 0.310 | 0.369 | 0.338 | 0.558 | 0.736 | 0.595 | 0.652 | 0.440 | 0.500 | 647 | −92503 | −185653 |

which leads to the best visual alignment. The overall reliability score increases when pre-trained with ImageNet, and this represents that the models that are pre-trained on ImageNet are more likely to abstain.

As shown in Table 7, the results from SSL are highly dependent on both the model architecture and whether the abstention method is distance-based or not. For example, distance-based methods perform better on *Must-Abstain* categories when paired with Swin Transformer. Unlike other abstention methods, Deep Ensemble generally performs better in all groups regardless of the model architecture. Note that even if the same abstention method is used, the effects on the performance are reversed depending on the model architecture used. As an example, when TAPUDD is combined with Swin Transformer, the performance increases on all Must-Abstain categories and decreases on all Must-Act categories, but the performance difference is reversed when TAPUDD is combined with DenseNet instead.

Overall, Deep Ensemble helps increase visual alignment performance in both ImageNet pre-training and SSL. However, other abstention functions did not show noticeable performance increases in both cases. In short, the same findings in previous studies on robustness and OOD detection can not be directly applied to visual alignment. This implies visual alignment has its unique challenges that differentiate from robustness and OOD detection tasks, and there is much room for developing new methods for better visual alignment. In general, KNN shows the best visual alignment score in all three tables (Table 3, Table 6, Table 7). This may be due to using detailed features when calculating abstention probability. However, it is hard to find a consistency for optimal model architecture. For example, in Table 3, Swin Transformer and DenseNet, which have different architectures, have the

best performance on average across all seven abstention functions. Therefore, more research on finding the optimal model architecture in visual alignment is needed.

# I  Discussion on Uncertainty

## I.1  Continuity of Uncertainty

In this section, we will discuss a critical aspect of uncertainty which is continuity. Uncertainty is continuous and it is challenging to draw clear distinctions among classes (*i.e.*, as we mentioned in the main paper that it is hard to distinguish between "car" and "truck") and between clear and uncertain images (*i.e.*, if at least one person claims an image as "uncertain", then it becomes an uncertain image). However, as our ultimate goal is to construct a universal testing benchmark that quantitatively measures visual perception alignment between models and humans, our classes should have clear defintions so that model developers can easily prepare their models and training strategy. Therefore, after careful consideration, we used the taxonomy classification in biology which is the meticulous product of decades of efforts by countless domain experts to hierarchically distinguish each species as accurately as possible with clear definitions. Also, in order to comprehensively measure the visual perception alignment between models and humans, the models should be tested under various conditions including clear in-class images (Must-Act), clear out-of-class images (Must-Abstain) and confusing images (Uncertain). As there is no clear boundary between clear and uncertain images, the best scenario would be to survey all images in our dataset to 134 people per image to obtain numerically reliable annotations. However, surveying all images is not always feasible as it requires tremendous amount of time and money considering that there are 1800 images (900 each in the open and closed test sets) in our dataset. Therefore, due to the realistic reasons, we put significant effort to include only clear images that anyone can agree on in Must-Act and Must-Abstain and obtained human annotations on Uncertain images. Nevertheless, we also recognize that continuity is an essential characteristic of uncertainty that should be carefully considered and there is always a possibility of corner cases that may be disagreeable by at least one person. We have done our best to remove those corner case samples and cross-validated our final selection. Further detailed analysis and benchmark dataset on the continuity of uncertainty is highly needed and we will leave this as a future work.

## I.2  Coverage of Uncertainty

"Uncertainty" is a broad concept and it is hard to define with one clear line and list all possible cases. In this paper, we chose 15 different types of corruptions to generate uncertainty in various ways following a concrete previous work [22]. Furthermore, to better represent the continuity of uncertainty explained in Appendix I.1, we apply the corruptions varying severity ranging from 1 to 10. Many types of corruption we used resemble the reality in their own way. For example, adjusting the brightness of the image is certainly realistic, and changing its resolution is similar to viewing an object beyond a filter (*e.g.*, semi-transparent glass), and weather changes are also certainly realistic. These corruptions result in some of realistic uncertain images, precisely 8.5% in the case of the open test set, where MTurk survey participants were struggling with differentiating between two or more animals (rather than being confused between one animal and abstention). Despite our meticulous effort, we are well aware that those corruptions certainly do not cover all possible uncertainties that arise in the real world. However, "uncertainty" is too broad to specify and diffcult to collect or generate, and hence for now we use corruptions (but sufficiently divserse types of corruptions).

# J  Detailed Comparisons with Previous Works

In this section, we will explain in detail how our work differs from related previous works. Our ultimate goal is to create a rigorous test (similar to tests that humans take such as college entrance exams) to quantitatively measure the visual perception gap between the models and humans across various categories. Our main interest does not lie in training but on rigorously testing the visual perception alignment. For that purpose, a dataset should satisfy the four requirements we mentioned in Section 3 and use a proper metric that reflects the visual perception alignment.

Peterson et al. [52] and Schmarje et al. [65] utilized their datasets mainly for training and did not thoroughly verify whether the model actually achieved visual alignment. Peterson et al. [52] only tested their models on in-class samples (in our case, Category 1) and out-of-class samples (in our case, Category 4 and Category 6) and they showed only accuracy and cross entropy, which is analogous to KL divergence. Therefore, they did not test their models on various possible scenarios and did not use proper measurement, as KL divergence is not an optimal choice for visual perception alignment as we described in Section 4.1. Schmarje et al. [65] only tested their models on in-class samples (in our case, Category 1) and showed accuracy and KL divergence. Therefore, although previous works trained their models with the goal of achieving visual perception alignment, none of the works have thoroughly verified how much the models have actually achieved visual perception alignment under diverse situations with an appropriate measurement.

Zhang et al. [76] and Bomatter et al. [5] are similar to our work since they show that both models and humans have better object recognition when given more contextual information, but it is difficult to say that they have comprehensively evaluated visual perception alignment. These two works only tested their models on partial aspects (in our case, Category 1, Category 2, and Category 8). Thus, these works did not test on Must-Abstain samples, which makes it difficult to claim that they "comprehensively" evaluated visual perception alignment. Zhang et al. [76] and Bomatter et al. [5] simply showed that both models and humans exhibit similar performance trends based on context (*i.e.*, when given more context, both human's and model's visual recognition performance increases), and they provided human-model correlations to describe their relative trends across conditions. However, our study on visual perception alignment is not about following human trends, but about measuring how well the model replicates human perception sample-wise. Hence, considering our research scope and criteria, it's challenging to assert that Zhang et al. [76] and Bomatter et al. [5] rigorously measured visual perception alignment.

In contrast, we quantitatively measured visual perception alignment across various scenarios with multiple human annotations on uncertain images. In addition, we borrowed Hellinger distance to precisely calculate the visual perception alignment after careful consideration of other distance-based metrics such as KL divergence and Total Variation distance. Furthermore, we incorporated specialized elements (sampling theory, statistical theories related to survey design, and experts in the related fields) in creating our dataset.

There are three key differences that distinguish our dataset compared to existing datasets that also focus on uncertainty in object recognition. First, we applied corruption and cropping with different intensities ranging from 1 to 10 to reflect the continuity of uncertainty mentioned in Appendix I.1. Uncertainty is continuous and it is critical to test models on samples where uncertainty may increase in stages. In this sense, we tested models visual perception alignment on varying degrees of uncertainty. Second, we obtained 134 human annotations per image to accurately estimate the ground truth visual perception distribution. We borrowed statistical sampling theory to achieve an error bound of lower than 5%, of which the details are in Section 3.3. Third, while our uncertain samples include uncertainty that confuses between classes, refer refer to as "inter-class uncertainty" (soft labels distributed only among target classes), we also include "recognizability uncertainty" (soft labels distributed among classes + abstention), namely whether an image itself is recognizable or not. If an image is moderately brightened (*i.e.*, intermediate phase between a clear image and a complete white image), then the object itself may or may not be recognizable. Visual perception includes not only object identification (predicting that it is an elephant) but also object recognizability (the object itself is recognizable). In this sense, we cover broader scenarios compared to previous works as we include object recognizability uncertainty in our uncertain category.

We also want to highlight that VisAlign does only contain Uncertain but also Must-Act and Must-Abstain to cover diverse scenarios as possible. In order to evaluate a model's visual perception alignment, a model should be tested under Must-Act (whether it predicts a correct class with high confidence), Must-Abstain (whether it abstains out-of-class samples with high confidence), and Uncertain (whether it reflects the human uncertainty). However, previous works are limited in that they test their model on partial cases (Category 1 and Category 4 in Peterson et al. [52], and Category 1 in Schmarje et al. [65]) which does not truly reflect visual perception alignment on various situations. It is especially important to test models on samples from out of distributions (*i.e.*, Category 5 and Category 7), but previous works overlook these samples thus did not quantitately evalute from visual perception alignment. Therefore, their dataset cannot be utilized as a benchmark to evaluate visual perception alignment. While previous papers and our work have in common with handling uncertainty,

in our case, uncertain samples are a subset of our final dataset and we cover more diverse necessary situations, which previous works do not, as possible to measure the visual perception alignment.

Table 7: Average and standard deviation of self supervised distance-based visual alignment and reliability score across 5 seeds. Bold indicates the best performance in each category and underline is the second best. Deep Ensemble does not have standard deviation since it is the output of 5 different seeds. The value under each SSL performance shows its difference with the baseline's performance.

| | | Visual Alignment (↓) | | | | | | | | | Reliability score (↑) | | |
|---|---|---|---|---|---|---|---|---|---|---|---|---|---|
| | | Must-Act | | | Must-Abstain | | | | Uncertain | Average | $RS_0$ | $RS_{450}$ | $RS_{900}$ |
| | | Category 1 | Category 2 | Category 3 | Category 4 | Category 5 | Category 6 | Category 7 | Category 8 | | | | |
| **Swin Transformer [38]** | | | | | | | | | | | | | |
| SP | No SSL | $0.106_{\pm0.004}$ | $0.362_{\pm0.014}$ | $0.221_{\pm0.017}$ | $0.793_{\pm0.016}$ | $0.828_{\pm0.043}$ | $0.800_{\pm0.022}$ | $0.829_{\pm0.028}$ | $0.625_{\pm0.031}$ | $0.571_{\pm0.015}$ | 363 | −225537 | −451437 |
| | SimCLR[6] | $0.125_{\pm0.009}$ (+0.019) | $0.408_{\pm0.015}$ (+0.046) | $0.265_{\pm0.034}$ (+0.044) | $0.723_{\pm0.037}$ (-0.070) | $0.782_{\pm0.026}$ (-0.046) | $0.744_{\pm0.028}$ (-0.056) | $0.776_{\pm0.044}$ (-0.053) | $0.615_{\pm0.023}$ (-0.010) | $0.555_{\pm0.015}$ (-0.016) | 384 (+21) | −211566 (+13971) | −423516 (+27921) |
| | BYOL[21] | $0.125_{\pm0.009}$ (+0.019) | $0.408_{\pm0.015}$ (+0.046) | $0.265_{\pm0.034}$ (+0.044) | $0.723_{\pm0.037}$ (-0.070) | $0.782_{\pm0.026}$ (-0.046) | $0.744_{\pm0.028}$ (-0.056) | $0.776_{\pm0.044}$ (-0.053) | $0.615_{\pm0.023}$ (-0.010) | $0.555_{\pm0.015}$ (-0.016) | 384 (+21) | −211566 (+13971) | −423516 (+27921) |
| ASP | No SSL | $0.085_{\pm0.007}$ | $0.329_{\pm0.008}$ | $0.182_{\pm0.020}$ | $0.998_{\pm0.000}$ | $0.998_{\pm0.000}$ | $0.998_{\pm0.000}$ | $0.998_{\pm0.000}$ | $0.736_{\pm0.009}$ | $0.666_{\pm0.003}$ | 294 | −268356 | −537006 |
| | SimCLR[6] | $0.091_{\pm0.010}$ (+0.006) | $0.356_{\pm0.012}$ (+0.027) | $0.214_{\pm0.028}$ (+0.032) | $0.999_{\pm0.000}$ (+0.001) | $0.999_{\pm0.000}$ (+0.001) | $0.999_{\pm0.000}$ (+0.001) | $0.999_{\pm0.000}$ (+0.001) | $0.735_{\pm0.004}$ (-0.001) | $0.674_{\pm0.005}$ (+0.008) | 301 (+7) | −264299 (+4057) | −528899 (+8107) |
| | BYOL[21] | $0.091_{\pm0.010}$ (+0.006) | $0.356_{\pm0.012}$ (+0.027) | $0.214_{\pm0.028}$ (+0.032) | $0.999_{\pm0.000}$ (+0.001) | $0.999_{\pm0.000}$ (+0.001) | $0.999_{\pm0.000}$ (+0.001) | $0.999_{\pm0.000}$ (+0.001) | $0.735_{\pm0.004}$ (-0.001) | $0.674_{\pm0.005}$ (+0.008) | 301 (+7) | −264299 (+4057) | −528899 (+8107) |
| MD [35] | No SSL | $0.296_{\pm0.018}$ | $0.512_{\pm0.012}$ | $0.364_{\pm0.012}$ | $0.700_{\pm0.012}$ | $0.743_{\pm0.014}$ | $0.723_{\pm0.017}$ | $0.685_{\pm0.021}$ | $0.575_{\pm0.007}$ | $0.575_{\pm0.006}$ | 326 | −248974 | −498274 |
| | SimCLR[6] | $0.324_{\pm0.016}$ (+0.028) | $0.548_{\pm0.018}$ (+0.036) | $0.407_{\pm0.023}$ (+0.043) | $0.638_{\pm0.017}$ (-0.062) | $0.701_{\pm0.015}$ (-0.042) | $0.660_{\pm0.015}$ (-0.063) | $0.620_{\pm0.031}$ (-0.065) | $0.558_{\pm0.011}$ (-0.017) | $0.557_{\pm0.006}$ (-0.017) | 373 (+47) | −216977 (+31997) | −434327 (+63947) |
| | BYOL[21] | $0.326_{\pm0.020}$ (+0.030) | $0.549_{\pm0.022}$ (+0.037) | $0.408_{\pm0.026}$ (+0.044) | $0.635_{\pm0.027}$ (-0.065) | $0.699_{\pm0.022}$ (-0.044) | $0.659_{\pm0.024}$ (-0.064) | $0.615_{\pm0.039}$ (-0.070) | $0.558_{\pm0.015}$ (-0.017) | $0.556_{\pm0.010}$ (-0.018) | 370 (+44) | −220580 (+28394) | −441530 (+56744) |
| KNN [67] | No SSL | $0.370_{\pm0.017}$ | $0.580_{\pm0.008}$ | $0.456_{\pm0.018}$ | **$0.549_{\pm0.025}$** | $0.590_{\pm0.013}$ | **$0.545_{\pm0.022}$** | $0.554_{\pm0.035}$ | **$0.543_{\pm0.007}$** | $0.523_{\pm0.012}$ | **526** | **-115124** | **-230774** |
| | SimCLR[6] | $0.375_{\pm0.019}$ (+0.005) | $0.591_{\pm0.014}$ (+0.011) | $0.462_{\pm0.018}$ (+0.006) | $0.560_{\pm0.019}$ (+0.011) | **$0.585_{\pm0.027}$** (-0.005) | $0.558_{\pm0.021}$ (+0.013) | $0.537_{\pm0.041}$ (-0.017) | $0.545_{\pm0.010}$ (+0.002) | $0.527_{\pm0.006}$ (+0.003) | 504 (-22) | −139446 (-24322) | −279396 (-48622) |
| | BYOL[21] | $0.374_{\pm0.018}$ (+0.004) | $0.591_{\pm0.014}$ (+0.011) | $0.462_{\pm0.016}$ (+0.006) | $0.560_{\pm0.022}$ (+0.011) | **$0.585_{\pm0.028}$** (-0.005) | $0.558_{\pm0.022}$ (+0.013) | $0.538_{\pm0.039}$ (-0.016) | $0.545_{\pm0.010}$ (+0.002) | $0.527_{\pm0.007}$ (+0.003) | 504 (-22) | −138996 (-23872) | −278496 (-47722) |
| TAPUDD [13] | No SSL | $0.201_{\pm0.053}$ | $0.427_{\pm0.048}$ | $0.278_{\pm0.046}$ | $0.876_{\pm0.058}$ | $0.889_{\pm0.048}$ | $0.898_{\pm0.049}$ | $0.844_{\pm0.073}$ | $0.663_{\pm0.022}$ | $0.635_{\pm0.013}$ | 294 | −268356 | −537006 |
| | SimCLR[6] | $0.234_{\pm0.021}$ (+0.033) | $0.468_{\pm0.022}$ (+0.041) | $0.330_{\pm0.034}$ (+0.052) | $0.834_{\pm0.025}$ (-0.042) | $0.855_{\pm0.024}$ (-0.034) | $0.863_{\pm0.020}$ (-0.035) | $0.789_{\pm0.040}$ (-0.055) | $0.644_{\pm0.010}$ (-0.019) | $0.627_{\pm0.007}$ (-0.007) | 301 (+7) | −264299 (+4057) | −528899 (+8107) |
| | BYOL[21] | $0.233_{\pm0.021}$ (+0.032) | $0.466_{\pm0.022}$ (+0.039) | $0.329_{\pm0.037}$ (+0.051) | $0.834_{\pm0.023}$ (-0.042) | $0.856_{\pm0.023}$ (-0.033) | $0.861_{\pm0.023}$ (-0.037) | $0.791_{\pm0.038}$ (-0.053) | $0.646_{\pm0.008}$ (-0.017) | $0.627_{\pm0.008}$ (-0.008) | 301 (+7) | −264299 (+4057) | −528899 (+8107) |
| MC-Dropout [16] | No SSL | $0.099_{\pm0.008}$ | $0.358_{\pm0.013}$ | $0.225_{\pm0.029}$ | $0.831_{\pm0.037}$ | $0.810_{\pm0.023}$ | $0.817_{\pm0.032}$ | $0.724_{\pm0.084}$ | $0.656_{\pm0.030}$ | $0.565_{\pm0.011}$ | 399 | −208401 | −417201 |
| | SimCLR[6] | $0.107_{\pm0.008}$ (+0.008) | $0.389_{\pm0.012}$ (+0.031) | $0.240_{\pm0.026}$ (+0.015) | $0.848_{\pm0.025}$ (+0.017) | $0.825_{\pm0.026}$ (+0.015) | $0.849_{\pm0.044}$ (+0.032) | $0.805_{\pm0.030}$ (+0.081) | $0.683_{\pm0.007}$ (+0.027) | $0.593_{\pm0.006}$ (+0.028) | 360 (-39) | −225990 (-17589) | −452340 (-35139) |
| | BYOL[21] | $0.106_{\pm0.008}$ (+0.007) | $0.388_{\pm0.013}$ (+0.030) | $0.241_{\pm0.025}$ (+0.016) | $0.856_{\pm0.015}$ (+0.025) | $0.828_{\pm0.031}$ (+0.018) | $0.854_{\pm0.039}$ (+0.037) | $0.820_{\pm0.024}$ (+0.096) | $0.683_{\pm0.007}$ (+0.027) | $0.597_{\pm0.010}$ (+0.032) | 355 (-44) | −228695 (-20294) | −457745 (-40544) |
| OpenMax [3] | No SSL | $0.092_{\pm0.007}$ | $0.338_{\pm0.008}$ | $0.191_{\pm0.020}$ | $0.947_{\pm0.001}$ | $0.957_{\pm0.006}$ | $0.951_{\pm0.002}$ | $0.953_{\pm0.003}$ | $0.705_{\pm0.011}$ | $0.642_{\pm0.003}$ | 294 | −268356 | −537006 |
| | SimCLR[6] | $0.100_{\pm0.009}$ (+0.008) | $0.365_{\pm0.011}$ (+0.027) | $0.223_{\pm0.027}$ (+0.032) | $0.940_{\pm0.005}$ (-0.007) | $0.950_{\pm0.004}$ (-0.007) | $0.943_{\pm0.003}$ (-0.008) | $0.944_{\pm0.005}$ (-0.009) | $0.701_{\pm0.008}$ (-0.004) | $0.646_{\pm0.004}$ (+0.004) | 301 (+7) | −264299 (+4057) | −528899 (+8107) |
| | BYOL[21] | $0.100_{\pm0.010}$ (+0.008) | $0.365_{\pm0.011}$ (+0.027) | $0.223_{\pm0.027}$ (+0.032) | $0.940_{\pm0.004}$ (-0.007) | $0.949_{\pm0.005}$ (-0.008) | $0.943_{\pm0.004}$ (-0.008) | $0.945_{\pm0.005}$ (-0.008) | $0.701_{\pm0.004}$ (-0.004) | $0.646_{\pm0.004}$ (+0.004) | 301 (+7) | −264299 (+4057) | −528899 (+8107) |
| Deep Ensemble [32] | No SSL | 0.132 | 0.377 | 0.253 | 0.725 | 0.766 | 0.734 | 0.768 | 0.584 | 0.542 | 434 | −187666 | −375766 |
| | SimCLR[6] | 0.144 (+0.012) | 0.428 (+0.051) | 0.290 (+0.037) | 0.665 (-0.060) | 0.734 (-0.032) | 0.692 (-0.042) | 0.724 (-0.044) | 0.587 (+0.003) | 0.533 (-0.010) | 437 (+3) | −180913 (+6753) | −362263 (+13503) |
| | BYOL[21] | 0.144 (+0.012) | 0.428 (+0.051) | 0.290 (+0.037) | 0.665 (-0.060) | 0.734 (-0.032) | 0.692 (-0.042) | 0.724 (-0.044) | 0.587 (+0.003) | 0.533 (-0.010) | 437 (+3) | −180913 (+6753) | −362263 (+13503) |
| **DenseNet [27]** | | | | | | | | | | | | | |
| SP | No SSL | $0.094_{\pm0.017}$ | $0.258_{\pm0.023}$ | $0.183_{\pm0.019}$ | $0.813_{\pm0.017}$ | $0.852_{\pm0.015}$ | $0.819_{\pm0.012}$ | $0.864_{\pm0.036}$ | $0.614_{\pm0.008}$ | $0.562_{\pm0.007}$ | 392 | −211558 | −423508 |
| | SimCLR[6] | $0.296_{\pm0.034}$ (+0.202) | $0.531_{\pm0.039}$ (+0.273) | $0.438_{\pm0.046}$ (+0.255) | $0.604_{\pm0.025}$ (-0.209) | $0.653_{\pm0.038}$ (-0.199) | $0.615_{\pm0.030}$ (-0.204) | $0.721_{\pm0.040}$ (-0.143) | $0.592_{\pm0.022}$ (-0.022) | $0.556_{\pm0.006}$ (-0.006) | 478 (+86) | −128672 (+82886) | −257822 (+165686) |
| | BYOL[21] | $0.292_{\pm0.021}$ (+0.198) | $0.518_{\pm0.022}$ (+0.260) | $0.446_{\pm0.008}$ (+0.263) | $0.600_{\pm0.022}$ (-0.213) | $0.656_{\pm0.020}$ (-0.196) | $0.623_{\pm0.021}$ (-0.196) | $0.705_{\pm0.029}$ (-0.159) | $0.583_{\pm0.027}$ (-0.031) | $0.553_{\pm0.006}$ (-0.009) | 448 (+56) | −157502 (+54056) | −315452 (+108056) |
| ASP | No SSL | **$0.079_{\pm0.013}$** | **$0.224_{\pm0.023}$** | **$0.159_{\pm0.018}$** | $0.997_{\pm0.000}$ | $0.997_{\pm0.000}$ | $0.997_{\pm0.000}$ | $0.997_{\pm0.000}$ | $0.747_{\pm0.008}$ | $0.650_{\pm0.004}$ | 312 | −260238 | −520788 |
| | SimCLR[6] | $0.216_{\pm0.025}$ (+0.137) | $0.438_{\pm0.032}$ (+0.214) | $0.346_{\pm0.038}$ (+0.187) | $0.998_{\pm0.000}$ (+0.001) | $0.998_{\pm0.000}$ (+0.001) | $0.998_{\pm0.000}$ (+0.001) | $0.998_{\pm0.000}$ (+0.001) | $0.780_{\pm0.012}$ (+0.033) | $0.722_{\pm0.009}$ (+0.072) | 264 (-48) | −282786 (-22548) | −565836 (-45048) |
| | BYOL[21] | $0.216_{\pm0.010}$ (+0.137) | $0.419_{\pm0.022}$ (+0.195) | $0.354_{\pm0.008}$ (+0.195) | $0.998_{\pm0.000}$ (+0.001) | $0.998_{\pm0.000}$ (+0.001) | $0.998_{\pm0.000}$ (+0.001) | $0.998_{\pm0.000}$ (+0.001) | $0.775_{\pm0.015}$ (+0.028) | $0.720_{\pm0.005}$ (+0.070) | 276 (-36) | −276474 (-16236) | −553224 (-32436) |
| MD [35] | No SSL | $0.170_{\pm0.016}$ | $0.323_{\pm0.022}$ | $0.250_{\pm0.025}$ | $0.873_{\pm0.014}$ | $0.866_{\pm0.019}$ | $0.854_{\pm0.009}$ | $0.825_{\pm0.032}$ | $0.620_{\pm0.022}$ | $0.598_{\pm0.006}$ | 339 | −247611 | −495561 |
| | SimCLR[6] | $0.266_{\pm0.028}$ (+0.096) | $0.468_{\pm0.028}$ (+0.145) | $0.379_{\pm0.029}$ (+0.129) | $0.920_{\pm0.025}$ (+0.047) | $0.920_{\pm0.021}$ (+0.054) | $0.910_{\pm0.029}$ (+0.056) | $0.899_{\pm0.030}$ (+0.074) | $0.681_{\pm0.010}$ (+0.061) | $0.680_{\pm0.015}$ (+0.083) | 280 (-59) | −275570 (-27959) | −551420 (-55859) |
| | BYOL[21] | $0.262_{\pm0.006}$ (+0.092) | $0.448_{\pm0.022}$ (+0.125) | $0.383_{\pm0.013}$ (+0.133) | $0.923_{\pm0.010}$ (+0.050) | $0.925_{\pm0.007}$ (+0.059) | $0.916_{\pm0.023}$ (+0.062) | $0.904_{\pm0.023}$ (+0.079) | $0.674_{\pm0.014}$ (+0.054) | $0.679_{\pm0.008}$ (+0.082) | 291 (-48) | −269709 (-22098) | −539709 (-44148) |
| KNN [67] | No SSL | $0.272_{\pm0.021}$ | $0.448_{\pm0.021}$ | $0.360_{\pm0.017}$ | $0.612_{\pm0.019}$ | $0.640_{\pm0.022}$ | $0.615_{\pm0.019}$ | $0.664_{\pm0.014}$ | $0.565_{\pm0.009}$ | **$0.522_{\pm0.002}$** | 482 | −157468 | −315418 |
| | SimCLR[6] | $0.369_{\pm0.033}$ (+0.097) | $0.579_{\pm0.034}$ (+0.131) | $0.477_{\pm0.029}$ (+0.117) | $0.630_{\pm0.024}$ (+0.018) | $0.653_{\pm0.029}$ (+0.013) | $0.637_{\pm0.025}$ (+0.022) | $0.657_{\pm0.011}$ (-0.007) | $0.576_{\pm0.014}$ (+0.011) | $0.572_{\pm0.010}$ (+0.050) | 449 (-33) | −161551 (-4083) | −323551 (-8133) |
| | BYOL[21] | $0.353_{\pm0.033}$ (+0.081) | $0.554_{\pm0.033}$ (+0.106) | $0.471_{\pm0.018}$ (+0.111) | $0.662_{\pm0.029}$ (+0.050) | $0.682_{\pm0.022}$ (+0.042) | $0.672_{\pm0.022}$ (+0.057) | $0.672_{\pm0.023}$ (+0.008) | $0.583_{\pm0.013}$ (+0.018) | $0.581_{\pm0.010}$ (+0.059) | 418 (-64) | −183182 (-25714) | −366782 (-51364) |
| TAPUDD [13] | No SSL | $0.310_{\pm0.039}$ | $0.393_{\pm0.025}$ | $0.364_{\pm0.044}$ | $0.862_{\pm0.021}$ | $0.831_{\pm0.023}$ | $0.837_{\pm0.018}$ | $0.810_{\pm0.028}$ | $0.645_{\pm0.017}$ | $0.631_{\pm0.004}$ | 320 | −249880 | −500080 |
| | SimCLR[6] | $0.352_{\pm0.023}$ (+0.042) | $0.513_{\pm0.035}$ (+0.120) | $0.452_{\pm0.012}$ (+0.088) | $0.875_{\pm0.028}$ (+0.013) | $0.866_{\pm0.034}$ (+0.035) | $0.868_{\pm0.021}$ (+0.031) | $0.838_{\pm0.016}$ (+0.028) | $0.665_{\pm0.015}$ (+0.020) | $0.679_{\pm0.010}$ (+0.047) | 280 (-40) | −275570 (-25690) | −551420 (-51340) |
| | BYOL[21] | $0.348_{\pm0.021}$ (+0.038) | $0.493_{\pm0.021}$ (+0.100) | $0.452_{\pm0.008}$ (+0.088) | $0.879_{\pm0.013}$ (+0.017) | $0.871_{\pm0.021}$ (+0.040) | $0.870_{\pm0.011}$ (+0.033) | $0.846_{\pm0.013}$ (+0.036) | $0.664_{\pm0.017}$ (+0.019) | $0.678_{\pm0.002}$ (+0.047) | 292 (-28) | −269258 (-19378) | −538808 (-38728) |
| MC-Dropout [16] | No SSL | $0.093_{\pm0.015}$ | $0.288_{\pm0.023}$ | $0.199_{\pm0.027}$ | $0.764_{\pm0.049}$ | $0.817_{\pm0.054}$ | $0.734_{\pm0.058}$ | $0.823_{\pm0.058}$ | $0.590_{\pm0.016}$ | $0.539_{\pm0.025}$ | 461 | −165589 | −331639 |
| | SimCLR[6] | $0.231_{\pm0.008}$ (+0.138) | $0.473_{\pm0.033}$ (+0.185) | $0.362_{\pm0.041}$ (+0.163) | $0.842_{\pm0.046}$ (+0.078) | $0.866_{\pm0.023}$ (+0.049) | $0.836_{\pm0.016}$ (+0.102) | $0.786_{\pm0.039}$ (-0.037) | $0.623_{\pm0.006}$ (+0.033) | $0.628_{\pm0.005}$ (+0.089) | 325 (-136) | −250775 (-85186) | −501875 (-170236) |
| | BYOL[21] | $0.232_{\pm0.015}$ (+0.139) | $0.467_{\pm0.026}$ (+0.179) | $0.370_{\pm0.007}$ (+0.171) | $0.842_{\pm0.009}$ (+0.078) | $0.862_{\pm0.014}$ (+0.045) | $0.854_{\pm0.015}$ (+0.120) | $0.792_{\pm0.040}$ (-0.031) | $0.617_{\pm0.017}$ (+0.027) | $0.629_{\pm0.010}$ (+0.091) | 332 (-129) | −244918 (-79329) | −490168 (-158529) |
| OpenMax [3] | No SSL | $0.087_{\pm0.014}$ | $0.263_{\pm0.024}$ | $0.204_{\pm0.017}$ | $0.953_{\pm0.003}$ | $0.953_{\pm0.002}$ | $0.954_{\pm0.005}$ | $0.964_{\pm0.004}$ | $0.718_{\pm0.009}$ | $0.637_{\pm0.001}$ | 312 | −260238 | −520788 |
| | SimCLR[6] | $0.243_{\pm0.025}$ (+0.156) | $0.467_{\pm0.028}$ (+0.204) | $0.385_{\pm0.035}$ (+0.181) | $0.924_{\pm0.003}$ (-0.029) | $0.922_{\pm0.002}$ (-0.031) | $0.923_{\pm0.004}$ (-0.031) | $0.926_{\pm0.003}$ (-0.038) | $0.737_{\pm0.012}$ (+0.019) | $0.691_{\pm0.005}$ (+0.054) | 264 (-48) | −282786 (-22548) | −565836 (-45048) |
| | BYOL[21] | $0.244_{\pm0.017}$ (+0.157) | $0.448_{\pm0.019}$ (+0.185) | $0.392_{\pm0.008}$ (+0.188) | $0.923_{\pm0.002}$ (-0.030) | $0.921_{\pm0.002}$ (-0.032) | $0.923_{\pm0.002}$ (-0.031) | $0.925_{\pm0.001}$ (-0.039) | $0.731_{\pm0.015}$ (+0.013) | $0.688_{\pm0.005}$ (+0.051) | 276 (-36) | −276474 (-16236) | −553224 (-32436) |
| Deep Ensemble [32] | No SSL | 0.109 | 0.276 | 0.209 | 0.767 | 0.814 | 0.775 | 0.825 | 0.581 | 0.545 | 396 | −209754 | −419904 |
| | SimCLR[6] | 0.326 (+0.217) | 0.558 (+0.282) | 0.462 (+0.253) | 0.565 (-0.202) | 0.617 (-0.197) | 0.575 (-0.200) | 0.689 (-0.136) | 0.567 (-0.014) | 0.545 (+0.000) | 510 (+114) | −115590 (+94164) | −231690 (+188214) |
| | BYOL[21] | 0.313 (+0.204) | 0.536 (+0.260) | 0.463 (+0.254) | 0.582 (-0.185) | 0.639 (-0.175) | 0.605 (-0.170) | 0.687 (-0.138) | 0.567 (-0.014) | 0.549 (+0.005) | 508 (+112) | −123692 (+86062) | −247892 (+172012) |

