# OpenReview forum: "VisAlign: Dataset for Measuring the Alignment between AI and Humans in Visual Perception"
_NeurIPS.cc/2023/Track/Datasets_and_Benchmarks — NeurIPS 2023 Datasets and Benchmarks Poster_

### Official Review · Reviewer_ChH6 · 2023-07-05
**Contributions are limited; concatenation of existing datasets to solve existing problems with limited novel insights**

**Rating:** 7
**Confidence:** 5

**Strengths:**

1. The paper is generally well-written.
2. The problem of measuring visual alignment between ai and humans in visual perception is an important and interesting problem.
3. Multiple aspects of alignments are covered and assessed. This includes 3 supercateogories and 8 fine-grained subcategories.
4. The dataset design is well-structured and well-organized (some concerns are expressed below).

**Additional Feedback:**

The quality of the online Mturk experiments is not always good. Have the authors introduced additional controls to ensure the quality of the data collected? If so, please provide implementation details on the controls. For example, how does the paper make sure that the participants are paying attention to the screen when doing the experiments, and are there foil trials to check participants understanding of the tasks?

**Clarity:**

The writing is good and easy to follow.
However, some dataset details are missing, such as IRB, and mturk experiment design and specifications about control conditions.

**Correctness:**

Limited contributions and lack of novelty are my main concerns about the paper. Detailed justification for my arguments:

1. The datasets are limited to the re-compilation of existing datasets used for testing model generalization and robustness. If these existing datasets have already been used for testing state-of-the-art models, what are the new insights or novelties we can gain from this work? Other than simply unifying the datasets, models, and metrics, what are the other contributions (such as new insights, and new suggestions about new model developments)?

2. The paper argues that this is the first work measuring visual perception alignment with humans. However, this is not true. I listed several example works here which have already studied some aspects of visual alignments with humans in object recognition, such as context-aware object recognition [1][2], generalization, inductive biases, and robustness [3][4].
Please cite these works and discuss the key differences between these works.

3. Open-set object recognition problem is a well-known problem in computer vision. This work shares similar motivations with this line of research. An example review paper is here [5]. Please discuss why this study is beneficial for open-set object recognition task. How would the open-set recognition methods play a role here? Have all the open-set recognition methods been tested or adapted for the problem studied here?

[1] Putting visual object recognition in context, CVPR, 2020

[2] When Pigs Fly: Contextual Reasoning in Synthetic and Natural Scenes, ICCV, 2021

[3] ImageNet-trained CNNs are biased towards texture; increasing shape bias improves accuracy and robustness, ICLR, 2019

[4] Understanding Robustness of Transformers for Image Classification, ICCV, 2021

[5] Recent Advances in Open Set Recognition: A Survey, TPAMI, 2020

**Documentation:**

The paper promises to release the code and data upon acceptance.

**Ethics:**

No ethical concerns, as far as I can see.

I searched for the IRB for experiments conducted on Amazon Mechanical Turks in the main and supplementary material; and I could not find any information related to this. Have approvals been obtained from the relevant institutions?

**Limitations:**

See the limitations and suggestions below.

**Opportunities For Improvement:**

See the limitations and suggestions below.

**Relation To Prior Work:**

See limitations above.
Some references are missing
The papers fail to compare with some existing works and benchmark these methods on the dataset.

**Summary And Contributions:**

The paper proposes a dataset measuring visual alignment between ai and humans in object recognition tasks. The dataset contains groups of images such as must-act, must-abstain and uncertain. 12 models and methods are tested and their results are analyzed.

---

> ### Author Response · Authors · 2023-08-24
> **Response to Reviewer ChH6 (1/4)**
>
> Thank you for the time and effort spent in carefully reviewing our work. Please kindly find the responses below.
>
> **Q. The datasets are limited to the re-compilation of existing datasets used for testing model generalization and robustness. If these existing datasets have already been used for testing state-of-the-art models, what are the new insights or novelties we can gain from this work? Other than simply unifying the datasets, models, and metrics, what are the other contributions (such as new insights, and new suggestions about new model developments)?**
>
>  We first want to emphasize that our dataset is not a mere re-compilation of existing datasets. We established four requirements that a dataset should meet to serve as a universal visual alignment benchmark. Although not mentioned explicitly in the original submission, the arguments in Section 3 follow these four requirements (the respective arguments’ locations in the original submission are mentioned at the end of each requirement):
> 1) Clear definition for each class: Each class must have a clear definition so that model developers can easily prepare their models and training strategies. An example where this is not satisfied can be found in existing datasets like CIFAR-10, where the definition between ‘automobile’ and ‘truck’ is not perfectly distinct and can vary by individual. After careful consideration, we chose the taxonomic classification in biology, a meticulous product of expert effort to hierarchically distinguish species. In lines 96-100, we mention that strict class definitions are critical.
> 2) Class familiarity to average individuals: Since the Uncertain samples are constructed using an Amazon Mechanical Turk survey with 134 people per sample, classes should be recognizable without requiring specialized knowledge. We mention the importance of familiarity in lines 100-105.
> 3) Coverage of diverse and realistic scenarios: This is vital for accurate visual alignment assessment. This is the reason we have a total of eight categories, as mentioned in lines 112-113.
> 4) Ground truth labels for each sample: This is critical as we need human visual perception labels for all samples to calculate visual perception alignment. We explain why accurately estimating the ground truth distribution, as we did using 134 labels per sample, is important in Section 3.3.
>
> We listed these requirements at the beginning of Section 3 in the revised paper. Despite the importance of these requirements, existing datasets fail to meet at least one of these requirements, as shown in Table 1 on page 7 of the revised paper.
>
> There are a total of 8 categories in our benchmark. While we borrowed existing datasets in certain categories (Categories 1, 4, 6, 7) that met the requirements, we manually created samples when no existing datasets were available (Category 2, 3, 5, 8). The authors put significant effort into creating images, especially in Category 2 and 5, where no existing datasets are available. We manually fine-tuned prompts given to Stable Diffusion and filtered all samples to ensure they met the requirements. Details are in Appendix D of the supplementary material.
>
> Another distinguishing feature of our dataset is the collection of gold human perception labels in “Uncertain”. While there are datasets with corruptions (ImageNet-C [1], Park et al. [2]), those datasets lack gold human labels and cannot be applied directly to our task. We surveyed 134 people per image to achieve an error bound of less than 5% to reflect the general population gold human perception label.
>
> In conclusion, existing datasets are unqualified for quantitatively measuring visual alignment, and our dataset cannot be created by simply compiling existing datasets. The authors put significant effort into building a benchmark that employs carefully chosen classification targets across 8 different settings (even if you try to simply compile multiple datasets, carefully harmonizing the classes alone is a challenging task already) with every sample having ground-truth human labels, thus meeting the above mentioned four requirements.
>
> There are three key insights that can be gained from experiments. First, abstention functions have a greater influence on visual perception alignment than model architecture. Second, distance-based methods (MD, KNN, and TAPUDD) perform well in Must-Act while probability-based methods (SP and ASP) excel in Must-Abstain. From these results, we propose that a new abstention function that considers both distance and probability might be needed to achieve better visual alignment. Third, KNN has demonstrated superior overall performance. We conjecture that this is because, unlike methods that calculate distance by clustering samples (MD, TAPUDD), KNN can capture subtle characteristics of individual samples since it calculates distance on a per-sample rather than clustering. Parts of Section 5.2 have been updated in the revision to include some missing insights.

---

> > ### Author Response · Authors · 2023-08-24
> > **Response to Reviewer ChH6 (2/4)**
> >
> > **Q. The paper argues that this is the first work measuring visual perception alignment with humans. However, this is not true. I listed several example works here which have already studied some aspects of visual alignments with humans in object recognition, such as context-aware object recognition, generalization, inductive biases, and robustness. Please cite these works and discuss the key differences between these works.**
> >
> > Thank you for introducing the related works. Zhang et al. (2020) [3] and Bomatter et al. (2021) [4] have small relevance with our work, since they demonstrate that both models and humans have better object recognition when given more context information (e.g. It is easier to recognize a computer mouse when there is a computer keyboard beside it). Additionally, Geirhos et al. (2019) [5] and Bhojanapalli et al. (2021) [6] are not quite relevant to AI-human visual alignment in that they tested the model's robustness in object recognition against perturbations that do not affect object identifiability.
> >
> > Our work differs from the above mentioned papers in that we quantitatively measured visual perception alignment between AI and humans. In order to achieve this goal, visual alignment should be tested under diverse situations including clear photo-realistic images, images with uncommon backgrounds, slightly corrupted images, images with different styles, and images of different objects. While previous works have focused on specific situations (like varying context or altering object textures), they have not provided a comprehensive measure of visual alignment.
> >
> > Our dataset is divided into three main groups (Must-Act, Must-Abstain, and Uncertain) and further categorized into eight categories. We aimed to cover as many scenarios as possible. Additionally, we carefully selected Hellinger distance among other distances to correctly reflect visual alignment. We evaluated five models and seven abstention functions on our benchmark. Thus, we described our study as the “first work that comprehensively explores visual perception alignment between models and humans”. Thank you for suggesting the related works, we added them in the related works in line 104 page 3 in the revised paper.
> >
> > **Q. Open-set object recognition problem is a well-known problem in computer vision. This work shares similar motivations with this line of research. Please discuss why this study is beneficial for open-set object recognition task. How would the open-set recognition methods play a role here? Have all the open-set recognition methods been tested or adapted for the problem studied here?**
> >
> > Thank you for your comment. We first want to emphasize that the motivations behind open-set recognition (OSR) and our work are distinct. OSR aims to achieve high accuracy with in-class samples while rejecting samples from out-of-classes. In contrast, our work strives to evaluate whether AI models possess well-aligned visual perception as humans across various situations, including Must-Act (in-class samples), Must-Abstain (out-of-class samples), and Uncertain.
> >
> > OSR typically aims to correctly predict Category 1 samples while rejecting Category 4 samples. Some OSR studies do also consider adversarial images (i.e., Category 3 samples), as described in the mentioned review paper [7]. Therefore it can be seen as a sub-problem of AI-human visual perception alignment, which encompasses a broader discourse by proposing to measure how similar an AI model’s visual perception is to human visual perception in diverse settings that are not typically considered OSR, such as samples of different styles (Category 7) and Uncertain images (Category 8).
> >
> > Therefore, if an AI model’s visual perception is perfectly aligned with human visual perception, then it will have automatically solved OSR. However, the experiment results in our paper tell us that visual alignment is far from solved, and new methods that are more advanced than typical OOD detection methods or uncertainty estimation methods are required. And this is where OSR methods can play a role, because they aim to partially address AI-human visual alignment, just like OOD detection methods and uncertainty estimation methods.
> >
> > We have explored the connection further by running an experiment using OpenMax [8], a well-known OSR method. OpenMax satisfies the three conditions of our abstention function selection criteria, which are: 1) applicable on any model architecture, 2) do not require Out-of-Distribution or other Must-Abstain samples during training, and 3) do not require a supplementary model.The results can be found in the revision in Table 3 on page 9. OpenMax shows strong performance especially in Must-Abstain but has low visual perception alignment in Must-Act. This verifies that while open-set object recognition might have potential to contribute to AI-human visual alignment, its current solution is likely to only partially achieve the entire objective.

---

> > > ### Author Response · Authors · 2023-08-24
> > > **Response to Reviewer ChH6 (3/4)**
> > >
> > > **Q. Some dataset details are missing, and mturk experiment design and specifications about control conditions.**
> > >
> > > MTurk experiment designs are primarily related to Section 3.3 which describes the acquisition of human labels in Uncertain samples. Due to space limitation, we could not explain this thoroughly in the main paper. However, we included the necessary details in the supplementary materials as stated “Details on survey instructions, response filtering process, and participant statistics are provided in the Appendix.” in line 191 page 5 of the original manuscript (line 248, page 6 of the revised manuscript).
> > >
> > > In the supplementary material, we detail the instructions given to the MTurk workers, the process for filtering and rejecting low-quality responses, and demographic information of the workers. Specifically, we informed workers that they had 11 choices (10 mammals + abstention) for each image and described situations in which they must choose ‘abstain’. Examples of these situations include: “None of the 10 labels describe the object observed in the image”, “The object observed in the image is unrecognizable”, “You are not sure which label describes the object observed in the image”, and “Any other similar situation”. To aid understanding, we provided images with optimal answers, as seen in Figure 2 page 20 (page 22 in the revision) of the supplementary material.
> > >
> > > Since the survey was conducted via the MTurk platform online, we could not manually control workers’ survey conditions, such as resolution or brightness of monitors. However, we added distractor images (from Category 4) where qualified workers should always choose ‘abstain’. If a worker chose something other than ‘abstain’ (e.g, ‘elephant’ or ‘giraffe’) for these distractors, we rejected all responses from that worker. Related content can be found in lines 177 on page 5 of the original manuscript (line 232, page 6 of the revised manuscript).
> > >
> > > **Q. The quality of the online Mturk experiments is not always good. Have the authors introduced additional controls to ensure the quality of the data collected? If so, please provide implementation details on the controls. For example, how does the paper make sure that the participants are paying attention to the screen when doing the experiments, and are there foil trials to check participants understanding of the tasks?**
> > >
> > > It is a valid concern regarding the quality of the MTurk responses. It is practically impossible to hire only the MTurkers who have good eyesight, have high-quality monitors, conduct the survey in a brightly lit room, etc. Therefore we adopted two measures, which we already described in the original manuscript. Firstly, in the MTurk survey, we included control samples from Category 4 (12% of total surveyed images) for which the MTurk workers always have to choose ‘abstain’ (i.e. ensure that the MTurkers were not just randomly choosing answers). We rejected all responses from the workers who chose other than ‘abstain’ on these control samples. We mentioned the control samples in line 177 page 5 of the original manuscript (line 232, page 6 in the revised manuscript).
> > >
> > > Secondly, we calculated intra-annotator consistency, which evaluates whether responses from an individual worker are consistent (i.e. workers were consciously making an effort to best answer the questions). To do this, we inserted two sets of identical images and considered the answers from the worker to be consistent if they choose the same answers for these identical images. We obtained a Fleiss’ Kappa value of 0.91 for intra-annotator consistency, indicating almost perfect agreement. This value suggests that MTurk workers were attentive during experiments and understood our task well. Intra-annotator consistency is described in line 189 page 5 of the original manuscript (line 247, page 6 in the revised manuscript).

---

> > > > ### Author Response · Authors · 2023-08-24
> > > > **Response to Reviewer ChH6 (4/4)**
> > > >
> > > > **References**
> > > >
> > > > [1] Hendrycks, Dan, and Thomas Dietterich. "Benchmarking neural network robustness to common corruptions and perturbations." arXiv preprint arXiv:1903.12261 (2019).
> > > >
> > > > [2] Park, Jeonghoon, et al. "Natural attribute-based shift detection." arXiv preprint arXiv:2110.09276 (2021).
> > > >
> > > > [3]  Zhang, Mengmi, Claire Tseng, and Gabriel Kreiman. "Putting visual object recognition in context." Proceedings of the IEEE/CVF conference on computer vision and pattern recognition. 2020.
> > > >
> > > > [4]  Bomatter, Philipp, et al. "When pigs fly: Contextual reasoning in synthetic and natural scenes." Proceedings of the IEEE/CVF International Conference on Computer Vision. 2021.
> > > >
> > > > [5]  Geirhos, Robert, et al. "ImageNet-trained CNNs are biased towards texture; increasing shape bias improves accuracy and robustness." arXiv preprint arXiv:1811.12231 (2018).
> > > >
> > > > [6]  Bhojanapalli, Srinadh, et al. "Understanding robustness of transformers for image classification." Proceedings of the IEEE/CVF international conference on computer vision. 2021.
> > > >
> > > > [7]  Recent Advances in Open Set Recognition: A Survey, TPAMI, 2020
> > > >
> > > > [8] A. Bendale and T. E. Boult, “Towards open set deep networks,” Proceedings of the IEEE conference on computer vision and pattern recognition, pp. 1563–1572, 2016.

---

> > > > > ### Comment · Reviewer_ChH6 · 2023-08-25
> > > > > **Acknowledgement of reading the authors' responses**
> > > > >
> > > > > Thank you for providing the responses to all my questions above. Some of the questions are addressed. However, the two remaining issues below are still unresolved. At the moment, I would like to keep my score as it is:
> > > > >
> > > > > 1. The term on "visual perception alignment" is a broad topic. It is NOT only about classifying must-act, abstain, and uncertain. The papers I mentioned above involve alignment of visual diets (humans use intensive data sampling within a scene versus AI models sample iid from random scenes), and alignment of visual perception when both models and humans are presented with contextual images. The current work only focuses on one single aspect of visual perception alignment.
> > > > >
> > > > > 2. Yes, open-set recognition models will NOT correctly classify all four categories and there will continue to be a performance gap. However, it is interesting to adapt the algorithms to solve the current task and quantitatively assess the gap of these SOTA models in the current tasks. The baseline comparisons are limited.

---

> > > > > > ### Author Response · Authors · 2023-08-26
> > > > > > **Response to Reviewer ChH6**
> > > > > >
> > > > > > We sincerely thank the reviewer for the insightful and informative feedback. We carefully address the points raised in the reviews. Please find the individual responses as below.
> > > > > >
> > > > > > **Q. The term on "visual perception alignment" is a broad topic. It is NOT only about classifying must-act, abstain, and uncertain. The papers I mentioned above involve alignment of visual diets (humans use intensive data sampling within a scene versus AI models sample iid from random scenes), and alignment of visual perception when both models and humans are presented with contextual images. The current work only focuses on one single aspect of visual perception alignment.**
> > > > > >
> > > > > > As you pointed out, visual perception alignment is indeed broad and, in fact, it could be said that it covers a countless number of cases. Thus, due to the broad nature of visual perception alignment, creating a dataset that satisfies all possible situations is practically impossible, not to mention in a single study. Hence, we did not claim we covered all aspects of visual perception alignment, and we explicitly stated the scope of our work to be focused on the most fundamental visual perception alignment, specifically in the context of image classification as we mentioned in line 30 on page 2 (“we use image classification as the target task, which is more fundamental to machine perception”).
> > > > > >
> > > > > > The contribution of our work lies in our pioneering attempt to comprehensively measure visual alignment for the first time. While the existing papers you mentioned also delve into the visual perception of human and AI (mostly for training purposes), our work is distinct in that: 1) We based our dataset creation process on sampling theory, statistical theories related to survey design, and insights from experts in relevant fields; 2) We meticulously measured visual alignment across diverse scenarios and provided a suitable visual alignment metric.
> > > > > >
> > > > > > Lastly, as you mentioned, addressing other aspects of visual alignment (such as visual diet) is meaningful work, but it is impossible to address all relevant aspects of visual perception in a single trial, especially when it is the first trial ever. (Note that we use the term “first” in terms of trying to rigorously build a test set based on sampling theory and statistical theory to quantitatively measure the visual perception gap between humans and AI across diverse scenarios). Considering that this is the first attempt ever, and the rigorous effort put into designing and constructing the test benchmark across 8 categories, it is unreasonable to deny the merit of this work simply because it started with the fundamental object recognition with three action choices (act, abstain, uncertain).
> > > > > >
> > > > > > **Q. Yes, open-set recognition models will NOT correctly classify all four categories and there will continue to be a performance gap. However, it is interesting to adapt the algorithms to solve the current task and quantitatively assess the gap of these SOTA models in the current tasks. The baseline comparisons are limited.**
> > > > > >
> > > > > > As you requested, we provided discussion regarding the relationship between open-set recognition (OSR) to our dataset, and you yourself are agreeing to our discussion.  As you requested, we additionally tested OpenMax, which is a popular OSR method, on our dataset and added it in the revised paper. From the experiments, we found that OpenMax is not a perfect solution and there is a room for improvement to achieve better visual perception alignment.
> > > > > >
> > > > > > Given that we have addressed your comments to the best of our effort, we believe it would make the review process far more constructive to provide a more concrete suggestion/comment than simply concluding “The baseline comparisons are limited”. Practically, it is difficult for us to run ALL existing SOTA OSR algorithms in the world on our dataset. We already explained why OSR methods will not solve all 8 categories (which you agreed to), and we demonstrated that with a popular OSR method. If there is any particular OSR method you think we must add to our experiment to fill the logical hole in our paper, please feel free to tell us, and we will run additional experiments as long as there is time.
> > > > > >
> > > > > > However, note that we also tested probability-based and distance-based out-of-distribution (OOD) detection methods and uncertainty measurements on our dataset, which is already comprehensive enough. OSR is just another possible means to decide whether to abstain or not. So if you want to suggest another OSR method besides OpenMax, please keep in mind that it should have distinct characteristics than all other baselines (prob-based OODs, distance-based OODs, uncertainty measures, OpenMax). And keep in mind our abstention function selection criteria as well (1. must be applicable on any model architecture, 2. does not require OOD or other Must-Abstain samples during training, and 3. do not require a supplementary model, mentioned at line 332 page 8)

---

> > > > > > > ### Comment · Reviewer_ChH6 · 2023-08-28
> > > > > > > **Acknowledgement of reading the authors' responses**
> > > > > > >
> > > > > > > Thank you for providing the responses to all my questions above. My question 2 is resolved (thus, increasing my score to 5). However, I still have concerns about the first question.
> > > > > > >
> > > > > > > Q1 follow-up: The authors claim "While the existing papers you mentioned also delve into the visual perception of human and AI (mostly for training purposes)...". This is not true for some papers, e.g. [3] and [4]. Both [3] and [4] curate TEST set images for object recognition under various contextual conditions to study visual perception alignment. The models are NOT trained on these contextual images beforehand. Thus, the claim made by the authors: "Note that we use the term “first” in terms of trying to rigorously build a test set based on sampling theory and statistical theory to quantitatively measure the visual perception gap between humans and AI across diverse scenarios." is invalid. Please correct the relevant statements in the manuscript and the rebuttal; otherwise, provide justification as to why this is the "first" piece of work studying visual perception alignment.
> > > > > > >
> > > > > > > Minor: It would be great if the authors could point out the exact line number of the committed changes in the revised manuscript.
> > > > > > >
> > > > > > > [3] Zhang, Mengmi, Claire Tseng, and Gabriel Kreiman. "Putting visual object recognition in context." Proceedings of the IEEE/CVF conference on computer vision and pattern recognition. 2020.
> > > > > > >
> > > > > > > [4] Bomatter, Philipp, et al. "When pigs fly: Contextual reasoning in synthetic and natural scenes." Proceedings of the IEEE/CVF International Conference on Computer Vision. 2021.

---

> > > > > > > > ### Author Response · Authors · 2023-08-28
> > > > > > > > **Response to Reviewer ChH6 (1/2)**
> > > > > > > >
> > > > > > > > We thank the reviewer for the second comment and are grateful that our responses have resolved your second question. For the first question, please find the response below.
> > > > > > > >
> > > > > > > > **Q. The authors claim "While the existing papers you mentioned also delve into the visual perception of human and AI (mostly for training purposes)...". This is not true for some papers, e.g. [3] and [4]. Both [3] and [4] curate TEST set images for object recognition under various contextual conditions to study visual perception alignment. The models are NOT trained on these contextual images beforehand. Thus, the claim made by the authors: "Note that we use the term “first” in terms of trying to rigorously build a test set based on sampling theory and statistical theory to quantitatively measure the visual perception gap between humans and AI across diverse scenarios." is invalid. Please correct the relevant statements in the manuscript and the rebuttal; otherwise, provide justification as to why this is the "first" piece of work studying visual perception alignment.**
> > > > > > > >
> > > > > > > > We mentioned Peterson et al. [1] and Schmarje et al. [2] as the previous research that mostly used visual perception dataset for training purposes.
> > > > > > > >
> > > > > > > > To clarify, we revised our claim (as you can check from our discussion with Reviewer 82Bt) to make our contribution scope much more specific: “This is the first work to **rigorously build a test set based on sampling theory and statistical theory to quantitatively measure** the visual perception gap between humans and AI **across diverse scenarios.**” We would like to provide more explanation.
> > > > > > > >
> > > > > > > > Both papers you mention are similar to our work since they show that both models and humans have better object recognition when given more contextual information, but it is difficult to say that they comprehensively evaluated visual perception alignment. For a comprehensive evaluation, a dataset must satisfy our four requirements mentioned in the previous response, tested under diverse scenarios, and must use an appropriate metric for measuring alignment. But these two works only tested their models on partial aspects (the full context is equivalent to our Category 1, texture only and incongruent to Category 2, minimal context and blurred context to Category 8) Thus, these works did not test on Must-Abstain samples, which makes it difficult to claim that they “comprehensively” evaluated visual perception alignment.
> > > > > > > >
> > > > > > > > In the two papers you mentioned, they simply showed that both models and humans exhibit similar performance trends based on context (i.e., when given more context, both human’s and model’s visual recognition performance increases), and they provided human-model correlations to describe their relative trends across conditions. However, our study on visual perception alignment is not about following human trends, but about measuring how well the model replicates human perception sample-wise. Hence, considering our research scope and criteria, it's challenging to assert that [3] and [4] rigorously measured visual perception alignment.
> > > > > > > >
> > > > > > > > In contrast, we quantitatively measured visual perception alignment across various scenarios with multiple human annotations on uncertain images. Additionally, we adopted the Hellinger distance for precise calculation of visual perception alignment after careful consideration of other distance-based metrics like KL divergence and Total Variation distance. Furthermore, we incorporated specialized elements (sampling theory, statistical theories related to survey design, and experts in the related fields) in creating our dataset.
> > > > > > > >
> > > > > > > > For clarity, we modify our contribution as follows: "This is the first work to **construct a test benchmark** for quantitatively measuring the visual perception alignment between humans and AI **across diverse scenarios (8 categories in total).**" (found on line 66 page 3 in the revised paper). You might still have reservations about the use of the word "first," but after thoroughly researching existing works, we found that there were no previous works that measured visual perception alignment exhaustively across diverse categories using appropriate measurement. If you still find issues with our revised claim, please feel free to provide your feedback.
> > > > > > > >
> > > > > > > > We apologize if there was confusion for not mentioning the exact lines. The revisions were made multiple times which shifted the line numbers. We made sure the line numbers are correct in this response.
> > > > > > > >
> > > > > > > > Thank you again for your feedback.

---

> > > > > > > > > ### Author Response · Authors · 2023-08-28
> > > > > > > > > **Response to Reviewer ChH6 (2/2)**
> > > > > > > > >
> > > > > > > > > **References**
> > > > > > > > >
> > > > > > > > > [1] Peterson, Joshua C., et al. "Human uncertainty makes classification more robust." Proceedings of the IEEE/CVF International Conference on Computer Vision. 2019.
> > > > > > > > >
> > > > > > > > > [2] Schmarje, Lars, et al. "Is one annotation enough?-A data-centric image classification benchmark for noisy and ambiguous label estimation." Advances in Neural Information Processing Systems 35 (2022): 33215-33232.
> > > > > > > > >
> > > > > > > > > [3] Putting visual object recognition in context, CVPR, 2020
> > > > > > > > >
> > > > > > > > > [4] When Pigs Fly: Contextual Reasoning in Synthetic and Natural Scenes, ICCV, 2021

---

> > > > > > > > > > ### Comment · Reviewer_ChH6 · 2023-08-29
> > > > > > > > > > **Acknowledgement of reading the authors' responses**
> > > > > > > > > >
> > > > > > > > > > Thank you for the detailed comparisons between the current and existing works. I appreciate the contributions of the paper. The claim on "first" is more reasonable than the initial version. The dataset is attractive to the community in studying visual perception alignments.
> > > > > > > > > >
> > > > > > > > > > Please ensure you incorporate the expanded discussion with the relevant works here in the appendix (due to the page limit) in the final version.
> > > > > > > > > >
> > > > > > > > > > I have increased the final score to 7.

---

> > > > > > > > > > > ### Author Response · Authors · 2023-08-29
> > > > > > > > > > > **Response to Reviewer ChH6**
> > > > > > > > > > >
> > > > > > > > > > > We sincerely are grateful that the reviewer recognized the novelty and contribution of our paper and found our dataset to be valuable for the community studying visual perception alignment. Your feedback indeed helped improve our work, and it was a pleasure to engage in multiple rounds of discussion about our paper.
> > > > > > > > > > >
> > > > > > > > > > > As you suggested, we added our previous response into both the related works section (line 102, page 3) and Appendix J.
> > > > > > > > > > >
> > > > > > > > > > > Thank you again for your time and valuable comments. We are happy to have had you as a reviewer.

---

### Official Review · Reviewer_82Bt · 2023-07-07
**One dataset for measuring the degree of alignment**

**Rating:** 7
**Confidence:** 5
**Correctness:** See issues from improvement possiblit…
**Clarity:** The paper is well written.

**Strengths:**

- The paper is well written and good to understand
- The motivation and experiments are clear
- The definition of the categories is well defined and covers many different perpsectives while sucessfully circumventing the need for crawling possibly copyrighted images from the web.
- the idea to use Hellinger distance instead of KL divergence is a nice new idea and is well motivated
- The assement of uncertainty with MTurk is interesting and a valueable addition

**Additional Feedback:**

Nothing

**Documentation:**

Documentation seems to be sufficient

**Limitations:**

The limitations are reported to be discussed. However, I do not know where and even the word "limit" does not occur in the pdf.

**Opportunities For Improvement:**

- The statement "the authors are the first to comprehensivly explore visual perception between models and humans" is an exaggeration or in my opinion just wrong. As they state themselves in line 79, Peterson et al did the same before. Additionally, a missing reference Schmarje et al. 2022 Is one annotation enough? A data-centric image classification benchmark for noisy and ambiguous label estimation, did the same but on 10 different datasets including higher resolution images. Especially, the work of Schmarje et al. leads to very similar insights than this work and thus has to be covered. I'm confident that the authors just did not know about the work but it clearly contradicts some statements of this work which has to be addressed.
- The class Uncertain only depicts corrupted or cropped images. It does not cover tasks of subjective interpretation such as Mazeika et al 2022, How Would The Viewer Feel? Estimating Wellbeing From Video Scenarios. Corruption is a common issue and they investigate multiple forms but the inclusion of other source would clearly improve this evaluation.
- I'm uncertain about the usefulness of the realibility score. I understand the motivation of the score and find it interesting. However, why don't you just report the percentage of reliable (+1 in Table 2), maybe the other values as well. The scores of -8509 and -23365 are in my opinion more difficult to interpret than e.g. 89% reliablity vs. 95% reliability.
- You aknowledge Uncertainty as an issue but have taken the image from 'imagenet' which suffers from this issue. See for example image 'n02391373_6753.JPEG' from the test set. Is the groundtruth label here zebra or human. The man is standing above the zebra and both mammals use up about similar space in the image. 'n01322983_6266.JPEG' depicts an image montage of an ice bear in a newspaper. Is this image really photorealisitic and thus different enough to your category 7? It seems like the data was not clearly checked. These cases should most likely also be n the class 'uncertain'. The examples are just two I found after a fast look through the data, there are most likely many more which are not prevented systematically.
- Some changes in category 3, lead to a change in class or make them uncertain `ILSVRC2012_val_00032762-bear-vgg16-1`from open-test category is just fur with four legs. I would agree most likely it is a bear but the introduced uncertainty here is not covered.
- I could not answer the question why anyone should use this proposed dataset over the previously discussed works in table 1 and the mentioned work above. I find the topic important but this dataset and evaluation seems to combine just previously defined parts. The most important part is the combination of the class defintiion "must-act" and "must-abstain" with uncertainty. However, if you take the data from CIFAR10H and use the given probability distribution you can easily define such classes as well. I'm certain the authors have a valid argument why their work is different and novel in comparison to previous work but I can't see it clearly at the moment.


*EDIT:* Please see the full discussion below for details

**Relation To Prior Work:**

See issues in improvement possiblites.

**Summary And Contributions:**

The authors cover a very important topic the alignment betwenn model and human.
The goal is to create a dataset which can be used to evaluate the reliability of the model with regard to three cases, must-act (sure it is correct), must-abstain (sure it is wrong), uncertain (not sure).
The categories must-act and must-abstain were discussed in previous publications (line 49) but Uncertain is often not considered.
The authors claim that they are the first to investigate this topic.

---

> ### Author Response · Authors · 2023-08-24
> **Response to Reviewer 82Bt (1/4)**
>
> Thank you for the valuable comments and suggestions. Please kindly find the responses below.
>
> **Q. The statement "the authors are the first to comprehensively explore visual perception ALIGNMENT between models and humans" is an exaggeration or in my opinion just wrong. As they state themselves in line 79, Peterson et al did the same before. Additionally, a missing reference Schmarje et al. 2022 Is one annotation enough? A data-centric image classification benchmark for noisy and ambiguous label estimation, did the same but on 10 different datasets including higher resolution images. Especially, the work of Schmarje et al. leads to very similar insights than this work and thus has to be covered. I'm confident that the authors just did not know about the work but it clearly contradicts some statements of this work which has to be addressed.**
>
> Thank you for your comment. We stated “to be the first to comprehensively explore visual perception alignment with models and humans” as we are the first to evaluate visual perception alignment, not the first to evaluate visual perception itself, between models and humans over 8 categories, aiming to cover as diverse situations as possible. Although Peterson et al. [1] and Schmarje et al. [2] discuss similar topics about uncertainty in datasets, neither work focuses on visual alignment. Peterson et al. [1] demonstrated that training models on soft labels improves the model’s generalization and robustness. Schmarje et al. [2] explored uncertainty of data from a data-centric perspective by training models with soft labels. Although both works utilize soft labels collected from multiple annotations, their main research focuses are not on visual alignment.
>
> Our paper evaluates visual perception alignment across diverse scenarios, categorized in 8 different categories. In order to evaluate the visual perception alignment with humans, the dataset must 1) cover diverse scenarios and 2) include gold human labels that reflect the general population. From this perspective, datasets in Peterson et al. [1] and Schmarje et al. [2] cannot be directly utilized in our study, since Peterson et al. [1] only handle ambiguity coming from low resolution images and Schmarje et al. mostly focus on clear and uncorrupted images. Additionally, both datasets have average annotations of 50 and 35, respectively, which do not represent the general population and cannot provide an accurate statistical error bound. However, in our dataset, we applied 15 corruption types with 10 different severity levels, ranging from clear images to entirely corrupted images. We collected 134 annotations per image to reflect the general population within a 5% error bound. Thank you for introducing the related works, we added them in the related works in line 94 page 3 in the revised paper.
>
> **Q. The class Uncertain only depicts corrupted or cropped images. It does not cover tasks of subjective interpretation such as Mazeika et al 2022, How Would The Viewer Feel? Estimating Wellbeing From Video Scenarios. Corruption is a common issue and they investigate multiple forms but the inclusion of other source would clearly improve this evaluation.**
>
> The primary focus of our paper is visual alignment in the context of object recognition. Therefore, we limited the scope of our investigation to specific uncertainties that can influence object recognition, namely corruption and cropping. While we acknowledge that other sources of uncertainty might impact object recognition, our study concentrated on the corruption types drawn from well-established previous research (ImageNet-C [3]). During this process, we intentionally avoided subjective interpretations which are not directly related to object recognition.
>
> As for other sources of uncertainty that might impact object recognition, humans might be influenced by their personality (especially conscientiousness), the mood of the day, cultural background, eyesight (this is partially incorporated by blurred corruption), etc., but such factors are unlikely to affect AI’s visual perception. And furthermore, such uncertainty factors are difficult to control (e.g. we would have to hire a wide range of humans with varying eyesights), and therefore difficult to use as one of the aspects of our carefully controlled alignment dataset. In fact, we designed the MTurk survey to remove unaddressed confounders such as cultural background or eyesight as much as possible, by setting the recognition target as universally known mammals, including control samples to detect low-quality annotators, and measuring intra & inter-annotator consistencies.
> Thank you for introducing Mazeika et al. [4], we have included it in the related work of the revised paper on line 87 page 3.

---

> > ### Author Response · Authors · 2023-08-24
> > **Response to Reviewer 82Bt (2/4)**
> >
> > **Q. I'm uncertain about the usefulness of the reliability score. I understand the motivation of the score and find it interesting. However, why don't you just report the percentage of reliable (+1 in Table 2), maybe the other values as well. The scores of -8509 and -23365 are in my opinion more difficult to interpret than e.g. 89% reliability vs. 95% reliability.**
> >
> > First of all, let us re-establish how we re-purposed our dataset to calculate the reliability scores, so that you don’t have to re-read the manuscript. For the Must-Act samples, the best-case scenario is the model always making the right classification. If the model, however, abstained rather than misclassified, we consider the model “reliable”. Therefore we assigned 0 for Abstention and -c for Incorrect Prediction. For the Must-Abstain samples, the correct action is to abstain (hence +1), and all other actions are “unreliable” (hence -c).
> > For the Uncertain samples, we divided them into two groups just for the sake of calculating the reliability score, namely “Must-Act among Uncertain” and “Must-Abstain among Uncertain”. The former group consists of the Uncertain samples for which more than 50% (i.e. the \lambda in the manuscript) of the MTurkers chose the pre-corruption label. The latter group is the other way around. For the former group, the score assignment is the same as the Must-Act samples, because we treat this group as having a ground-truth label (again, this is just an assumption we use for calculating the reliability score). For the latter group, the score assignment is the same as the Must-Abstain samples, except that the model receives 0 points for predicting the pre-corruption label, because we consider it more “reliable” for a model to manifest potentially super-human visual perception than making an incorrect prediction. (Note that it is debatable whether being able to predict the pre-corruption label is a mark of super-human visual perception, because when the image is sufficiently corrupted, it is unlikely there is any recognizable signal. Again, this super-human visual perception is just an assumption we use for calculating the reliability score.)
> >
> > Given this scoring system, if we were to report only the percentage of reliable instances (+1 in Table 2) as suggested, then abstention and misclassification in the Must-Act group would be treated equally, where the score becomes a mere reflection of the model’s classification accuracy. Therefore, a distinct score is needed to differentiate between abstention and misclassification in the Must-Act group. The same logic applies to the Must-Abstain group, specifically for the “Must-Abstain among Uncertain” samples, where we assumed that it is more “reliable” for the model to predict the pre-corruption label than predicting other animal classes.
> >
> > Additionally, setting different values for the cost c depending on the use case is another essential consideration factor. As mentioned in the revised version in line 313 page 8, in the most extreme scenario, the cost c would be equal to the total number of samples in the test set. This would mean that even a single mistake would result in a negative final score, and abstaining from all decisions on Must-Act samples would be deemed more reliable than making even one incorrect prediction. In reality, AI models for medical applications could be evaluated in such an extremely conservative setting.
> >
> > **Q. You acknowledge Uncertainty as an issue but have taken the image from 'imagenet' which suffers from this issue. See for example image 'n02391373_6753.JPEG' from the test set. Is the groundtruth label here zebra or human. The man is standing above the zebra and both mammals use up about similar space in the image. 'n01322983_6266.JPEG' depicts an image montage of an ice bear in a newspaper. Is this image really photorealisitic and thus different enough to your category 7? It seems like the data was not clearly checked. These cases should most likely also be in the class 'uncertain'. The examples are just two I found after a fast look through the data, there are most likely many more which are not prevented systematically.**
> >
> > We are also well aware of the presence of multiple objects and non-photorealistic images in ImageNet. Therefore, we did not use all the images from the ImageNet test set; instead, we underwent an image filtering process to select images that 1) contain only one object per image, and 2) ensure photo-realism. You will gain an understanding of this process by observing Category 1 samples. We detail this information in the main paper line 152 on page 4: “To avoid misclassifications due to background objects, all samples exclusively contain one object. The authors manually scrutinized all test samples to ensure this.”. Additionally, on page 5 line 208, we state that “Considering that MUST-ACT samples are photo-realistic images confirmed by humans”.

---

> > > ### Author Response · Authors · 2023-08-24
> > > **Response to Reviewer 82Bt (3/4)**
> > >
> > > **Q. Some changes in category 3, lead to a change in class or make them uncertain ILSVRC2012_val_00032762-bear-vgg16-1from open-test category is just fur with four legs. I would agree most likely it is a bear but the introduced uncertainty here is not covered.**
> > >
> > > We want to note that the authors individually reviewed all the samples, including this particular image. In our assessment, we recognized the image as depicting a bear, with its mouth visible in the lower left corner. However, we acknowledge that depending on the perspective, it might appear to some as a foot rather than a mouth. While there may be some uncertainty in this specific image, we have once again verified that the objects in other images are clearly recognizable. Despite our meticulous efforts, we recognize that claiming out dataset to be perfectly flawless could be challenging. However, no human-made test is perfect, and we emphasize again that the authors have put painstaking effort into constructing this dataset, and we consider it to be as high-quality as any other widely used dataset in the AI community.
> > >
> > > **Q. I could not answer the question why anyone should use this proposed dataset over the previously discussed works in table 1 and the mentioned work above. I find the topic important but this dataset and evaluation seems to combine just previously defined parts. The most important part is the combination of the class defintiion "must-act" and "must-abstain" with uncertainty. However, if you take the data from CIFAR10H and use the given probability distribution you can easily define such classes as well.**
> > >
> > > We first want to emphasize that our dataset is constructed with a completely different objective than existing datasets. We established four requirements that a dataset should meet to serve as a universal visual alignment benchmark. Although not mentioned explicitly in the original submission, the arguments in Section 3 follow these four requirements (the respective arguments’ locations in the original submission are mentioned at the end of each requirement)
> > > 1) Clear definition for each class: Each class must have a clear definition so that model developers can easily prepare their models and training strategies. An example where this is not satisfied can be found in existing datasets like CIFAR-10, where the definition between ‘automobile’ and ‘truck’ is not perfectly distinct and can vary by individual. After careful consideration, we chose the taxonomic classification in biology, a meticulous product of expert effort to hierarchically distinguish species. In lines 96-100, we mention that strict class definitions are critical.
> > > 2) Class familiarity to average individuals: Since the Uncertain samples are constructed from an Amazon Mechanical Turk survey with 134 people per sample, classes should be recognizable without requiring specialized knowledge. (We mention the importance of familiarity in lines 100-105.)
> > > 3) Coverage of diverse and realistic scenarios: This is vital for accurate visual alignment assessment. This is the reason we have a total of eight categories, as mentioned in lines 112-113.
> > > 4) Ground truth labels for each sample: This is critical as we need human visual perception labels for all samples to calculate visual perception alignment. We explain why accurately estimating the ground truth distribution, as we did using 134 labels per sample, is important in Section 3.3.
> > >
> > > We listed these requirements at the beginning of Section 3 in the revised paper. Despite their importance, existing datasets fail to meet at least one of these requirements, as shown in Table 1 on page 7 of the revision paper.
> > >
> > > Another distinguishing feature of our dataset is the collection of diverse “Uncertain” samples and their gold human labels. While there are datasets with corruptions (ImageNet-C [3], Park et al. [5]), those datasets lack gold human labels or do not cover diverse situations. CIFAR10-H is also unqualified for our task since it only covers ambiguity coming from low resolution and it is difficult to claim that their labels reflect the general population due to the small number of annotations. In contrast, our dataset consists of diverse images ranging from clear images to completely black images, and we surveyed 134 people per image to achieve an error bound of less than 5% to reflect the general population gold human perception label.
> > >
> > > In conclusion, existing datasets are unqualified for quantitatively measuring visual alignment. The authors put significant effort into building a benchmark that employs carefully chosen classification targets across 8 different settings (even if you try to simply combine multiple datasets, carefully harmonizing the classes alone is a challenging task already) with every sample having ground-truth human labels, thus meeting the above mentioned four requirements.

---

> > > > ### Author Response · Authors · 2023-08-24
> > > > **Response to Reviewer 82Bt (4/4)**
> > > >
> > > > **Q. The limitations are reported to be discussed. However, I do not know where and even the word "limit" does not occur in the pdf.
> > > > Thank you for your comment.**
> > > >
> > > > In the initial manuscript, due to the space constraint, we only touched upon the limitation of our dataset in regards to not including potentially contentious topics such as gender or racial bias. Here we provide a more detailed description of our work’s limitations.
> > > >
> > > > One limitation of our work is that our dataset includes only 10 classes. This constraint arose because we had to collect multiple annotations from 134 people per image, and we needed to select classes that the average person would find familiar and not require specialized knowledge. The method of manually collecting gold human labels as used in our study cannot be easily applied in all domains. For instance, diagnosing chest X-rays typically involves identifying 14 diseases (e.g. CheXpert dataset uses 14 labels). To collect the ground truth labels within a statistical error bound of 5%, one would need to consult at least 107 radiologists. This approach is much less practical and requires more resources, especially since annotations from radiologists are more costly than those from average people. Therefore, more creative solutions are needed to acquire gold human labels in specialized domains.
> > > >
> > > > Another limitation is that we have chosen only 16 types of corruption in the Uncertain group and the degree of corruption in our dataset is discrete. In contrast, in the real world,  the types of corruption are endless and the degrees of corruption are continuous. However, this limitation could be mitigated by including more corruption types, combining different corruptions, and applying the corruptions at various continuous degrees.
> > > >
> > > > We have included these insights in the conclusion of our revision.
> > > >
> > > > **References**
> > > >
> > > > [1] Peterson, Joshua C., et al. "Human uncertainty makes classification more robust." Proceedings of the IEEE/CVF International Conference on Computer Vision. 2019.
> > > >
> > > > [2] Schmarje, Lars, et al. "Is one annotation enough?-A data-centric image classification benchmark for noisy and ambiguous label estimation." Advances in Neural Information Processing Systems 35 (2022): 33215-33232.
> > > >
> > > > [3] Hendrycks, Dan, and Thomas Dietterich. "Benchmarking neural network robustness to common corruptions and perturbations." arXiv preprint arXiv:1903.12261 (2019).
> > > >
> > > > [4] Mazeika, Mantas, et al. "How Would The Viewer Feel? Estimating Wellbeing From Video Scenarios." Advances in Neural Information Processing Systems 35 (2022): 18571-18585.
> > > >
> > > > [5] Park, Jeonghoon, et al. "Natural attribute-based shift detection." arXiv preprint arXiv:2110.09276 (2021).

---

> > > > > ### Comment · Reviewer_82Bt · 2023-08-25
> > > > > **Reply (4/4)**
> > > > >
> > > > > - Limitations:
> > > > >
> > > > > "In the initial manuscript, due to the space constraint, we only touched upon the limitation of our dataset in regards to not including potentially contentious topics such as gender or racial bias. Here we provide a more detailed description of our work’s limitations." For me this reads like we knew we did not cover it enough but now we have one page more and can add it. If this is the case this is not good style. The Limitations are okay but could be enhance with the discussed issues above.

---

> > > > ### Comment · Reviewer_82Bt · 2023-08-25
> > > > **Reply (3/4)**
> > > >
> > > > - Some changes in category 3, lead to a change in class or make them uncertain
> > > >
> > > > I agree that it is most likely a bear and that uncertainty is introduced here. You clearly seperate corruptions from category 3 to uncertainty. You either have to acknowledge this issue or provide multiple annotations (measure uncertainty) of these examples as well.
> > > > The issue with "high-quality as any other widely used dataset in the AI community" is that ImageNet is widely used but that it flaws are openly discussed (e.g. https://arxiv.org/abs/2006.07159 https://arxiv.org/abs/2205.04596). Thus the quality of current datasets is not good enough and have to create higher quality data then before. So be as good as accepted flawed datasets is not good enough. Of course your dataset can not be perfect but my job is to point out all issues I can find.
> > > >
> > > > - . I could not answer the question why anyone should use this proposed dataset over the previously discussed works
> > > >
> > > > I like the addition of your four criterias because it allows a direct evaluation of what you hope to achieve.
> > > >
> > > > My main issue ist with the disharmony of goal 1 vs. 3: You want to be realistic but want clear class definitions. As seen in the discussed literature above this is often just not the case. You yourself acknowledge in the limitations "in the real world, the types of corruption are endless and the degrees of corruption are continuous." I would add that uncertainty (corruption is only one form of it) is continous. In my opinion you can either a distinct class definitions and ignore all corner cases or you accept the continous regions between. You try to put the continous regions in between into the class uncertainty. I understand this approach but as disscussed above this results in corner cases (corrupted image of category 3) which should also be considered uncertain.
> > > >
> > > > I would say Peterson et al or Schmarje et al also cover all you set goals. The main limitations is here that fewer annotations are provided per image. You are correct your approximation is more numerical realiable. However, they do not use snythetic corruptions to introduce uncertainty, the have more different datasets and classes, they dont run into the issue of when to seperate uncertain and the rest.
> > > >
> > > > In conclusion, as stated above this research is similar to yours. You have some benefits they have some benefits. Thus, I believe both elements can exist in the research community but I see no benefit of your appraoch above the previous ones. If this conclusion is correct the proposed gain in your paper is minimal and thus questionable if it should be accepted to a high-quality conference like NeurIPS.

---

> > > ### Comment · Reviewer_82Bt · 2023-08-25
> > > **Reply (2/4)**
> > >
> > > Reliability score:
> > >
> > > I understand the motivation now better. You want to decicde between misclassification and abstention should be treated differently.My point is that the score are really difficult to understand. They are not really well interpretable. I showed these numbers and their explanation to multiple people and all had difficulty to interpret them. Due the variable c which might even be different between datasets the numbers have no fixed reference point. I would prefer a table which states just all the cases you describe and their total/relative number. I general I prefer a unified score but in this example I have the feeling you are loosing to much information.
> > >
> > >
> > > Uncertain example images:
> > >
> > > The provided examples are from your dataset. I understand that you want to achieve " 1) contain only one object per image, and 2) ensure photo-realism" but both examples illustrate cases where this is not the case. (I stated above these images are from the test set, now they are in the training set. I don't if I misspelled above or if you changed there location).
> > > Take image "elephant14.jpg" from the test set. This is image is not complety photorealistic and thus violates your own goal.

---

> > ### Comment · Reviewer_82Bt · 2023-08-25
> > **Reply (1/4)**
> >
> > - The statement "the authors are the first to comprehensively explore visual perception ALIGNMENT between models and humans" is an exaggeration or in my opinion just wrong.
> >
> > I totally disagree with your interpretation of the provided work. Their main goal is using soft labels to quantify human perception and then train models which are similar to these scores. Thus, they create visual alignment.
> >
> > The difference is my opinion the used approach soft labels vs. Must act and must abstain class. I believe both topics are equally valid approaches to cover visual alignment which are both valid.
> >
> > Your setup has more annotations per image and more synthethic corruptions but less datasets and in general a different setup. I see some positive (more annotations) and some negative parts (synthetic corruptions ). I dont want you to change your approach but I believe you have to acknowledge the previous research in slightly different approaches. You have to remove the above described statement or I would say you are reporting misinformation.
> >
> >
> > - The class Uncertain only depicts corrupted or cropped images
> >
> > " During this process, we intentionally avoided subjective interpretations which are not directly related to object recognition." I see that but I question its validity. Peterson et al showed that people can not differentiate between cats and dogs in some images. If you want to have realitistic uncertainty this approach should be considered. Just creating synthetic corruptions is not realistic. I know that synthetic corruptions allow a better evaluation in a scientific setup and thus I do accept their usage but you are loosing on the realism of your work. Thus I repeat "Corruption is a common issue and they investigate multiple forms but the inclusion of other source would clearly improve this evaluation." Images often include already uncertainty see examples below which you ignore instead of covering. I would say this should be in addition and not a replacement.

---

> ### Comment · Reviewer_82Bt · 2023-08-25
> **Reason for downgrading 5 to 4**
>
> I commented on all replies and have to conclude that my impression of the work degraded. I saw issues in my initial review with the work but were positive that these could be argued away or with some additions could be minized. After the replies now I have the impression the root of my issues are not addressed or maybe even not recognized. I'm happy to discuss this issues further with you and hope that you can improve your work but for the moment the hoped substantial improvement did not happen. I will lower my score accordingly.
>
> I will improve or degrade my score further based on the replies in the remaining discussion time. No replies will be interpreted as accepting my issues and this lead to further downgrading. I sincerly hope that substantial improvements can be given in the remaining time. This was my favourite work with the highest potential of all reviewed work in this year.

---

> > ### Author Response · Authors · 2023-08-26
> > **Response to Reviewer 82Bt (1/5)**
> >
> > We are sincerely grateful that you thoroughly read our responses, and spent your time to engage with us via another turn of discussion (which happens very rarely in paper reviews). We have discussed your opinions to carefully come up with new responses. We also thank you for recognizing the potential of our work. However, we noticed that most of your comments are focused mainly on one aspect of VisAlign, namely Category 8, while our dataset encompasses 8 total categories. We politely ask that you give our work a fair assessment by considering its distinct goal (VisAlign is constructed to serve as a universal test for any AI models to measure their visual perception alignment), construction principle (VisAlign was built upon carefully chosen classes, sampling theory and statistical theory), and coverage (VisAlign tests AI in terms of total 8 categories). Please find our individual responses below.
> >
> > **Q. I understand the motivation now better. You want to decide between misclassification and abstention should be treated differently. My point is that the score are really difficult to understand. They are not really well interpretable. I showed these numbers and their explanation to multiple people and all had difficulty to interpret them. Due the variable c which might even be different between datasets the numbers have no fixed reference point. I would prefer a table which states just all the cases you describe and their total/relative number. I general I prefer a unified score but in this example I have the feeling you are loosing to much information.**
> >
> > Thank you for your suggestion. We especially thank you for gathering different opinions from your colleagues on the interpretability of our reliability score (we are excited that at least someone considered our work valuable enough to show it to their colleagues). Following your suggestion, we provided a separate table with the values for each case separately in the supplementary material, which can be found in Appendix G.
> >
> > By using our metric, as you mentioned, we can assess the reliability of a given model with a single value. Although the cost 'c' varies for different tasks, making it challenging for us to arbitrarily determine its value, this very aspect actually makes our proposed metric applicable in a versatile manner. In real-world scenarios, users can set the value of 'c' and follow our calculation method to choose the model with the highest score. The variable 'c' can be seen as the "strictness criterion for a reliable model" and can also be interpreted as "how many misclassifications correspond to a single accurate classification." This 'c' value can be set as an integer ranging from 0 to the total size of the test set. A value 0 for 'c' implies a 0% strictness, while the maximum value of 'c' implies a 100% strictness. This metric is also used in other research papers such as Whitehead et al [1].
> >
> > We meticulously designed this metric to enable both absolute and relative reference points. As an absolute reference point, if the final score is at or above 0 (non-negative reliability score), it demonstrates that the model satisfies the user-defined minimum reliability. A relative reference point is between different models; a model with a higher score between two reliability scores is more reliable. Bringing back the example in your original question (-8509 and -23365 from results of R_{450}, which indicates 50% strictness), we can interpret the scores of -8509 and -23365 in two ways: 1) Both are negative numbers far from 0, implying that both models are not reliable at all, and 2) the former can be considered relatively "more reliable" than the latter.
> >
> > **Q. I understand that you want to achieve " 1) contain only one object per image, and 2) ensure photo-realism" but both examples illustrate cases where this is not the case. (I stated above these images are from the test set, now they are in the training set. I don't if I misspelled above or if you changed there location). Take image "elephant14.jpg" from the test set. This is image is not completely photorealistic and thus violates your own goal.**
> >
> > We have not made changes to the dataset during the first revision; the images you mentioned are indeed from the train set. We did not scrutinize the train set thoroughly since the main contribution of this work lies in the test set and we encourage researchers to freely collect their own train samples.
> >
> > Images that we defined as not photo-realistic are those that are clearly different from a real animal's form, such as cartoons or sketches, as can be seen in images of Category 7. In “elephant14.jpg” the target object, which is the elephant, is shown with its photorealistic aspects preserved despite the graphics and the letters that were edited onto the image. Nevertheless, we will replace this image with a more strictly photorealistic one if the text in the image causes concern for you. (Please do let us know if this was the source of your concern)

---

> > > ### Author Response · Authors · 2023-08-26
> > > **Response to Reviewer 82Bt (2/5)**
> > >
> > > **Q3. The statement "the authors are the first to comprehensively explore visual perception ALIGNMENT between models and humans" is an exaggeration or in my opinion just wrong. I totally disagree with your interpretation of the provided work. Their main goal is using soft labels to quantify human perception and then train models which are similar to these scores. Thus, they create visual alignment.
> > > The difference is my opinion the used approach soft labels vs. Must act and must abstain class. I believe both topics are equally valid approaches to cover visual alignment which are both valid. Your setup has more annotations per image and more synthethic corruptions but less datasets and in general a different setup. I see some positive (more annotations) and some negative parts (synthetic corruptions ). I dont want you to change your approach but I believe you have to acknowledge the previous research in slightly different approaches. You have to remove the above described statement or I would say you are reporting misinformation.**
> > >
> > > As previous works train the model with the goal of human visual perception, it could be seen that there is some overlap between them and our work. However, we used the phrase “first to” since our main focus is not on training the model, but on creating a rigorous test (similar to tests that humans take such as college entrance exams) to quantitatively measure the visual perception gap between the model and humans across various categories. i.e. Our main interest does not lie in training but on rigorously testing the visual perception alignment. For that purpose, a dataset should satisfy the four requirements we mentioned above and use a proper metric that reflects the visual perception alignment. The previous works you mentioned utilized the datasets mainly for training and did not thoroughly verify whether the model actually achieved visual alignment. Peterson et al. only test their models on in-class samples (in our case, Category 1) and out-of-class samples (in our case, Category 4 and Category 6) and they showed only accuracy and cross entropy (which is analogous to KL divergence) of the models. Therefore, they did not test their models on various possible scenarios and did not use proper measurement, as KL divergence is not an optimal choice for visual perception alignment as we described in line 282 page 7 in our paper. Schmarje et al. et al also only tested their models on in-class samples (in our case, Category 1) and showed accuracy and KL divergence. Therefore, although previous works trained their models with the goal of achieving visual perception alignment, none of the works have thoroughly verified how much the models have actually achieved visual perception alignment under diverse situations with an appropriate measurement. In contrast, we quantitatively measured visual perception alignment across various scenarios with multiple human annotations on uncertain images. In addition, we borrowed Hellinger distance to precisely calculate the visual perception alignment after careful consideration of other distance-based metrics such as KL divergence and Total Variation distance.
> > >
> > > We acknowledge your concern; therefore, we modify our contribution as follows: "This is the first work **to construct a test benchmark for quantitatively measuring** the visual perception alignment between humans and AI **across diverse scenarios (8 categories in total)**." (found on line 75 page 3 in the revised paper). You might still have reservations about the use of the word "first," but after thoroughly researching existing works, we found that there were no previous works that measured visual perception alignment exhaustively across diverse categories using appropriate measurement. If you still find issues with our revised claim, please feel free to provide your feedback.

---

> > > > ### Author Response · Authors · 2023-08-26
> > > > **Response to Reviewer 82Bt (3/5)**
> > > >
> > > > **Q. The class Uncertain only depicts corrupted or cropped images. Peterson et al showed that people can not differentiate between cats and dogs in some images. If you want to have realitistic uncertainty this approach should be considered. Just creating synthetic corruptions is not realistic. Thus I repeat "Corruption is a common issue and they investigate multiple forms but the inclusion of other source would clearly improve this evaluation." Images often include already uncertainty see examples below which you ignore instead of covering. I would say this should be in addition and not a replacement.**
> > > >
> > > > In the original question, you mentioned "It does not cover tasks of subjective interpretation such as Mazeika et al 2022", which led us to assume that you were referring to the subjective interpretation of emotions. We responded based on this understanding.
> > > >
> > > > We agree with you that synthetically corrupting images is not likely to cover all realistic uncertainties. However, what is realistic uncertainty? You mentioned some people not being able to differentiate between cats and dogs. But that is inter-class uncertainty, and probably only one type of “realistic uncertainties” (We say probably, because it is difficult to even define “realistic uncertainty”). If you were specifically looking for inter-class uncertainty, then we are glad to tell you that a certain portion of Category 8 samples are manifesting inter-class uncertainty; for 8.5% of the Category 8 samples, the 134 MTurk survey participants (per sample) were struggling with differentiating between two or more animals due to the uncertainty (rather than being confused between one animal and abstention). Furthermore, many types of corruption we use are “realistic” in their own way. For example, adjusting the brightness of the image is certainly realistic, and changing its resolution is similar to viewing an object beyond a filter (e.g. semi-transparent glass), and weather changes are also certainly realistic.
> > > >
> > > > We hope this fact alleviates your concern regarding our dataset only consisting of “synthetic” uncertainties and not incorporating inter-class uncertainties. However, as we said above, “realistic uncertainties” is too broad to specify and difficult to collect or generate, and hence for now we use corruptions (but sufficiently diverse types of corruption), and we will add this as a limitation. If there are another specific type of “realistic uncertainties” you think we must add to our dataset in order to make it convincing to the research community, then please feel free to let us know. We will strive to add as long as time allows. (After all, we want to make a rigorous visual alignment test for AI that is adoptable by as many people/institutions as possible, at least in theory)
> > > >
> > > > **Q. Some changes in category 3, lead to a change in class or make them uncertain. I agree that it is most likely a bear and that uncertainty is introduced here. You clearly seperate corruptions from category 3 to uncertainty. You either have to acknowledge this issue or provide multiple annotations (measure uncertainty) of these examples as well.**
> > > >
> > > > We understand your concern regarding the uncertainty within this particular image. We agree that we must strive to minimize any errors or controversies regarding the quality of the dataset, and therefore we replaced the said bear image with a more clear one. Please find the revised dataset and let us know if you still see the same problem.

---

> > > > > ### Author Response · Authors · 2023-08-26
> > > > > **Response to Reviewer 82Bt (4/5)**
> > > > >
> > > > > **Q. I could not answer the question why anyone should use this proposed dataset over the previously discussed works (1/2)**
> > > > >
> > > > > We sincerely agree with your concern regarding the continuity of uncertainty and the challenge of drawing clear distinctions among classes (i.e., as we mentioned in our paper that it is hard to distinguish between “car” and “truck”) and uncertain images and the rest. However, our goal is to present a universal testing benchmark that quantitatively measures visual perception alignment between models and humans. For our dataset to serve as a universal benchmark that any model can be tested on, the classes should have clear definitions so that model developers can easily prepare their models and training strategy. Therefore, after careful consideration, we used the taxonomic classification in biology which is the meticulous product of decades of effort by countless domain experts to hierarchically distinguish each species as accurately as possible with clear definition. Also, in order to comprehensively measure the visual perception alignment between models and humans, the models should be tested under various conditions including clear in-class images (Must-Act), clear out-of-class images (Must-Abstain) and confusing images (Uncertain). Again, we agree with your concern that there is no clear boundary between clear images and uncertain images. The best scenario would be to survey all images in our dataset to 134 people per image to obtain numerically reliable annotations. However, it requires a tremendous amount of time and money considering that there are 900 images in our dataset. Therefore, due to the realistic reasons, the authors put significant effort to include only clear images in Must-Act and Must-Abstain and obtained human annotations on Uncertain images. As for the corrupted image of Category 3, we have replaced the problematic bear image with a more clear one, so we believe this corner case is no longer a problem. Likewise, we had put our best effort to construct our dataset to encompass diverse scenarios and serve as a benchmark. We sincerely agree with your concern that uncertainty is continuous and there is always the possibility of corner cases. But please note that the authors also considered the points, scrutinized the dataset to remove possible corner cases and put the best effort to construct high-quality benchmark dataset under the realistic constraints.
> > > > >
> > > > > We want to emphasize three key differences that distinguish our uncertain category (i.e. Category 8) from prior works that also focus on uncertainty in object recognition. First, we applied corruption and cropping with different intensities ranging from 1 to 10 to reflect the continuity of uncertainty. As you mentioned, uncertainty is continuous and it is critical to test models on samples where uncertainty may increase in stages. In this sense, we tested models visual perception alignment on varying degrees of uncertainty. Second, we obtained 134 human annotations per image to accurately estimate the ground truth visual perception distribution. Third, while our uncertain samples include inter-class uncertainty (soft labels distributed only among target classes), we also include recognizability uncertainty (soft labels distributed among classes + abstention), namely whether an image itself is recognizable or not. If an image is moderately brightened (i.e., intermediate phase between a clear image and a complete white image), then the object itself may or may not be recognizable. Visual perception includes not only object identification (predicting that it is an elephant) but also object recognizability (the object itself is recognizable). In this sense, we cover broader scenarios compared to previous works as we include object recognizability uncertainty in our uncertain category.
> > > > >
> > > > > We also want to highlight that our dataset not only contains Uncertain but also contains Must-Act and Must-Abstain to cover as diverse scenarios as possible. In order to evaluate a model’s visual perception alignment, a model should be tested under Must-Act (whether it predicts a correct class with high confidence), Must-Abstain (whether it abstains out-of-class samples with high confidence), and Uncertain (whether it reflects the human uncertainty). However, previous datasets are limited in that they did not cover diverse situations with sufficient number of annotations. Therefore, their dataset cannot be utilized as a universal test benchmark to measure visual perception alignment (it would be unfair to ask them to serve as a test benchmark, since that was not their objective to begin with). It is true that previous papers and our work share some common interests in that both deal with uncertainty, but in our case, uncertain samples are a subset of our final dataset, and we cover as diverse situations as possible.

---

> > > > > > ### Author Response · Authors · 2023-08-26
> > > > > > **Response to Reviewer 82Bt (5/5)**
> > > > > >
> > > > > > **Q. I could not answer the question why anyone should use this proposed dataset over the previously discussed works (2/2)**
> > > > > >
> > > > > > In principle, no two works are exactly the same, and all works have merit (or benefits according to your words) in their own way. And therefore all works can co-exist in the research community (excluding forged/doctored/plagiarized works). So we are on the same page as you here. But we never claimed that our work can replace all previous works nor have we ever claimed that our work is above all previous works. We are simply claiming that we are making a contribution that is distinct from all previous works; “This is the first universal test benchmark constructed based on clear class definition, sampling theory and statistical theory to rigorously measure the visual perception alignment between humans and AI across 8 different categories.” Therefore we respectfully, but wholeheartedly disagree with your conclusion that “the proposed gain is minimal”. Given that our work is truly the first work to construct a rigorous test benchmark to quantitatively measure the visual perception alignment across diverse categories, which was never attempted before, we see no logical reason that NeurIPS is too high-quality a conference for our work to have a place in.
> > > > > >
> > > > > > **Q. "In the initial manuscript, due to the space constraint, we only touched upon the limitation of our dataset in regards to not including potentially contentious topics such as gender or racial bias. Here we provide a more detailed description of our work’s limitations." For me this reads like we knew we did not cover it enough but now we have one page more and can add it. If this is the case this is not good style. The Limitations are okay but could be enhance with the discussed issues above.**
> > > > > >
> > > > > > Thank you for your feedback, we added your concern about continuity and realistic uncertainty. We are thankful to receive such high-quality and in-depth concerns regarding our work. You can find additional limitations in line 410 page 10.
> > > > > >
> > > > > > **References**
> > > > > >
> > > > > > [1] Whitehead, Spencer, et al. "Reliable visual question answering: Abstain rather than answer incorrectly." European Conference on Computer Vision. Cham: Springer Nature Switzerland, 2022.

---

> > > > > > > ### Comment · Reviewer_82Bt · 2023-08-28
> > > > > > > **Reply to (5/5)**
> > > > > > >
> > > > > > > I read this after my reply to 4/5 and agree that different ideas should be accepted. I as reviewer just have to ensure that the quality and gained insights are appropriate for publication. I was sceptically last time but are now more positive after this round of discussion.

---

> > > > > > ### Comment · Reviewer_82Bt · 2023-08-28
> > > > > > **Reply (4/5)**
> > > > > >
> > > > > > Thanks for the nice summary. This really helped me. I would like if you could include this (or a shorter version) in the paper but I can not see where (due to the page limit). If you find a way (maybe in the supplement) it would be helpful also to others.
> > > > > >
> > > > > > I like also to repeat your argument of out of distribution from above. I would urgue to include this argument. I could argue against some of the your difference or how others might do a bit better (e.g. 900 images times about 100 annotations are only 90,000 annotations which is do able even on a low budget). In general, these arguments feel a bit too much like personal taste and thus I would like to exclude them.
> > > > > >
> > > > > > If you rework the discussion around table 1, related work and your requirements based on our discussion, I can drop this issue. I see now that you have some difference and they now only have to be reflected in the work. This might not be easy due to the page limit but I'm certain you can do it and maybe you can shift the more lengthier arguments into the supplement.

---

> > > > > ### Comment · Reviewer_82Bt · 2023-08-28
> > > > > **Reply (3/5)**
> > > > >
> > > > > - We hope this fact alleviates your concern regarding our dataset only consisting of “synthetic” uncertainties and not incorporating inter-class uncertainties.
> > > > >
> > > > > The provided exaplanations reduce the issue but do not alleviate them. I still stand to the argument that including a more diverse uncertainties would improve your work. Due to the fact that we are getting close to the end of the discussion phase it is unrealistic that you add these elements. I wanted to ask for a more extended paragraph in the limitations which you already did. I guess at this point I would prefer more diverse uncertainties but is good enough for the moment.
> > > > >
> > > > > - . Some changes in category 3, lead to a change in class or make them uncertain.
> > > > >
> > > > > Some as above, the question is a more issues remaining in the dataset which I have not found. I'm not totally conviced but would ask of you to check on this issue (and the ones above) again and then I will have to trust you on this point.

---

> > > > ### Comment · Reviewer_82Bt · 2023-08-28
> > > > **Reply to (2/5)**
> > > >
> > > > I appreciate the relaxation of the mentioned statement and can accept it in its current form.
> > > >
> > > > Let me rephrase where I see the difference to Peterson  et al. and Schmarje et al. you create must abstain images especially examples with out of distribution properties such as category 5 and 7.
> > > > More over you use the Hellinger distance which sounds to me more suitable to the task.
> > > >
> > > > What I'm missing is that is currently not well reflected in your work. I would like to propose the following change to your table 1 which would reflect this more.
> > > >  |           | Req 1 |    Req 2   | Req 3  |   Req 4  |   Req  5 |
> > > > | --- | --- | --- | --- | ---|
> > > > | Peterson et al.     | Y    |          Y          |   N      |    Y       |      N |
> > > > | Schmarje et al.   |   Y      |        N     |        Y     |     Y      |      N |
> > > >
> > > > (sorry for the formatting, a table seems not be possible or I'm mising some formation elements)
> > > >
> > > > Req 5. Out of distribution images to test the robustness.
> > > >
> > > > (Please note that CIFAR10H is included in the work of Schmarje et al so (category 4 and 6 should also be included there).)
> > > >
> > > > This is just a proposal and you have to decide how and if you want to include it but the out of distribution argument is the first time that I thought this is the difference to previous work.

---

> > > ### Comment · Reviewer_82Bt · 2023-08-28
> > > **Reply to (1/5)**
> > >
> > > I find the discussion of NeurIPS one of the best review processes and thus find it very important to engage in the discussion.
> > >
> > > I see why it might seem like I'm focusing on category 8 but this is not my attention. I'm questioning the complete creation of your dataset. One major difference to previous work is the existence of the category uncertain (see table 1 of first version). I agree that you should assume something between must act and must abstain but in your work the borders between uncertain and must act (or abstain) seem not well enough defined. However, I will keep this point in mind while reading the rest of the replies.
> > >
> > > - About the metric:
> > >
> > > I appreciate the added table and the explanation. I'm still not sure if I would use the metric (I guess this is also a bit personal taste which should not be part of this review) but I found your paragraph about "We meticulously designed this metric to enable both absolute and relative reference points. " quite convincing. I would suggest adding it to your work.
> > >
> > > - about the example images:
> > >
> > > I think replacing would be a good compromise, however I have the fear that more image could be borderline. This comes back to the interaction of category 8 and the rest of the categories. You analyze human uncertain only on the uncertain images and thus assume all other categories to be perfect. I showed you that at least one person disagreed on one example. My question is would I find more such borderline image if I would check the rest of the test set further. Are your guidelines precise enough to cover all corner cases. Due to the fact that you have to check them only on your test set.  You may be able to achieve this seperation. I'm not happy with the design choice here but it might be just enough to be acceptable.

---

> > ### Comment · Reviewer_82Bt · 2023-08-28
> > **Reasons for increasing score**
> >
> > The current version clearly improved above the previous version. I will reraise my score.
> >
> > Tomorrow on the last day I will state my final score. Based on the given replies and edits to the paper I think a score between 4 and 6 (maybe 7) is likely.

---

> > > ### Author Response · Authors · 2023-08-28
> > > **Response to Reviewer 82Bt (1/2)**
> > >
> > > We again sincerely thank you for continuing to engage with us in this discussion, and we are especially grateful for the suggestions you have given us. This has been a very productive discussion so far, and we are truly fortunate to have you as a reviewer and to have such a meaningful discussion. Your feedback and comments indeed improved our work. The following are the responses to your new comments.
> > >
> > > **Q. I appreciate the added table and the explanation. I'm still not sure if I would use the metric (I guess this is also a bit personal taste which should not be part of this review) but I found your paragraph about "We meticulously designed this metric to enable both absolute and relative reference points. " quite convincing. I would suggest adding it to your work.**
> > >
> > > We are glad that we could make our argument more convincing. In fact, we have added this explanation to our revision, which can be found at line 315 in page 8 in the revised paper. Thank you for your suggestion.
> > >
> > > **Q. Interaction of category 8 and the rest of the categories, Some changes in category 3, lead to a change in class or make them uncertain.**
> > >
> > > We also agree with your concerns on both comments. However, as we have mentioned above, running a survey on all of our samples is challenging for realistic reasons such as time and money requirements (in our opinion, this is something that must be addressed somehow so that AI alignment research can keep up with the speed of AI development). We have done our best to discard disagreeable samples, and we have cross-validated our final selection. We did our best to remove corner case samples that may be disagreeable by at least one person. In line with our concerns, we have added this point in our limitations (line 415-419 page 10) and Appendix I.1.
> > >
> > > **Q. Let me rephrase where I see the difference to Peterson et al. and Schmarje et al. you create must abstain images especially examples with out of distribution properties such as category 5 and 7. Moreover you use the Hellinger distance which sounds to me more suitable to the task. What I'm missing is that is currently not well reflected in your work. I would like to propose the following change to your table 1 which would reflect this more. Req 5. Out of distribution images to test the robustness. (Please note that CIFAR10H is included in the work of Schmarje et al so (category 4 and 6 should also be included there).) This is just a proposal and you have to decide how and if you want to include it but the out of distribution argument is the first time that I thought this is the difference to previous work.**
> > >
> > > We thank you for the proposal of Requirement 5 and updates on Table 1. The “diverse scenarios” mentioned in Requirement 3 was also supposed to include out of distribution you mentioned in Requirement 5, but the meaning was not clearly delivered through our paper. We clarify the explanation of Requirement 3 in our revision (at line 137 page 4) and include Peterson et al in Table 1.
> > >
> > > As we update our Table 1, we want to highlight several points. In the case of Peterson et al. (CIFAR-10H), it does not satisfy Requirement 1 (clear definitions for each classes), as mentioned in the example as it is hard to distinguish between “car” and “truck” in line 126, and it partially satisfies Req. 3 (it covers some uncertainty but not out of distribution). Schemarje et al. also does not satisfy Requirement 1 (as it includes CIFAR10), and it partially satisfies Req. 3 for the same reason as Peterson et al.
> > >
> > > We also added that previous works did not test their models on out of distribution samples in related works which can be found in 105 page 3 in the revised paper.
> > >
> > > Thank you again for suggesting a new requirement and we again apologize for the unclarity of Requirement 3.
> > >
> > > **Q. The provided explanations reduce the issue but do not alleviate them. I still stand to the argument that including a more diverse uncertainties would improve your work. Due to the fact that we are getting close to the end of the discussion phase it is unrealistic that you add these elements. I wanted to ask for a more extended paragraph in the limitations which you already did. I guess at this point I would prefer more diverse uncertainties but is good enough for the moment.**
> > >
> > > We agree and are also aware that we cannot cover all cases of uncertainty by only using synthetic noise, as we mentioned in our previous response. The limitations regarding this issue are added in the limitation section (lines 419-423 page 10), and we added further discussions in the supplementary material. Please refer to Appendix I.2 for the above mentioned limitations.

---

> > > > ### Author Response · Authors · 2023-08-28
> > > > **Response to Reviewer 82Bt (2/2)**
> > > >
> > > > **Q. Thanks for the nice summary. This really helped me. I would like if you could include this (or a shorter version) in the paper but I can not see where (due to the page limit). If you find a way (maybe in the supplement) it would be helpful also to others. I like also to repeat your argument of out of distribution from above. I would urgue to include this argument. I could argue against some of the your difference or how others might do a bit better (e.g. 900 images times about 100 annotations are only 90,000 annotations which is do able even on a low budget). In general, these arguments feel a bit too much like personal taste and thus I would like to exclude them. If you rework the discussion around table 1, related work and your requirements based on our discussion, I can drop this issue. I see now that you have some difference and they now only have to be reflected in the work. This might not be easy due to the page limit but I'm certain you can do it and maybe you can shift the more lengthier arguments into the supplement.**
> > > >
> > > > Thank you for taking your time reading the long summary we provided. We would also like to add this to our paper, but due to the page limit, you can find this in the revised supplementary material Appendix I.1.
> > > >
> > > > We also included the argument regarding out of distribution features in related works (lines 103-108 of page 3), since it helped understand better the differences with previous works. Also, we added the three key differences between our dataset to the previous datasets (line 86 page 3) and inappropriateness of visual alignment measurement (test on only partial dataset, did not test on out of distribution samples, and use accuracy and KL divergence as metric) of previous works (line 103 page 3) in the related works. Due to space limitations, we have included detailed explanations about comparisons to previous works in Appendix J.
> > > >
> > > > Thank you for your feedback on telling us which of our arguments have been more persuasive.

---

> ### Comment · Reviewer_82Bt · 2023-08-29
> **Final Score**
>
> Dear Authors, I thank you for the fruitful discussion and the many revisions. I agree that you work improved significantly over time.
> I see some issue remaining like more annotations across all categories to quantify uncertainty. However, I believe this work should be accepted to NeurIPS in its current form. I was uncertain about the score 6 or 7 and decided in the end for the 7 because I wanted to honor the dedication and willingness to discuss multiple times. Congratulations to the work and good luck!

---

> > ### Author Response · Authors · 2023-08-29
> > **Response to Reviewer 82Bt**
> >
> > We sincerely thank you for your thorough reviews and the time you dedicated to our discussions. It was a valuable opportunity for us to engage in such high-quality discussions. Your feedback indeed enhanced the quality of our work. We are grateful that you appreciate our effort and for suggesting for acceptance at NeurIPS. We consider ourselves exceptionally fortunate to count you as our reviewer, engaging in such a multiple turns of high-quality discussions, which happens rarely in paper reviews.

---

### Official Review · Reviewer_PfuR · 2023-07-20
**Review of Paper752**

**Rating:** 6
**Confidence:** 3
**Correctness:** yes
**Clarity:** See Opportunities For Improvement and…

**Strengths:**

•	This paper is the first work to explore the novel model-human visual alignment problem.

•	The paper is generally well-organized and quite easy to follow.

•	The process of how to construct the dataset is detailly describe


**Additional Feedback:**

N/A

**Documentation:**

The code and dataset are not available for now. It will be good if the reviewer can see the dataset during the review process. The maintenance plan is not mentioned.

**Ethics:**

No concerns

**Limitations:**

This paper do not explicitly discuss the limitations.

**Opportunities For Improvement:**

1.	This paper does not discuss the reason why no current methods perform well across all categories. The exploration of the dataset should provide advice for proposing new methods.
2.	It would be good if more tasks, such as detection and segmentation, were discussed.


**Relation To Prior Work:**

yes

**Summary And Contributions:**

The paper proposes a benchmark to evaluate model-human visual alignment. The dataset contains Must-Act, Must-Abstain, and Uncertain groups. The popular visual perception models are evaluated on this benchmark.

---

> ### Author Response · Authors · 2023-08-24
> **Response to Reviewer PfuR**
>
> Thank you for the time and effort spent in carefully reviewing our work. Please kindly find the responses below.
>
> **Q. This paper does not discuss the reason why no current methods perform well across all categories. The exploration of the dataset should provide advice for proposing new methods.**
>
> Thank you for your suggestion. In the context of our study, as discussed in Section 5.2, distance-based methods (MD, KNN and TAPUDD) generally show high performance on the Must-Act categories. We conjecture the reason is that the distance-based methods infer a shorter distance between a class (one of 10 mammals) and the test image if the model recognizes at least one feature relevant to the said class. On the other hand, probability-based methods (SP and ASP) are strong on Must-Abstain categories. This is presumably because they abstain based on high entropy, which is reduced when the model is not only confident that its predicted class is correct but also that the remaining classes are incorrect. In short, the model must be confident that the predicted class is correct and that the remaining classes are incorrect, which is a challenging task for the model.  Furthermore, no method performed well on Uncertain. This is likely because approximating the probability across the 11 classes (10 mammals + abstention) is more challenging than predicting only one class as in Must-Act and Must-Abstain.
>
> Our experimental results showed that KNN had the best overall performance. We speculate that KNN, which calculates the sample-wise distance between the feature embeddings, captures the nuanced characteristics of the samples better and performs better than methods that calculate distances based on a clusters, which are MD and TAPUDD. We’ve added this insight in the revision in Section 5.2, line 369 page 10.
>
> Based on this observation, we believe a new abstention function which takes advantage of the strengths of both the distance-based and probability-based methods is needed to perform well on all categories of visual alignment. Once again, we sincerely thank you for your suggestion. We added the related analysis in Section 5.2 line 384 page 10 in the revised paper.
>
>
> **Q. It would be good if more tasks, such as detection and segmentation, were discussed.**
>
> We believe that the discussion of visual alignment can be further expanded to detection and segmentation tasks as well, which are also important tasks. However, for now, we focus on the most basic and fundamental task of object recognition as a starting point. As evident from the experimental results, there is still no method that achieves strong performance in all 8 aspects of visual alignment for object recognition. This suggests that there are challenges yet to be solved in object recognition.
>
> When advancing to image detection and segmentation, there are several challenges one may encounter. For instance, we collected more than 100 human labels for each image in the Uncertain group to minimize the error bound to below 5%. Similarly, segmentation will also need large amounts of human annotations to have valid gold human labels which will require much more resources than object recognition. Such a challenge not only exists in the visual alignment domain, but any AI alignment research that tries to address questions without absolute ground truths, as the ground truths must come from, ideally, the entire human population, or at least a statistically significant subset of the population (that’s why we hired more than 100 annotators per Uncertain sample). How to overcome this challenge in conducting AI alignment research will be an invaluable topic in the future. We added your suggestion as a possible future work in the revised paper which can be found at line 414 page 10.
>
> **Q. The code and dataset are not available for now. It will be good if the reviewer can see the dataset during the review process. The maintenance plan is not mentioned.**
>
> The code and dataset are publicly available now at https://github.com/jiyounglee-0523/VisAlign. For maintenance, we plan to upload a new version if any correction is needed. We mentioned the maintenance plan in the supplementary material, page 19 Section “A.7 Maintenance”.

---

### Official Review · Reviewer_9Ee9 · 2023-07-21
**The authors present a novel dataset for AI and human visual perception alignment, which is important for advancing and building interpretable artificial intelligence**

**Rating:** 7
**Confidence:** 4
**Correctness:** yes
**Clarity:** yes

**Strengths:**

The dataset fully considers multiple scenarios, in addition to easily categorizable data, generated based on spurious correlation, generated based on adversarial perturbation, and some confusing and difficult to categorize using a gold human perception label, which greatly facilitates the development of artificial intelligence.

**Additional Feedback:**

Currently only some regular scenarios are considered, is it possible to consider some unique human visual phenomena such as visual illusion, and further consider some scenarios where people would make mistakes, but maybe deep learning would not.

**Documentation:**

yes

**Limitations:**

yes

**Opportunities For Improvement:**

Currently only some regular scenarios are considered, is it possible to consider some unique human visual phenomena such as visual illusion, and further consider some scenarios where people would make mistakes, but maybe deep learning would not.

**Relation To Prior Work:**

yes

**Summary And Contributions:**

The authors propose a dataset for measuring the degree of alignment between AI and humans in visual perception. The dataset consists of three groups, Must-Act, Must-Abstain and Uncertain, and all the samples have a gold human perception label. The authors use verify their dataset with several famous algorithms.

---

> ### Author Response · Authors · 2023-08-24
> **Response to Reviewer 9Ee9**
>
> Thanks again for the time and effort to review our work and provide valuable comments.
>
> **Q. Currently only some regular scenarios are considered, is it possible to consider some unique human visual phenomena such as visual illusion, and further consider some scenarios where people would make mistakes, but maybe deep learning would not.**
>
> Thank you for your insightful and very relevant feedback. The consideration of visual illusions, or more broadly, “visual signals that confuse humans but not machines” is certainly an interesting yet controversial topic in our opinion. For example, given the “Checker shadow illusion (https://en.wikipedia.org/wiki/Checker_shadow_illusion)”, uninformed humans will always be confused, but machines that can calculate pixel values will never be confused. Then can we call this machine to be “aligned” to humans? Do we want machines never to be fooled by the visual signals that humans are always fooled by? Wouldn’t such machines often behave in a way that is startling and incomprehensible to humans? Human visual perception has evolved in this particular way (i.e. to be fooled by the Checker shadow illusion), because it was beneficial to our survival. But suddenly, do we want the machines to never be fooled by such phenomena and act in their own non-human way?
>
> In our estimation, the abstract definition and somewhat ambiguous objective of the “alignment research” are the source of this debatable argument. What is the objective of the “AI Alignment”? Do we want machines to always act/think in a similar manner as humans? Or do we want machines to surpass humans whenever/wherever possible? Some might say that the machines should be allowed to possess superhuman capabilities, but must only use their capabilities when it’s beneficial for humans. Then who decides what is beneficial to humans? Do the machines decide what is good for humans before using their superior capabilities?
>
> Such are questions that are not answerable at this moment, probably not in this year, or not even in this decade. As the first research to claim that machine’s visual perception should be “aligned” to human visual perception, we admit our scope is relatively narrow in that we only deal with mammal recognition in various settings. This was not an arbitrary choice, but a careful design based on thorough discussion as to what it means for machines to be “aligned” to humans. And our current best answer to the question “To what extent should the machines’ visual perception be aligned with human visual perception?” is “At least for simple object recognition in eight different settings, machines should show similar behavior as humans”. We believe both the question and the answer to “AI Alignment” will be more refined as the AI community continues its research effort in the time to come. We added a dedicated section in the supplementary material (Section E) to discuss how we view AI-human visual alignment in the context of AI’s superhuman capabilities.

---

### Author Response · Authors · 2023-08-24
**Thank you for the reviews**

Dear reviewers and ACs,

We thank all the reviewers for their insightful feedback. We have addressed each comment from the reviewers and made the necessary revisions to our paper based on their recommendations. Please kindly find the responses below.

---

### Decision · Program_Chairs · 2023-09-22

**Decision:**

Accept (Poster)

**Comment:**

All the reviewers have voted for accept. Congrats!